# FoSSIL: A Unified Framework for Continual Semantic Segmentation in 2D and 3D Domains

## Abstract

Evolving visual environments challenge continual semantic segmentation by introducing the complexities of *class-incremental learning*, *domain incremental learning*, *limiting available annotations*, and necessitating the use of unlabeled data. In this work, we present the framework FoSSIL (**F**ew-sh**o**t **S**emantic **S**egmentation for **I**ncremental **L**earning), which extensively benchmarks continual semantic segmentation, spanning both 2D natural scenes and 3D medical volumes. Our evaluation encompasses diverse and realistic settings, leveraging both labeled (few-shot) and unlabeled data. Building on this benchmark, we introduce *guided noise injection* to mitigate overfitting due to novel few-shot classes from various domains. Furthermore, we leverage semi-supervised learning for unlabeled data to augment few-shot novel classes. We propose a *filtering* mechanism to remove highly confident but incorrectly predicted pseudo-labels, further improving performance. Results across class-incremental, few-shot, and domain incremental scenarios with unlabeled data validate our strategies for robust semantic segmentation in complex, evolving settings, highlighting both the effectiveness and generality of our approach. Our findings illustrate that the proposed framework forms a simple yet powerful recipe for continual semantic segmentation in dynamic real-world environments. Our large-scale benchmarking across natural 2D and medical 3D domains exposes key failure modes of existing methods and offers a roadmap for building robust continual segmentation models.

## 1 Introduction

The pursuit of truly intelligent systems necessitates continuous learning and adaptation in open-world environments. While continual learning (CL) (Wang et al. (2024); Yuan & Zhao (2024b)) has advanced across various tasks, a critical gap remains in addressing the significant complexities of real-world dense prediction tasks like semantic segmentation. In demanding applications such as autonomous driving and medical image analysis, semantic segmentation models are confronted with a continuous influx of data characterized by both novel semantic categories or *class-incremental learning* (CIL Zhou et al. (2024)) and evolving data distributions or *domain incremental learning* (DIL Mirza et al. (2022)), posing a formidable challenge to their adaptability and robustness.

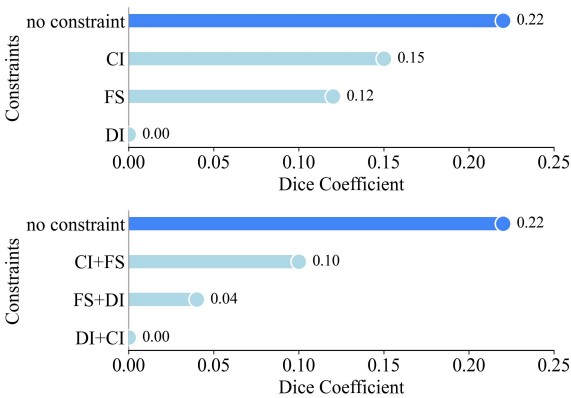

Figure 1: Class-incremental (CI), few-shot (FS), and domain-incremental (DI) constraints all lead to significantly reduced Dice scores compared to the unconstrained fine-tuning ("no constraint") on the common base model.

This discrepancy between idealized CL scenarios and real-world semantic understanding poses significant challenges. In the realistic setting of continual learning, prevalent CL methods struggle with catastrophic forgetting, significantly exacerbated by shifts in the semantic label space and input

data characteristics. Furthermore, data scarcity necessitates effective *few-shot learning* (Tao et al. (2020); Qiu et al. (2023); Tian et al. (2024)) within these continuous learning streams. The confluence of these factors creates a challenging landscape where models must rapidly adapt to new concepts with limited supervision while preserving previously acquired knowledge. Specifically, the need to balance plasticity (acquiring new knowledge) and stability (retaining old knowledge) becomes paramount, yet exceedingly difficult, under these conditions. Figure 1 shows that the base model achieves its best performance under unconstrained fine-tuning, but suffers substantial accuracy drops under class-incremental (CI), few-shot (FS), or domain-incremental (DI) settings. Each constraint stresses the model differently. CI shifts decision boundaries, FS leads to unstable gradients and overfitting, and DI distorts feature distributions, leading to large performance degradations relative to the unconstrained baseline. Leveraging unlabeled data through *semi-supervised learning* (Kang et al. (2023b); Cui et al. (2024)) holds immense potential. However, the dynamic introduction of novel classes complicates the reliable utilization of pseudo-labels, as initial model biases can lead to the propagation of incorrect information. A key challenge is that this initial bias can accumulate, steadily degrading the model's ability to learn.

Many works and surveys Zhang et al. (2025); Yuan & Zhao (2024a); Xu et al. (2024); Kwak et al. (2025); Zhu et al. (2025b); Xue et al. (2025)) emphasize the need for realistic CL settings that jointly handle class-increments, domain shifts, and few-shot data, with unlabeled data.

Despite its practical significance, the realistic setting of few-shot learning and semi-supervised learning for complex tasks like semantic segmentation remains largely underexplored in the context of continual learning. Existing CL methods, often evaluated on simpler tasks, are not designed to handle these complexities. The core challenge lies in developing a learning framework that can effectively handle the data distributions that change over time and the need for rapid adaptation to new classes with limited labeled data, while simultaneously mitigating catastrophic forgetting and the propagation of errors from noisy pseudo-labels.

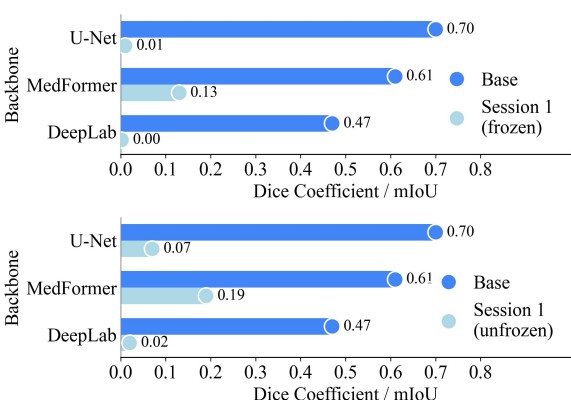

Figure 2: Fine-tuning the Session 0 base model in incremental Session 1 causes all backbones to drop in performance, whether their weights are partially frozen (top) or fully unfrozen (bottom).

This work directly confronts these critical, largely unaddressed challenges in continual learning for semantic segmentation. Our proposed framework **FoSSIL** (**F**ew-sh**o**t **S**emantic **S**egmentation for **I**ncremental **L**earning) investigates continual learning in realistic and demanding scenarios where semantic classes and data domains evolve and may *reappear* over time, constrained by few-shot data within each learning session. Crucially, FoSSIL models a realistic setting in which each incremental session introduces some form of novelty, such as a new class in a previously seen domain, a previously seen class in a new domain, or a new class in a new domain, while ensuring that a class and domain pair that has already appeared does not occur again. Limited supervision, especially under domain shift, causes the model to overfit quickly and destabilizes adaptation. To reduce this instability, we introduce *guided noise injection*, which uses gradient sensitivity to regulate noise on classifier weights and prevent overfitting during adaptation. Building on this foundation, FoSSIL combines an exemplar-free prototype replay strategy with guided noise injection and a *refinement mechanism* for pseudo-labels in a semi-supervised setting that leverages abundant unlabeled data. This enables reliable learning, robustness, and improved retention of previously acquired knowledge across both 2D and 3D segmentation architectures.

To ground our contributions, we conduct extensive benchmarking that spans both 2D natural and 3D medical domains, systematically evaluating class-incremental, domain incremental, few-shot, and semi-supervised continual learning settings. FoSSIL benchmarks a wide spectrum (around *thirty-seven*) of state-of-the-art methods with detailed ablations on *twelve* datasets with different *backbones* like U-Net (Ronneberger et al. (2015)), DeepLabv3+ (Chen et al. (2018)) and Transformers-based (Kirillov et al. (2023)). It highlights key failure modes in current approaches,

such as overfitting in few-shot regimes, difficulties in adapting across domains, and error amplification from pseudo-labels, and proposes novel strategies to overcome them. We find that fine-tuning popular backbones (e.g., U-Net, DeepLabv3+, MedFormer Gao et al. (2022)) on novel few-shot classes from varied domains, whether with partially frozen or fully unfrozen weights, leads to severe performance drops in incremental sessions, highlighting the severity of the problem. Figure 2 shows that the naive fine-tuning of the base session model is unstable across all backbones. Freezing most layers in Session 1 limits the model's ability to learn new features, while fully unfreezing shifts the representation space and disrupts earlier classes. These drops indicate that neither freezing nor fully unfreezing trivially is reliable and that incremental training requires preserving earlier representations while allowing controlled adaptation.

Our key contributions to multi-constrained continual learning for semantic segmentation are fourfold: **(i) Firstly,** we introduce the FoSSIL framework, a comprehensive continual segmentation benchmark spanning twelve 2D natural and 3D medical datasets. We *re-implemented* and *adapted* thirty-seven relevant methods, revealing fundamental limitations of existing approaches. **(ii) Secondly,** we propose an exemplar-free prototype replay strategy for both class- and domain-incremental few-shot learning, improving memory efficiency and privacy, and introduce a guided noise injection mechanism that enhances generalization to new classes and domains. **(iii) Thirdly,** we integrate *semi-supervised* learning to expand few-shot classes using unlabeled data, employing learned prototypes to filter erroneous high-confidence pseudo-labels and thereby improve supervisory quality. **(iv) Finally,** our approaches exhibit strong cross-architecture generalization (3D U-Net, DeepLabv3+, Transformers), surpassing even large-scale pre-trained models like SAM (Kirillov et al. (2023); Kerssies et al. (2024)) on the FoSSIL benchmark.

## 2 RELATED WORK

**Class-Incremental/Domain Incremental Learning in Semantic Segmentation:** UCL Ahn et al. (2019) applies uncertainty-aware regularization to reduce forgetting. MiB Cermelli et al. (2020) manages background shift through distillation and classifier initialization. CLIP-CT Zhang et al. (2023) adapts new classes using pseudo-labeling and CLIP-guided heads. MDIL Garg et al. (2022) decomposes parameters into domain-invariant and domain-specific components with adaptive distillation. C-Flat Bian et al. (2024) encourages flat minima, UCB Liang et al. (2025) adjusts pseudo-labels via uncertainty and class balance estimates, and STAR Eskandar et al. (2025) stabilizes learning through parameter-space perturbations.

**Few-shot Class-Incremental Learning:** Cermelli et al. (2021) introduced PIFS, combining prototype learning with distillation to learn new classes from few samples without storing old data. Subspace Akyürek et al. (2021) constrains novel class weights to the base-class subspace to reduce forgetting and overfitting. C-FSCIL Hersche et al. (2022) uses hyperdimensional representations with a frozen meta-learned encoder and a growing set of quasi-orthogonal prototypes. FACT Zhou et al. (2022) allocates embedding space for future classes via virtual prototypes and manifold mixup. Gen-Replay Liu et al. (2022) performs data-free replay by synthesizing uncertain samples of prior classes. NC-FSCIL Yang et al. (2023) fixes classifier prototypes as a simplex ETF. GAPS Qiu et al. (2023) handles partial annotations through copy–paste synthesis to generate dense labels. SoftNet Kang et al. (2023a) decomposes networks into major/minor subnetworks via adaptive soft masks for stability and fast adaptation. FSCIL-SS Jiang et al. (2023) combines pseudo-labeling and distillation to learn novel classes under limited supervision. FeCAM Goswami et al. (2023) is exemplar-free, using class covariances and Mahalanobis distance for improved prototype classification. BCM Sakai et al. (2024) mines base classes most relevant to novel-class prediction.

**Semi-Supervised Learning based approaches:** Killamsetty et al. (2021) proposed RETRIEVE, a coreset selection method for efficient and *robust* semi-supervised learning. NNCSL Kang et al. (2023b) uses soft nearest neighbors to stabilize continual semi-supervised learning by reducing forgetting on unlabeled features and overfitting on scarce labels. UaD-CE Cui et al. (2024) introduces uncertainty-aware distillation with class equilibrium to generate balanced pseudo-labels and transfer knowledge from reliable exemplars. CSL Liu & Liu (2025) improves pseudo-label selection by separating reliable and unreliable predictions in a confidence–distribution space.

**Miscellaneous:** Khosla et al. (2020) introduced SupCL, extending contrastive learning with labels for stronger representations. SimCLR Chen et al. (2020) learns features by aligning augmented views without labels, and UnSupCL-HNM Robinson et al. (2021) enhances this via hard negative

mining. Wang et al. (2021) unified multi-task learning and gradient-based meta-learning under a shared optimization framework. Bouniot et al. (2022) analyzed few-shot learning through multi-task representation theory, contrasting gradient- and metric-based approaches. Liu et al. (2023) proposed a CLIP-driven universal model for multi-organ segmentation using CLIP (Radford et al. (2021)) text embeddings to handle partial labels. HALO Franco et al. (2024) uses hyperbolic networks for pixel-level active learning under domain shift.

**To the best of our knowledge, FoSSIL is the first framework to jointly address CIL, DIL, and few-shot semantic segmentation by leveraging unlabeled data to strengthen scarce classes.**

## 3 FoSSIL Framework

We formalize the multi-constrained continual learning problem for semantic segmentation as a sequence of sessions $\mathcal{S} = \{\mathcal{S}_0, \mathcal{S}_1, \mathcal{S}_2, \ldots, \mathcal{S}_T\}$ where each session $\mathcal{S}_t$ is characterized by both a semantic class space $\mathcal{C}_t$ and a domain distribution $\mathcal{D}_t$. At each session $\mathcal{S}_t$, a learning model encounters a dataset $\mathbb{D}_t = \{(x_i, y_i)\}_{i=1}^{N_t}$ where $x_i \in \mathcal{X}$ represents input images drawn from the domain $\mathcal{D}_t$ and $y_i \in \mathcal{Y}_t \subseteq \mathcal{C}_t$ denotes the pixel-wise semantic labels, with $|\mathbb{D}_t| = N_t$ being the number of available samples in session $t$. $\mathcal{S}_0$ denotes the base session, which contains domain(s) with abundant labeled data ($N_0$), while the remaining sessions are few-shot sessions with sparsely labeled domains.

### 3.1 Classes, Domains, and Data Across Sessions

The continual learning sequence accommodates three realistic scenarios that distinguish our framework from idealized settings. **Same Classes, Different Domains:** Different sessions may share semantic classes while those classes belong to different domains: $\mathcal{C}_t = \mathcal{C}_{t'}$ and $\mathcal{D}_t \neq \mathcal{D}_{t'}$. **Different Classes, Same Domain:** New classes may be introduced across sessions belonging to the same domain: $\mathcal{C}_t \neq \mathcal{C}_{t'}$ and $\mathcal{D}_t = \mathcal{D}_{t'}$. **Different Classes, Different Domain:** The most challenging scenario with both semantic and domain shift: $\mathcal{C}_t \neq \mathcal{C}_{t'}$ and $\mathcal{D}_t \neq \mathcal{D}_{t'}$. Importantly, we explicitly exclude the case where same classes of the same domain are repeated across sessions.

Each incremental session ($t \neq 0$) is trained on few-shot labeled data, where $N_t \ll N_0$ and $N_t = K \cdot |\mathcal{C}_t|$. Here, $K$ denotes the number of labeled examples per class (typically $K \in \{5, 10, 20, 30\}$). In addition, each session may also access unlabeled data, defined as $\mathcal{U}_t = \{x_j^{(u)}\}_{j=1}^{M_t}$, where $M_t \gg N_t$ indicates the number of unlabeled samples drawn from the same domain $\mathcal{D}_t$.

### 3.2 Exemplar-free prototype Replay

For each class $c \in \mathcal{C}_t$, we extract a compact prototype from the feature space. Given a trained model with intermediate feature extractor $\phi$. For each sample $(x_i, y_i)$, embedding $E_i = \phi(x_i) \in \mathbb{R}^{D \times H \times W}$.

In session $t$, for class $c$, we extract features corresponding to class pixels as $\mathcal{F}_c^{(i)} = \{E_i : y_i = c\}$.

The class prototype is given as,

$$p_c^{(i)} = \frac{1}{|\mathcal{F}_c^{(i)}|} \sum_{\mathbf{f} \in \mathcal{F}_c^{(i)}} \mathbf{f} \tag{1}$$

Across all samples in session $t$ containing class $c$:

$$P_c = \frac{1}{NS_c} \sum_{j \in NS_c} \frac{p_c^{(j)}}{||p_c^{(j)}||_2} \tag{2}$$

where $NS_c \subseteq N_t$ is the number of samples containing class $c$.

In session $t + 1$, we replay all prototypes learned from previous session, $\mathcal{P}_t = \{P_c\}$ for all $c \in \mathcal{C}_t$ as,

$$\mathcal{L}_{\text{proto}} = \sum_{P_c \in \mathcal{P}_t} \mathcal{L}_{\text{CE}}(F(P_c), c) \tag{3}$$

where $F$ is the final classifier layer and $\mathcal{L}_{\text{CE}}$ is the cross-entropy loss. $\mathcal{L}_{\text{proto}}$ optimizes the model parameters in session $t + 1$. We store a single prototype (a few KBs) for each class when it first appears and reuse the same prototype in all later sessions.

Figure 3: At session $t$, lightweight prototypes ($\mathcal{P}_{t-1}$) from sessions $0, \ldots, t-1$ are replayed at the pixel classifier layer $F$ (loss $\mathcal{L}_{\text{proto}}$) of the decoder of segmentation model $M_s$, where guided noise injection is applied to improve robustness. When unlabeled data is present, a mean-teacher model with teacher network $M_t$ generates pseudo-labels via similarity matching (threshold $\tau_{\text{sim}}$), and only high-confidence predictions ($\tau_{\text{conf}}$) are used through a consistency loss.

### 3.3 GUIDED NOISE INJECTION (GNI)

The guided noise injection mechanism regulates noise using parameter gradients, which serve as a proxy for determining the appropriate magnitude of noise to add to each parameter.

The method maintains a gradient buffer $G$ that accumulates squared gradients $(\nabla_{w_{ij}} \mathcal{L})^2$ where $\mathcal{L}$ is the loss function and $w_{ij}$ are the weight parameters of the classifier layer $F$ with weight matrix $\mathbb{W}$.

For any $G_{ij} \in G$ the inverse is computed as $G_{ij}^{-1} = \frac{1}{G_{ij}+\epsilon}$ where $\epsilon = 10^{-8}$ ensures numerical stability. To control the noise magnitude, the inverse gradients are normalized to a bounded range:

$$\tilde{G}_{ij}^{-1} = \frac{1 + G_{ij}^{-1} - \min(G^{-1})}{1 + \max(G^{-1}) - \min(G^{-1})} \tag{4}$$

The weights $\mathbb{W}$ are perturbed as:

$$\tilde{\mathbb{W}} = \mathbb{W} + \tilde{G}^{-1} \odot \mathcal{N}(0, I) \tag{5}$$

Hence the noise $\mathcal{N}(0, I)$ added to $\mathbb{W}$ is guided by $\tilde{G}^{-1}$.

Large gradients correspond to low noise injection, whereas small gradients allow for higher noise injection. Critical weight parameters with large gradients, which are actively contributing to learning, receive minimal noise injection, whereas parameters that have begun overfitting and no longer contribute significantly are injected with higher noise for regularization. To our knowledge, GNI represents the first gradient-guided stochastic perturbation method designed for domain-adaptive, continual, and few-shot semantic segmentation. Please refer to Table 8 and Appendix D for *robustness and sensitivity analysis* of guided noise injection.

### 3.4 PROTOTYPE-GUIDED PSEUDO-LABEL REFINEMENT

To leverage abundant unlabeled data while mitigating the risk of noise and error propagation, we introduce a prototype-guided pseudo-label refinement (or filtering) strategy within a mean-teacher (Tarvainen & Valpola (2017)) based framework. To mitigate confirmation bias in standard pseudo-labeling, we introduce a mechanism that validates pseudo-labels through both predictive confidence and feature-space consistency with prototypes.

For an unlabeled input $x_j^{(u)}$, the student network $M_s$ and teacher network $M_t$ generate pseudo-label predictions, denoted by $\hat{y}_s$ and $\hat{y}_t$, respectively:

$$\hat{y}_s, \mathcal{F}_s = M_s(x_j^{(u)}), \quad \hat{y}_t, \mathcal{F}_t = M_t(x_j^{(u)}) \tag{6}$$

where $\mathcal{F}_s$ and $\mathcal{F}_t$ are features representations learned by student and teacher, respectively.

We compute the confidence of the predictions ($\hat{y}_s$ and $\hat{y}_t$) as,

$$c'(p,q) = \arg\max_c(\text{softmax}(\hat{y}(p,q))), \quad \text{conf}(p,q) = \max(\text{softmax}(\hat{y}(p,q))) \tag{7}$$

where $c'(p,q)$ is class for pixel $(p,q)$, $\text{conf}(p,q)$ is confidence of class $c'(p,q)$ and $\hat{y}(p,q)$ is pseudo-label at pixel $(p,q)$.

To validate predictions, the cosine similarity between features and prototypes is computed as,

$$\text{sim}(p,q) = \frac{\mathcal{F}(p,q) \cdot P_{c'(p,q)}}{||\mathcal{F}(p,q)||_2 ||P_{c'(p,q)}||_2} \tag{8}$$

where $\mathcal{F}(p,q)$ is feature of the pixel $(p,q)$ and $P_{c'(p,q)}$ is prototype corresponding to the predicted class $c'(p,q)$.

A pseudo-label at $(p,q)$ is retained only if it satisfies the following conditions:

$$\text{valid}(p,q) = (\text{conf}(p,q) > \tau_{\text{conf}}) \quad \text{and} \quad (\text{sim}(p,q) > \tau_{\text{sim}}) \tag{9}$$

where $\tau_{\text{conf}}$ and $\tau_{\text{sim}}$ are empirically determined thresholds.

The consistency loss in mean-teacher operates only on validated pseudo-labels:

$$\mathcal{L}_{\text{consistency}} = \frac{1}{|\mathcal{V}|} \sum_{(p,q) \in \mathcal{V}} ||\hat{y}_s(p,q) - \hat{y}_t(p,q)||_2^2 \tag{10}$$

where $\mathcal{V} = \{(p,q) : \text{valid}_s(p,q) \quad \text{and} \quad \text{valid}_t(p,q)\}$ represents pixels validated by both models.

This verification mechanism reduces pseudo-label noise by requiring both high prediction confidence and feature-space similarity to class prototypes. This ensures the pseudo-labels are robust to domain shifts, preventing the amplification of errors in the student-teacher feedback loop. The complete FoSSIL framework is illustrated in Figure 3. Earlier prototype-replay methods (Chen et al. (2023); Kong et al. (2025); Zhu et al. (2025a)) rely on prototypes solely for replay, while FoSSIL uses them for both replay and pseudo-label validation, a dual use absent in previous works. To our knowledge, no existing pseudo-labeling or regularization method provides prototype filtering, cross-session stability, and gradient-guided noise control for multi-constraint continual learning.

### 3.5 THEORETICAL ANALYSIS

GNI assigns each parameter $w_i$ a noise scale $\tilde{G}_i$ inversely proportional to its instantaneous squared gradient, and performs the update $\widetilde{w}_i = w_i + \tilde{G}_i \xi_i$ with $\xi_i \sim \mathcal{N}(0,1)$. This yields a parameter-wise stability–plasticity mechanism. Parameters with large gradients receive small $\tilde{G}_i$, ensuring stability, while parameters with small gradients receive larger $\tilde{G}_i$, promoting plasticity. A second-order expansion of the perturbed loss (Appendix A) introduces the term $\sum_i H_{ii} \tilde{G}_i^2$, which is minimized because $\tilde{G}_i$ is small precisely where curvature $H_{ii}$ is large. Thus GNI protects sharp directions and explores flat directions. From a Bayesian view (Appendix B), the resulting diagonal posterior $q = \mathcal{N}(\mathbb{W}, \text{diag}(\tilde{G}_i^2))$ achieves a smaller $\text{KL}(q\|p)$ than any isotropic posterior with matched variance, yielding a tighter PAC–Bayes generalization bound, where $p$ is the prior distribution over the model parameters. In Appendix C, we show that prototype-guided refinement stabilizes pseudo-label errors and lowers asymptotic error in mean-teacher models.

## 4 FoSSIL BENCHMARKS

We construct five challenging benchmarks for **3D medical** and **2D natural scene segmentation**, designed to simulate realistic clinical and autonomous driving scenarios with *multiple sessions*, *diverse domains*, and a *large number of novel classes*. Each benchmark features a base learning

Table 1: Summary of FoSSIL Benchmarks. $|\mathcal{C}_t|$ denotes the number of classes in session $i$. 'SS' denotes Semi-Supervised.

| Benchmark | Session 0 (Base) | Session 1 | Session 2 | Session 3 | Session 4 | Session 5 |
|---|---|---|---|---|---|---|
| Med FoSSIL-Disjoint | $|\mathcal{C}_0| = 15$ (TS) | $|\mathcal{C}_1| = 5$ (AMOS) | $|\mathcal{C}_2| = 6$ (BCV) | $|\mathcal{C}_3| = 4$ (MOTS) | $|\mathcal{C}_4| = 3$ (BraTS) | $|\mathcal{C}_5| = 4$ (VerSe) |
| Med FoSSIL-Mixed | $|\mathcal{C}_0| = 10$ (AMOS) | $|\mathcal{C}_1| = 8$ (BCV, MOTS) | $|\mathcal{C}_2| = 6$ (TS, AMOS) | $|\mathcal{C}_3| = 4$ (MOTS, TS) | $|\mathcal{C}_4| = 7$ (BraTS, VerSe) | – |
| Med SS-FoSSIL | $|\mathcal{C}_0| = 15$ (TS) | $|\mathcal{C}_1| = 5$ (AMOS) | $|\mathcal{C}_2| = 6$ (BCV) | $|\mathcal{C}_3| = 4$ (MOTS) | $|\mathcal{C}_4| = 3$ (BraTS) | $|\mathcal{C}_5| = 4$ (VerSe) |
| Natural-FoSSIL | $|\mathcal{C}_0| = 10$ (BDD) | $|\mathcal{C}_1| = 5$ (IDD) | $|\mathcal{C}_2| = 5$ (BDD, IDD) | – | – | – |
| SS-Natural-FoSSIL | $|\mathcal{C}_0| = 10$ (BDD) | $|\mathcal{C}_1| = 2$ (Cityscapes) | $|\mathcal{C}_2| = 2$ (IDD) | $|\mathcal{C}_3| = 3$ (IDD) | – | – |

Table 2: Performance of baselines on the 3-session Med FoSSIL-Disjoint benchmark, reported as Dice (0–1).

| Method | Session 0 | Session 1 | Session 2 |
|---|---|---|---|
| **PIFS** Cermelli et al. (2021) | 0.700 | 0.129 | 0.078 |
| **NC-FSCIL** Yang et al. (2023) | 0.394 | 0.077 | 0.081 |
| **CLIP-CT** Zhang et al. (2023) | 0.475 | 0.186 | 0.141 |
| **MiB** Cermelli et al. (2020) | 0.700 | 0.271 | 0.096 |
| **MDIL** Garg et al. (2022) | 0.779 | 0.115 | 0.097 |
| **C-FSCIL** Hersche et al. (2022) | 0.787 | 0.334 | 0.297 |
| **SoftNet** Kang et al. (2023a) | 0.820 | 0.305 | 0.146 |
| **GAPS** Qiu et al. (2023) | 0.700 | 0.334 | 0.253 |
| **FSCIL - SS** Jiang et al. (2023) | 0.700 | 0.115 | 0.089 |
| **Subspace** Akyürek et al. (2021) | 0.257 | 0.054 | 0.040 |
| **Gen-Replay** Liu et al. (2022) | 0.700 | 0.076 | 0.102 |
| **FeCAM** Goswami et al. (2023) | 0.700 | 0.048 | 0.042 |
| **FACT** Zhou et al. (2022) | 0.357 | 0.071 | 0.028 |
| **MAML** Bouniot et al. (2022) | 0.700 | 0.001 | 0.059 |
| **MAML + regularizer** Bouniot et al. (2022) | 0.700 | 0.001 | 0.062 |
| **MTL** Wang et al. (2021) | 0.700 | 0.079 | 0.088 |
| **UnSupCL** Chen et al. (2020) | 0.700 | 0.039 | 0.088 |
| **SupCL** Khosla et al. (2020) | 0.700 | 0.058 | 0.042 |
| **UnSupCL-HNM** Robinson et al. (2021) | 0.700 | 0.035 | 0.068 |
| **C-Flat** Bian et al. (2024) | 0.700 | 0.174 | 0.030 |
| **STAR** Eskandar et al. (2025) | 0.700 | 0.050 | 0.020 |
| **Saving100x** Chen et al. (2023) | 0.700 | 0.072 | 0.053 |
| **YoooP** Kong et al. (2025) | 0.700 | 0.176 | 0.028 |
| **UCB** Liang et al. (2025) | 0.700 | 0.267 | 0.127 |
| **BCM** Sakai et al. (2024) | 0.700 | 0.014 | 0.000 |
| **Adapt_replay** Zhu et al. (2025a) | 0.700 | 0.044 | 0.027 |
| **UCL** Ahn et al. (2019) | 0.700 | 0.430 | 0.325 |
| **FoSSIL (U-Net)** | 0.736 | **0.460** | **0.398** |

session on a large dataset followed by incremental sessions with limited labeled data (few-shot) and with significant domain shifts.

**3D Medical FoSSIL Benchmarks**: We develop three **3D medical** benchmarks using data from **TotalSegmentator** (TS - CT) (Wasserthal et al. (2023)), **AMOS** (mostly CT with few MRI samples) (Ji et al. (2022)), **BCV (CT)**(Landman et al. (2015), **MOTS (CT)** (Zhang et al. (2021)), **BraTS (MRI)** (Menze et al. (2014)), and **VerSe (CT)** (Sekuboyina et al. (2021)). All three benchmarks adopt a few-shot learning setup, using 5 training samples per class for incremental sessions, progressing from normal to tumor segmentation.

The three medical benchmarks: (i) **Med FoSSIL-Disjoint**, a 6-session, 37-class protocol with disjoint classes and domains; (ii) **Med FoSSIL-Mixed**, a 5-session, 35-class setup allowing recurrence of either classes or domains (but not both) and mixing datasets per session; and (iii) **Med Semi-Supervised-FoSSIL**, a semi-supervised variant of Med FoSSIL-Disjoint augmented with 8–30 unlabeled samples per session. Please refer to Table 1 for various classes and domains.

**2D Natural Scene FoSSIL Benchmarks**: We introduce two benchmarks for **autonomous driving** scenarios using data from **BDD100K** (Yu et al. (2020)), **Cityscapes** (Cordts et al. (2016)), and **IDD** (Varma et al. (2019)). These benchmarks feature a few-shot learning with 10 training samples per class. The two natural scene benchmarks for autonomous driving: (i) **Natural-FoSSIL**, a 3-session setup using BDD100K, Cityscapes, and IDD, designed to test representation adaptation under domain shifts and class recurrence; and (ii) **Semi-Supervised Natural-FoSSIL**, a 4-session variant that augments new classes with 400 unlabeled images per class to reflect realistic scenarios with limited annotations but abundant raw data. Details on all datasets, class and domain specifications, and unlabeled data are provided in Appendix G.

Table 3: Performance on the 6-session Med FoSSIL-Disjoint benchmark, reported as Dice (0–1).

| Method | Session 0 | Session 1 | Session 2 | Session 3 | Session 4 | Session 5 |
|---|---|---|---|---|---|---|
| **U-Net Vanilla** | 0.700 | 0.076 | 0.057 | 0.047 | **0.030** | 0.042 |
| **FoSSIL (U-Net)** | 0.736 | **0.460** | **0.398** | **0.329** | 0.025 | **0.324** |

Table 4: Performance on the Natural-FoSSIL benchmark, reported as mIoU (0–100).

| Method | Session 0 | Session 1 | Session 2 |
|---|---|---|---|
| **DeepLab Vanilla** | 47.76 | 2.18 | 3.86 |
| **GAPS** Qiu et al. (2023) | 47.76 | 23.42 | 16.68 |
| **MiB** Cermelli et al. (2020) | 47.76 | 2.50 | 2.37 |
| **MDIL** Garg et al. (2022) | 48.54 | 1.59 | 3.02 |
| **SAM Vanilla** Kerssies et al. (2024) | 66.0 | 32.6 | 30.81 |
| **FoSSIL (SAM)** | 66.0 | **33.2** | **31.22** |

Table 5: Performance on Med FoSSIL-Mixed benchmark. All values are reported as Dice coefficients (0-1).

| Method | Session 0 | Session 1 | Session 2 | Session 3 | Session 4 |
|---|---|---|---|---|---|
| **U-Net Vanilla** | 0.571 | 0.216 | 0.133 | 0.074 | 0.045 |
| **CLIP-driven** Liu et al. (2023) | 0.717 | **0.417** | 0.227 | 0.196 | 0.089 |
| **MedFormer Vanilla** | 0.613 | 0.198 | 0.134 | 0.052 | 0.067 |
| **SwinUNetr Vanilla** | 0.605 | 0.197 | 0.133 | 0.082 | 0.082 |
| **FoSSIL (SwinUNetr)** | 0.605 | 0.318 | 0.275 | 0.254 | 0.210 |
| **FoSSIL (MedFormer)** | 0.622 | 0.367 | **0.287** | **0.288** | **0.228** |

## 5 RESULTS AND ANALYSIS

We use mean Intersection over Union (mIoU), ranging from 0 to 100, for 2D natural scene benchmarks, and the Dice coefficient (Dice score), ranging from 0 to 1, for 3D medical benchmarks, as evaluation metrics. In each *incremental session*, we evaluate the classes introduced in the current session along with all classes encountered in previous sessions. The goal is to retain previously learned knowledge while effectively acquiring new information, handling data scarcity, and adapting to domain shifts. A *Vanilla* baseline consists of a plain backbone with no mechanisms to handle any constraints. Gen-Replay Liu et al. (2022) is implemented with a diffusion model adapted from Dorjsembe et al. (2024). Some baselines act only in incremental sessions and rely on a shared pre-trained base model, whereas others modify training in Session 0, thereby altering base session performance. See Appendix J for comparison with regularization-based methods.

We observe substantial performance drops across all Vanilla models and baselines. Notably, transformer-based models perform comparably to, or sometimes worse than, simpler encoder–decoder architectures like U-Net, and even heavily pre-trained models such as SAM show a clear decline. Across the medical benchmarks, baselines collapse after just two sessions in *Med FoSSIL-Disjoint* (Table 2), while FoSSIL with a U-Net backbone sustains strong performance across all five, demonstrating robustness to multiple constraints as shown in Table 2 and Table 3. In *Med FoSSIL-Mixed*, transformer backbones such as MedFormer, SwinUNetr, and a CLIP-driven U-Net all degrade over sessions, with the latter dropping despite pre-training on 21 of 35 classes (Table 5), highlighting the benchmark's difficulty. In *Med Semi-Supervised-FoSSIL*, adding unlabeled data significantly boosts FoSSIL (Table 7, Figure 4b), unlike existing semi-supervised methods that fail to exploit it. **This demonstrates that leveraging readily available unlabeled data can substantially improve multi-constraint continual learning for semantic segmentation.** Figure 4b compares FoSSIL with (SS FoSSIL-U-Net) and without (FoSSIL-U-Net) access to unlabeled data. With only a few labeled samples, the added unlabeled examples broaden the coverage of the new session and lead to a clear performance gain. This improvement shows that selective pseudo-labeling helps stabilize adaptation. In this setup, MedFormer effectively adapts to the substantial domain shift introduced by BraTS (MRI) domain in Session 4 (Table 1) and still maintains a strong score of 0.323 with FoSSIL (Table 7), outperforming U-Net. In natural scene benchmark (*Natural-FoSSIL*), even large-scale models like SAM Kirillov et al. (2023), pre-trained on a billion masks, exhibit forgetting (Table 4), yet FoSSIL consistently improves SAM, as well as U-Net and transformer backbones, showing broad applicability. Finally, in *Semi-Supervised Natural-FoSSIL*, FoSSIL serves as a plug-and-play module

Table 6: Performance on Semi-Supervised Natural-FoSSIL benchmark. All values are reported as mIoU (0-100).

| Method | Session 0 | Session 1 | Session 2 | Session 3 |
|---|---|---|---|---|
| **DeepLab Vanilla** | 47.76 | 1.04 | 1.51 | 0.43 |
| **MDIL** Garg et al. (2022) | 47.76 | 1.87 | 1.43 | 0.39 |
| **MiB** Cermelli et al. (2020) | 47.76 | 5.97 | 1.59 | 0.42 |
| **UaD-CE** Cui et al. (2024) | 47.76 | 1.88 | 1.74 | 0.69 |
| **NNCSL** Kang et al. (2023b) | 47.76 | 0.79 | 1.27 | 0.46 |
| **HALO** Franco et al. (2024) | 47.76 | 1.78 | 2.02 | 1.27 |
| **RETRIEVE** Killamsetty et al. (2021) | 47.76 | 1.57 | 1.89 | 0.39 |
| **GAPS** Qiu et al. (2023) | 47.76 | 19.73 | 18.76 | 14.45 |
| **FoSSIL + GAPS** | 47.76 | **27.84** | **27.69** | **25.47** |

Table 7: Performance on Med Semi-Supervised-FoSSIL benchmark. Results reported as Dice coefficients (0-1).

| Method | Session 0 | Session 1 | Session 2 | Session 3 | Session 4 | Session 5 |
|---|---|---|---|---|---|---|
| **U-Net Vanilla** | 0.700 | 0.076 | 0.058 | 0.047 | 0.030 | 0.043 |
| **NNCSL (U-Net)** Kang et al. (2023b) | 0.700 | 0.048 | 0.048 | 0.030 | 0.011 | 0.040 |
| **UaD-CE (U-Net)** Cui et al. (2024) | 0.700 | 0.082 | 0.075 | 0.067 | 0.031 | 0.049 |
| **FoSSIL (U-Net)** | 0.736 | **0.554** | **0.445** | **0.414** | **0.058** | **0.368** |
| **MedFormer Vanilla** | 0.659 | 0.065 | 0.062 | 0.059 | 0.051 | 0.040 |
| **UaD-CE (MedFormer)** Cui et al. (2024) | 0.659 | 0.052 | 0.048 | 0.065 | 0.037 | 0.032 |
| **NNCSL (MedFormer)** Kang et al. (2023b) | 0.659 | 0.142 | 0.095 | 0.144 | 0.010 | 0.048 |
| **CSL (MedFormer)** Liu & Liu (2025) | 0.659 | 0.040 | 0.020 | 0.000 | 0.000 | 0.000 |
| **FoSSIL (MedFormer)** | 0.640 | **0.431** | **0.368** | **0.335** | **0.323** | **0.293** |

that leverages unlabeled data to enhance GAPS results (Table 6), further underscoring its effectiveness across diverse baselines. Existing continual semi-supervised methods remain unreliable in these setups. The CT to BraTS MRI shift at Session 4 (Table 3 and 7) caused near complete forgetting in the U-Net. FoSSIL raises the Dice score only from 0.025 (Table 3) to 0.0576 (Table 7), highlighting the difficulty of the FoSSIL benchmarks.

Figure 4a shows how performance evolves across epochs within an incremental session where the model encounters new classes and a new domain under few-shot supervision. It is to be noted that both Vanilla SwinUNetr and FoSSIL-SwinUNetr begin Session 1 from the same Vanilla base model trained in Session 0, ensuring identical initialization. At epoch 0, both perform poorly because the new classes were unseen in the base session. As training progresses, the Vanilla model shows minimal improvement, suggesting that gradients from few-shot supervision are noisy for stable adaptation and disturb previously learned representations. In contrast, FoSSIL adapts well to the new classes, new domain, and limited labels. Its consistently higher and more stable trajectory demonstrates controlled and reliable learning throughout the session.

**Hyperparameters:** The robustness and sensitivity analysis of GNI in Table 8 shows that FoSSIL remains robust despite relying on only a minimal set of hyperparameters. We select pseudo-labels using the confidence threshold $\tau_{conf}$ and the similarity threshold $\tau_{sim}$. Lower values (e.g., 0.6) increase retention of pseudo-labels but admit low-confidence labels, while higher values reduce retention excessively. We adopt $\tau_{conf} = 0.7$ and $\tau_{sim} = 0.7$ across all datasets. Sensitivity analysis in the Appendix E confirms robustness.

**Ablations:** In the Med FoSSIL-Mixed benchmark, where FoSSIL improves the performance of MedFormer, we removed guided noise injection ($\tilde{G}^{-1}$) from Equation 5, and the results are plotted in Figure 5a. As shown, there is a significant drop in performance, highlighting the importance of the proposed guided noise injection strategy, which helps regularize the model under few-shot data and domain shifts. In the Semi-Supervised Natural-FoSSIL benchmark, where FoSSIL improves GAPS using unlabeled data, we removed the pseudo-label refinement strategy and evaluated FoSSIL's performance, as shown in Figure 5b. It is evident that the refinement strategy contributes to the improved performance of FoSSIL. Please see Appendix F for the layer-wise impact of GNI.

**Computational overhead:** The computational analysis (Table 9) for Session 1 shows that FoSSIL adds no extra cost yet yields notable performance improvements over the Vanilla baseline (w/o FoSSIL). FoSSIL remains lightweight, uses few hyperparameters, generalizes across backbones, and performs strongly on about 12 datasets compared with around 37 baselines.

Table 8: FoSSIL sensitivity/robustness analysis across $\varepsilon$, noise variance, and number of shots $K$.

| Setting | Varying $\varepsilon$ | | | | Variance of noise added to $\mathbb{W}$ | | | Shots $(K)$ | | |
|---|---|---|---|---|---|---|---|---|---|---|
| | $10^{-6}$ | $10^{-7}$ | $10^{-8}$ | $10^{-9}$ | 0.1 | 1 | 10 | 3 | 4 | 5 |
| **FoSSIL** | 0.443 | 0.437 | 0.460 | 0.482 | 0.460 | 0.460 | 0.408 | 0.414 | 0.434 | 0.460 |
| **Vanilla** | | | | | | 0.076 | | | | |

Table 9: Computational analysis for Session 1.

| Setting | Parameters (M) | | FLOPs (T) | | Training Time | |
|---|---|---|---|---|---|---|
| | FoSSIL | w/o FoSSIL | FoSSIL | w/o FoSSIL | FoSSIL | w/o FoSSIL |
| **Med FoSSIL-Disjoint** | 16.27 | 16.27 | 0.52 | 0.52 | 4h 06m | 4h 05m |
| **Med Semi-Supervised-FoSSIL** | 39.59 | 39.59 | 1.10 | 1.10 | 5h 18m | 5h 08m |
| **Natural-FoSSIL (SAM ViT-B)** | 88.90 | 88.90 | 0.37 | 0.37 | 1h 43m | 1h 35m |

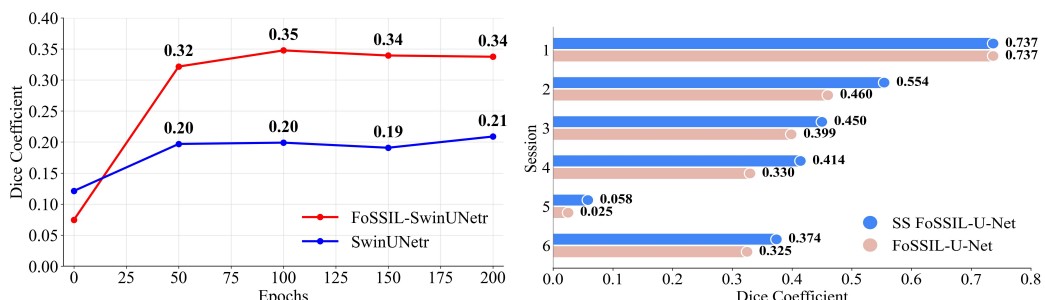

Figure 4: a) (left) Variation of performance of Vanilla SwinUnetr and with FoSSIL over the epochs. This illustrates how FoSSIL sustains performance across epochs. b) (right) Performance of FoSSIL without unlabeled data (Med FoSSIL-Disjoint) and with unlabeled data (Med Semi-Supervised-FoSSIL). 'SS' is Semi-Supervised.

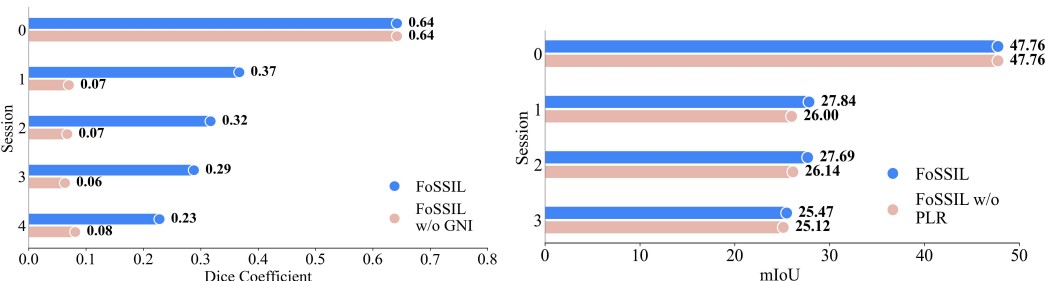

Figure 5: a) (left) FoSSIL without Guided Noise Injection (GNI) evaluated with MedFormer (Med FoSSIL-Mixed). b) (right) FoSSIL without Pseudo-Label Refinement (PLR) on Semi-Supervised Natural-FoSSIL benchmark.

## 6 CONCLUSION

We evaluated class-incremental, domain incremental, few-shot, and semi-supervised methods on the proposed benchmarks, which reveal a large performance gap that current approaches do not close. This highlights the need for more robust methods that can handle multiple constraints in continual semantic segmentation, as even large pre-trained and foundation models show clear degradation. Our framework, FoSSIL, reduces performance drop across sessions and demonstrates the benefit of guided noise injection and pseudo-label refinement. It also shows that readily available unlabeled data can substantially improve continual learning for semantic segmentation. In future work, FoSSIL will be extended to more demanding settings such as open vocabulary learning, detection, and other dense prediction tasks. We will also evaluate and refine additional foundation and vision language models on the FoSSIL benchmark.

## 7 LLM Usage

We used LLMs solely for grammar refinement, and table formatting.

## 8 Reproducibility

All our codes, benchmarks and implementations are available at https://github.com/anony34/FoSSIL.

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

APPENDIX CONTENTS

# A  THEORETICAL ANALYSIS OF GUIDED NOISE INJECTION (GNI)

## A.1  DEFINITIONS, NOTATION, AND ASSUMPTIONS

**Definition A.1** (Weight vector). *Let $\mathbb{W} \in \mathbb{R}^p$ denote the flattened vector containing all model parameters, where each component is indexed by $i \in \{1, \ldots, p\}$, and $w_i$ denotes the $i$-th parameter.*

**Definition A.2** (Gradient buffer). *At the current optimization step,*

$$g_i = \frac{\partial \mathcal{L}}{\partial w_i},$$

*denote the gradient of the loss $\mathcal{L}$ w.r.t. $w_i$, and define the elementwise squared gradient:*

$$G_i = g_i^2.$$

**Definition A.3** (Inverse gradient modulator (normalized)). *Introduce a stable inverse-gradient modulator:*

$$G_{\mathrm{inv},i} = \frac{1}{G_i + \varepsilon}, \quad \varepsilon > 0,$$

*and normalize to a bounded range to control noise magnitude:*

$$\tilde{G}_i = \frac{1 + G_{\mathrm{inv},i} - \min_j G_{\mathrm{inv},j}}{1 + \max_j G_{\mathrm{inv},j} - \min_j G_{\mathrm{inv},j}} \in (0, 1]. \tag{11}$$

**Definition A.4** (Guided noise operator). *Let $\xi = (\xi_1, \ldots, \xi_p)$ be i.i.d. standard Gaussian noise, $\xi_i \sim \mathcal{N}(0, 1)$. Define,*

$$\mathcal{U}(\mathbb{W}) = \mathbb{W} + \tilde{G} \odot \xi,$$

*i.e., componentwise:*

$$\widetilde{w}_i = w_i + \tilde{G}_i \, \xi_i.$$

**Assumption A.1** (Smoothness). *$\mathcal{L} : \mathbb{R}^p \to \mathbb{R}$ is twice continuously differentiable in a neighborhood of $\mathbb{W}$. Let $g = (g_1, \ldots, g_p)$ be the gradient vector and $H = \nabla^2 \mathcal{L}(\mathbb{W})$ the Hessian.*

## A.2  EXPECTED PERTURBED LOSS

Consider the Taylor expansion of $\mathcal{L}$ at $\mathbb{W}$ with increment $\Delta \mathbb{W} = \tilde{G} \odot \xi$:

$$\mathcal{L}(\mathbb{W} + \Delta \mathbb{W}) = \mathcal{L}(\mathbb{W}) + g^\top \Delta \mathbb{W} + \frac{1}{2} \Delta \mathbb{W}^\top H \Delta \mathbb{W} + R_3(\Delta \mathbb{W}),$$

where $R_3(\Delta \mathbb{W}) = O(\|\Delta \mathbb{W}\|^3)$.

**Linear term expectation:**

$$\mathbb{E}_\xi[g^\top \Delta \mathbb{W}] = \sum_i g_i \tilde{G}_i \mathbb{E}[\xi_i] = 0,$$

since $\mathbb{E}[\xi_i] = 0$.

**Quadratic term expectation (diagonal approximation):**

$$\frac{1}{2} \mathbb{E}_\xi[\Delta \mathbb{W}^\top H \Delta \mathbb{W}] = \frac{1}{2} \sum_{i,j} H_{ij} \tilde{G}_i \tilde{G}_j \mathbb{E}[\xi_i \xi_j] \approx \frac{1}{2} \sum_{i=1}^{p} H_{ii} \tilde{G}_i^2,$$

because $\mathbb{E}[\xi_i \xi_j] = 0$ for $i \neq j$ and $\mathbb{E}[\xi_i^2] = 1$.

**Higher order term:**

$$|R_3(\Delta \mathbb{W})| \leq C \|\Delta \mathbb{W}\|^3, \quad \mathbb{E}[|R_3|] = O(\mathbb{E}\|\Delta \mathbb{W}\|^3) = O(\max_i \tilde{G}_i^3).$$

for some constant $C$. Since $\tilde{G}_i \in (0, 1]$, the higher order term is bounded even for small $g_i^2$.

*Note:* For additional control of noise magnitude during training, one can scale the modulators by a factor $\alpha \in (0, 1]$:

$$\tilde{G}_i \; \longrightarrow \; \alpha\,\tilde{G}_i.$$

With $\Delta\mathbb{W} = \alpha\tilde{G} \odot \xi$ and $\xi \sim \mathcal{N}(0, I_p)$ this yields,

$$|R_3(\alpha\tilde{G} \odot \xi)| \leq C(\alpha \max_i \tilde{G}_i)^3 \|\xi\|^3,$$

and thus, taking expectation,

$$\mathbb{E}\big[|R_3(\alpha\tilde{G} \odot \xi)|\big] \leq C(\alpha \max_i \tilde{G}_i)^3 \mathbb{E}\|\xi\|^3 = O\big((\alpha \max_i \tilde{G}_i)^3\big).$$

The factor $\mathbb{E}\|\xi\|^3$ is finite (depends only on model parameters $p$). This ensures that the higher-order contributions remain bounded and controllable, while providing finer control of noise injection. The above expectation is derived under the assumption that the parameter dimensionality $p$ remains fixed within an incremental session.

Thus, expected perturbed loss:

$$\mathbb{E}_\xi[\mathcal{L}(\mathcal{U}(\mathbb{W}))] = \mathcal{L}(\mathbb{W}) + \frac{1}{2}\sum_{i=1}^{p} H_{ii}\tilde{G}_i^2 + O(\max_i \tilde{G}_i^3)$$

### A.2.1 INTERPRETATION AND THEORETICAL MOTIVATION

- $\tilde{G}_i$ is large for low-gradient coordinates ($g_i^2$ small), injecting more noise along flat directions (directions of low curvature) of the loss landscape.
- $\tilde{G}_i$ is small for high-gradient coordinates, suppressing perturbation along sensitive directions.
- The bounded normalization ensures that all $\tilde{G}_i \leq 1$, bounding the perturbation magnitude even when gradients are arbitrarily small, thus controlling the higher-order terms.
- Hence, GNI selectively perturbs parameters along directions of low sensitivity, promoting exploration along flat regions without significantly increasing expected loss.
- This justifies theoretically why GNI can encourage flatter minima, better generalization, and robust optimization behavior.

### A.3 SPECTRAL VIEWPOINT: HESSIAN EIGEN-STRUCTURE

Let $H = U\Lambda U^\top$ be the eigendecomposition of the Hessian, where $U$ is orthonormal and $\Lambda = \text{diag}(\lambda_1, \ldots, \lambda_p)$. Then each diagonal entry satisfies,

$$H_{ii} = \sum_{r=1}^{p} \lambda_r u_{ir}^2.$$

The expected second-order contribution with $\tilde{G}$ becomes,

$$\sum_{i=1}^{p} H_{ii}\tilde{G}_i^2 = \sum_{i=1}^{p}\sum_{r=1}^{p} \lambda_r u_{ir}^2 \tilde{G}_i^2 = \sum_{r=1}^{p} \lambda_r \sum_{i=1}^{p} (u_{ir}\tilde{G}_i)^2.$$

Define $v^{(r)} \in \mathbb{R}^p$ with components $v_i^{(r)} = u_{ir}\tilde{G}_i$. Then,

$$\sum_{i=1}^{p} H_{ii}\tilde{G}_i^2 = \sum_{r=1}^{p} \lambda_r \|v^{(r)}\|_2^2 = \sum_{r=1}^{p} \lambda_r \|U_{:,r} \odot \tilde{G}\|_2^2$$

### A.3.1 INTERPRETATION

GNI implements *anisotropic noise injection*: each Hessian eigenvalue $\lambda_r$ is weighted by $\|u^{(r)} \odot \tilde{G}\|_2^2$, the squared $\ell_2$-norm of $\tilde{G}$ along the $r$-th eigendirection. When $\tilde{G}$ is large on coordinates aligned with low-curvature directions (small $\lambda_r$), the noise contribution $\lambda_r\|u^{(r)} \odot \tilde{G}\|_2^2$ remains controlled by the small eigenvalue; conversely, high-curvature directions (large $\lambda_r$) are automatically protected because $\tilde{G}$ is small there. This concentrates noise along flat directions with minimal loss impact while protecting sensitive directions, encouraging flatter minima that improve generalization, with higher-order terms remaining bounded even when gradients vanish.

## B PAC-BAYES ANALYSIS OF GUIDED NOISE INJECTION

### B.1 SETUP AND ASSUMPTIONS

We interpret the guided noise injection (GNI),

$$\widetilde{\mathbb{W}} = \mathbb{W} + \tilde{G} \odot \xi, \qquad \xi \sim \mathcal{N}(0, I_p), \tag{12}$$

as defining a stochastic posterior over parameters, where $\mathbb{W} \in \mathbb{R}^p$ denotes the current model weights. Each coordinate $w_i$ is perturbed by scaled Gaussian noise $\tilde{G}_i \xi_i$, with normalized noise magnitude,

$$\tilde{G}_i = \frac{1 + G_{\text{inv},i} - \min_j G_{\text{inv},j}}{1 + \max_j G_{\text{inv},j} - \min_j G_{\text{inv},j}} \in (0, 1], \qquad G_{\text{inv},i} = \frac{1}{(\nabla_{w_i} \mathcal{L})^2 + \varepsilon} \tag{13}$$

Assumptions:

- The coordinates of $\widetilde{\mathbb{W}}$ are independent, yielding a diagonal posterior covariance.
- Larger $\tilde{G}_i$ corresponds to low-sensitivity directions (smaller gradient magnitude).
- The prior is isotropic Gaussian: $p(\theta) = \mathcal{N}(0, \tau^2 I_p)$, and the posterior induced by GNI is $q(\theta) = \mathcal{N}(\mathbb{W}, \Sigma_q)$ with diagonal covariance $\Sigma_q = \text{diag}(\tilde{G}_1^2, \ldots, \tilde{G}_p^2)$.

### B.2 KL DIVERGENCE BETWEEN POSTERIOR AND PRIOR

For general multivariate Gaussians $q = \mathcal{N}(\mu_q, \Sigma_q)$ and $p = \mathcal{N}(\mu_p, \Sigma_p)$, the KL divergence is:

$$\text{KL}(q\|p) = \frac{1}{2}\left[\text{tr}(\Sigma_p^{-1}\Sigma_q) + (\mu_p - \mu_q)^\top \Sigma_p^{-1}(\mu_p - \mu_q) - p + \ln\frac{\det\Sigma_p}{\det\Sigma_q}\right].$$

For the GNI posterior $q(\theta) = \mathcal{N}(\mathbb{W}, \text{diag}(\tilde{G}_1^2, \ldots, \tilde{G}_p^2))$ and isotropic prior $p(\theta) = \mathcal{N}(0, \tau^2 I_p)$, we have:

- Posterior (GNI): $\mu_q = \mathbb{W} = (w_1, \ldots, w_p)$, $\quad \Sigma_q = \text{diag}(\tilde{G}_1^2, \ldots, \tilde{G}_p^2)$,
- Prior: $\mu_p = 0$, $\quad \Sigma_p = \tau^2 I_p$ (isotropic).

**Trace term:** $\text{tr}(\Sigma_p^{-1}\Sigma_q)$ Since $\Sigma_p = \tau^2 I_p$ and $\Sigma_q = \text{diag}(\tilde{G}_1^2, \ldots, \tilde{G}_p^2)$:

$$\Sigma_p^{-1}\Sigma_q = (\tau^2 I_p)^{-1}\text{diag}(\tilde{G}_1^2, \ldots, \tilde{G}_p^2)$$
$$= \frac{1}{\tau^2}\text{diag}(\tilde{G}_1^2, \ldots, \tilde{G}_p^2).$$

The trace of a diagonal matrix is the sum of its diagonal elements, so

$$\text{tr}(\Sigma_p^{-1}\Sigma_q) = \sum_{i=1}^{p} \frac{\tilde{G}_i^2}{\tau^2}.$$

**Mean difference term:** $(\mu_p - \mu_q)^\top \Sigma_p^{-1}(\mu_p - \mu_q)$

$$(\mu_p - \mu_q)^\top \Sigma_p^{-1}(\mu_p - \mu_q) = (0 - \mathbb{W})^\top(\tau^{-2}I_p)(0 - \mathbb{W})$$
$$= \mathbb{W}^\top(\frac{1}{\tau^2}I_p)\mathbb{W}$$
$$= \frac{1}{\tau^2}\sum_{i=1}^{p} w_i^2.$$

This term measures how far the posterior mean $\mathbb{W}$ is from the prior mean 0, scaled by the prior variance.

**Log-determinant term:** $\ln \frac{\det \Sigma_p}{\det \Sigma_q}$   The determinant of a matrix is the product of its eigenvalues. For a diagonal matrix, the eigenvalues are the diagonal entries:

$$\det(\mathrm{diag}(d_1, \ldots, d_p)) = \prod_{i=1}^{p} d_i.$$

Hence,

$$\det \Sigma_p = \det(\tau^2 I_p) = (\tau^2)^p,$$

$$\det \Sigma_q = \det(\mathrm{diag}(\tilde{G}_1^2, \ldots, \tilde{G}_p^2)) = \prod_{i=1}^{p} \tilde{G}_i^2.$$

Then the log-determinant term becomes,

$$\ln \frac{\det \Sigma_p}{\det \Sigma_q} = \ln \frac{\tau^{2p}}{\prod_{i=1}^{p} \tilde{G}_i^2} = \sum_{i=1}^{p} \ln \frac{\tau^2}{\tilde{G}_i^2}.$$

This term penalizes how much the posterior *volume* differs from the prior. If posterior variance is smaller than prior, the log term is positive.

- $\frac{\tilde{G}_i^2}{\tau^2} \to$ contribution from posterior variance

- $\frac{w_i^2}{\tau^2} \to$ contribution from posterior mean

- $-\ln(\tilde{G}_i^2/\tau^2) \to$ penalizes shrinkage or expansion of posterior volume

Combining these, the KL divergence reduces to:

$$\mathrm{KL}(q\|p) = \frac{1}{2} \sum_{i=1}^{p} \Big[ \frac{\tilde{G}_i^2}{\tau^2} + \frac{w_i^2}{\tau^2} - 1 - \ln \frac{\tilde{G}_i^2}{\tau^2} \Big]. \tag{14}$$

### B.3   INTERPRETATION OF THE KL DIVERGENCE TERM

Each parameter $w_i$ contributes to the KL divergence:

$$\mathrm{KL}_i = \frac{1}{2} \Big[ \frac{\tilde{G}_i^2}{\tau^2} + \frac{w_i^2}{\tau^2} - 1 - \ln \frac{\tilde{G}_i^2}{\tau^2} \Big].$$

Define,

$$x_i = \frac{\tilde{G}_i^2}{\tau^2}, \qquad f(x) = \frac{1}{2}(x - \ln x - 1),$$

so that,

$$\mathrm{KL}_i = f(x_i) + \frac{w_i^2}{2\tau^2}. \tag{15}$$

**Low-gradient parameters:**   Parameters with small gradients have large $\tilde{G}_i$, injecting more noise. This increases posterior variance in low-sensitivity directions, **promoting exploration and regularization (plasticity)**, allowing these parameters to safely explore flat regions without significantly affecting expected loss. The KL contribution is influenced both by the noise scale $\tilde{G}_i^2/\tau^2$ and the weight magnitude $w_i^2/\tau^2$. Since $\tilde{G}_i \in (0, 1]$, choosing $\tau \in [\tilde{G}_{\min}, \tilde{G}_{\max}]$ ensures $x_i$ remains bounded, preventing the KL from growing excessively.

**High-gradient parameters:** Parameters with large gradients have small $\tilde{G}_i$, suppressing noise. The KL term is then dominated by $w_i^2/\tau^2$, reflecting that the posterior is tightly concentrated around the current weight, which **prevents drift of these important weight parameters (stability)**. The normalization in GNI ensures $\tilde{G}_i > 0$, preventing the log term from diverging. Intuitively, high-gradient parameters remain stable.

Thus, the KL contribution reflects both the posterior spread ($\tilde{G}_i$) and the weight magnitude ($w_i$), providing a principled trade-off between stochastic regularization and posterior concentration. This provides a rigorous, *anisotropic* regularization mechanism that adapts to the local geometry of the loss landscape.

**Theorem B.1** (PAC-Bayes Bound Improvement via GNI). *Let $p(\theta)$ denote a prior over model parameters, and let $q_{\mathrm{ani}(\theta)}$ and $q_{\mathrm{iso}(\theta)}$ denote the anisotropic and isotropic posteriors, respectively, both having the same total variance. Then, the anisotropic posterior induced by Guided Noise Injection (GNI) achieves a tighter PAC-Bayes bound than the isotropic posterior:*

$$\mathcal{L}_{\mathrm{PAC}}(q_{\mathrm{ani}(\theta)}, p(\theta)) \leq \mathcal{L}_{\mathrm{PAC}}(q_{\mathrm{iso}(\theta)}, p(\theta))$$

*Proof.* We define two posterior distributions over parameters $\theta$:

- **Anisotropic posterior (GNI)**:

$$q_{\mathrm{ani}}(\theta) = \mathcal{N}(\mathbb{W}, \Sigma_{\mathrm{ani}}), \quad \Sigma_{\mathrm{ani}} = \mathrm{diag}(\tilde{G}_1^2, \ldots, \tilde{G}_p^2),$$

  where $\tilde{G}_i \in (0, 1]$ is an adaptive variance, decreasing with gradient magnitude $|\nabla_{w_i}\mathcal{L}|$:

$$|\nabla_{w_i}\mathcal{L}| \text{ small} \Rightarrow \tilde{G}_i \text{ large}, \quad |\nabla_{w_i}\mathcal{L}| \text{ large} \Rightarrow \tilde{G}_i \text{ small}.$$

- **Isotropic posterior**:

$$q_{\mathrm{iso}}(\theta) = \mathcal{N}(\mathbb{W}, \sigma^2 I_p),$$

  with $\sigma^2 = \frac{1}{p}\sum_{i=1}^{p} \tilde{G}_i^2$ chosen so that the total variance matches that of the anisotropic posterior:

$$\mathrm{tr}(\Sigma_{\mathrm{ani}}) = \mathrm{tr}(\sigma^2 I_p) = p\sigma^2.$$

The prior is isotropic Gaussian:

$$p(\theta) = \mathcal{N}(0, \tau^2 I_p), \quad \tau > 0.$$

Using Equation 14, for the isotropic posterior with variance $\sigma^2$:

$$\mathrm{KL}(q_{\mathrm{iso}}\|p) = \frac{1}{2}\sum_{i=1}^{p}\left[\frac{\sigma^2}{\tau^2} + \frac{w_i^2}{\tau^2} - 1 - \ln\frac{\sigma^2}{\tau^2}\right] = \frac{1}{2}\left[\frac{p\sigma^2}{\tau^2} + \frac{\|\mathbb{W}\|_2^2}{\tau^2} - p - p\ln\frac{\sigma^2}{\tau^2}\right]. \quad (16)$$

Similar to Equation 15,

$$\mathrm{KL}_i^{\mathrm{iso}} = f(y_i) + \frac{w_i^2}{2\tau^2}. \quad (17)$$

where,

$$y_i = \frac{\sigma^2}{\tau^2} \quad \forall i.$$

Using a second-order Taylor expansion around $\mathbb{W}$:

$$\mathcal{L}(\theta) \approx \mathcal{L}(\mathbb{W}) + (\theta - \mathbb{W})^\top \nabla\mathcal{L}(\mathbb{W}) + \frac{1}{2}(\theta - \mathbb{W})^\top H(\theta - \mathbb{W}),$$

where,

$$H = \nabla_{\mathbb{W}}^2\mathcal{L}(\mathbb{W}) = U\Lambda U^\top, \quad \Lambda = \mathrm{diag}(\lambda_1, \ldots, \lambda_p), \quad U^\top U = I_p.$$

Taking expectation under $q(\theta)$ and using $\mathbb{E}_q[\theta - \mathbb{W}] = 0$:

$$\mathbb{E}_{\theta \sim q}[\mathcal{L}(\theta)] \approx \mathcal{L}(\mathbb{W}) + \frac{1}{2}\mathrm{tr}(H\Sigma_q). \tag{18}$$

For anisotropic posterior:

$$\mathbb{E}_{q_{\mathrm{ani}}}[\mathcal{L}(\theta)] - \mathcal{L}(\mathbb{W}) \approx \frac{1}{2}\mathrm{tr}(H\Sigma_{\mathrm{ani}}) = \frac{1}{2}\sum_{i=1}^{p} H_{ii}\tilde{G}_i^2,$$

$$\sum_{i=1}^{p} H_{ii}\tilde{G}_i^2 = \sum_{i=1}^{p}\sum_{r=1}^{p} \lambda_r u_{ir}^2 \tilde{G}_i^2 = \sum_{r=1}^{p} \lambda_r \sum_{i=1}^{p}(u_{ir}\tilde{G}_i)^2 = \sum_{r=1}^{p} \lambda_r \|U_{:,r} \odot \tilde{G}\|_2^2$$

For isotropic posterior:

$$\mathbb{E}_{q_{\mathrm{iso}}}[\mathcal{L}(\theta)] - \mathcal{L}(\mathbb{W}) \approx \frac{1}{2}\mathrm{tr}(H\Sigma_{\mathrm{iso}}) = \frac{1}{2}\sum_{i=1}^{p} H_{ii}\sigma^2 = \frac{1}{2}\sigma^2\mathrm{tr}(H),$$

$$\sigma^2\mathrm{tr}(H) = \sigma^2 \sum_{i=1}^{p} H_{ii} = \sigma^2 \sum_{r=1}^{p} \lambda_r.$$

Low-curvature directions ($\lambda_r$ small) are allowed larger variance $\tilde{G}_i \in (0, 1]$, contributing minimally to $\sum_{r=1}^{p} \lambda_r \|U_{:,r} \odot \tilde{G}\|_2^2$. High-curvature directions ($\lambda_r$ large) have small $\tilde{G}_i$, preventing large expected loss increase. Isotropic variance $\sigma^2$ does not adapt to curvature; high-curvature directions may receive excessive noise.

Hence, we obtain the inequality

$$\mathbb{E}_{\theta \sim q_{\mathrm{ani}}}[\mathcal{L}(\theta)] = \mathcal{L}(\mathbb{W}) + \frac{1}{2}\mathrm{tr}(H\Sigma_{\mathrm{ani}}) \leq \mathcal{L}(\mathbb{W}) + \frac{1}{2}\mathrm{tr}(H\Sigma_{\mathrm{iso}}) = \mathbb{E}_{\theta \sim q_{\mathrm{iso}}}[\mathcal{L}(\theta)]. \tag{19}$$

The KL divergence term satisfies convexity property of $f(x) = x - \ln x - 1$:

$$\frac{1}{p}\sum_{i=1}^{p} f\left(\frac{\tilde{G}_i^2}{\tau^2}\right) \leq f\left(\frac{1}{p}\sum_{i=1}^{p}\frac{\tilde{G}_i^2}{\tau^2}\right) = f\left(\frac{\sigma^2}{\tau^2}\right),$$

by Jensen's inequality. Hence,

$$\sum_{i=1}^{p} f\left(\frac{\tilde{G}_i^2}{\tau^2}\right) \leq pf\left(\frac{\sigma^2}{\tau^2}\right). \tag{20}$$

The mean term $\sum_i w_i^2/(2\tau^2)$ is identical for both posteriors (weight normalization ensures boundedness). Therefore, from Equation 15 and 17:

$$\mathrm{KL}(q_{\mathrm{ani}}\|p) \leq \mathrm{KL}(q_{\mathrm{iso}}\|p). \tag{21}$$

For dataset size $N$ and confidence level $\delta \in (0, 1)$, with probability at least $1 - \delta$, the PAC-Bayes bound[1] states:

$$\mathbb{E}_{\theta \sim q}[\mathcal{L}_{\mathrm{test}}(\theta)] \leq \mathbb{E}_{\theta \sim q}[\widehat{\mathcal{L}}(\theta)] + \sqrt{\frac{\mathrm{KL}(q\|p) + \ln\frac{N}{\delta}}{2(N-1)}}. \tag{22}$$

Combining (19) and (21):

[1]David A. McAllester. *PAC-Bayesian model averaging*. In *Proceedings of the Twelfth Annual Conference on Computational Learning Theory*, pp. 164–170, 1999. Available at: https://dl.acm.org/doi/10.1145/307400.307435.

$$\mathbb{E}_{q_{\mathrm{ani}}}[\mathcal{L}_{\mathrm{test}}] \leq \mathbb{E}_{q_{\mathrm{ani}}}[\widehat{\mathcal{L}}(\theta)] + \sqrt{\frac{\mathrm{KL}(q_{\mathrm{ani}}\|p) + \ln(N/\delta)}{2(N-1)}} \leq \mathbb{E}_{q_{\mathrm{iso}}}[\widehat{\mathcal{L}}(\theta)] + \sqrt{\frac{\mathrm{KL}(q_{\mathrm{iso}}\|p) + \ln(N/\delta)}{2(N-1)}},$$

$$\implies \mathcal{L}_{\mathrm{PAC}}(q_{\mathrm{ani}(\theta)}, p(\theta)) \leq \mathcal{L}_{\mathrm{PAC}}(q_{\mathrm{iso}(\theta)}, p(\theta)).$$

*Thus, the anisotropic posterior induced by GNI achieves a tighter PAC-Bayes bound than the isotropic posterior with matched total variance.* □

## C   THEORETICAL ANALYSIS OF PROTOTYPE-GUIDED PSEUDO-LABEL REFINEMENT (PRL)

This section analyzes the role of pseudo-labels in improving semi-supervised mean-teacher models with unlabeled data and explains how prototype-guided pseudo-label refinement may provide additional gains.

### C.1   SETUP

Consider a student-teacher semi-supervised segmentation framework:

- Labeled dataset: $\mathcal{D}_l$,
- Unlabeled dataset: $\mathcal{D}_u$,
- Student network: $M_s$, Teacher network: $M_t$ (EMA of $M_s$),
- Pseudo-label weight: $\gamma \in (0,1)$,
- Exponential Moving Average (EMA decay: $\alpha \in [0,1)$,

Let $\epsilon_0$ denote the base error of the student on labeled data. For unlabeled data, let $\epsilon_s(t)$ and $\epsilon_t(t)$ denote the fraction of incorrect pseudo-labels for the student and teacher at iteration $t$.

### C.2   PSEUDO-LABEL ERROR DYNAMICS WITHOUT PSEUDO-LABEL REFINEMENT

We begin by formulating the coupled dynamics between the student and teacher networks in the absence of pseudo-label refinement (PRL). Let $\epsilon_s(t)$ and $\epsilon_t(t)$ denote the expected pseudo-label errors (i.e., the fractions of incorrect pseudo-labels) produced by the student and teacher, respectively, at iteration $t$. The student updates its parameters using a mixture of ground-truth labeled data and teacher-generated pseudo-labels:

$$\epsilon_s(t+1) = (1-\gamma)\,\epsilon_0 + \gamma\,\epsilon_t(t), \tag{23}$$

where $\epsilon_0$ represents the intrinsic baseline error arising from labeled supervision, and $\gamma \in [0,1]$ controls the relative contribution of pseudo-labeled data in training. Intuitively, $\gamma = 0$ corresponds to purely supervised learning based solely on labeled data, whereas $\gamma \to 1$ denotes a regime where training is dominated by pseudo-labels generated by the teacher.

The teacher, on the other hand, is an exponentially moving average (EMA) of the student:

$$\epsilon_t(t+1) = \alpha\epsilon_t(t) + (1-\alpha)\epsilon_s(t+1), \tag{24}$$

where $\alpha \in [0,1)$ is the EMA decay constant. This coupling forms a delayed feedback loop: the teacher smooths over the temporal trajectory of student errors.

Substituting Equation 23 into Equation 24 gives:

$$\epsilon_t(t+1) = \alpha\epsilon_t(t) + (1-\alpha)\big[(1-\gamma)\epsilon_0 + \gamma\epsilon_t(t)\big]$$
$$= [\alpha + (1-\alpha)\gamma]\epsilon_t(t) + (1-\alpha)(1-\gamma)\epsilon_0.$$

This establishes a *non-homogeneous linear recurrence relation* describing how teacher error evolves in conjunction with student learning dynamics.

Let us define:

$$\lambda = \alpha + (1-\alpha)\gamma,$$

The update then simplifies to:

$$\epsilon_t(t+1) = \lambda\epsilon_t(t) + (1-\alpha)(1-\gamma)\epsilon_0$$

The general solution is:

$$\epsilon_t(t) = \lambda^t\epsilon_t(0) + (1-\alpha)(1-\gamma)\epsilon_0 \sum_{k=0}^{t-1} \lambda^k,$$

where $\epsilon_t(0)$ denotes the initial teacher pseudo-label error at iteration $t = 0$.

The term $\lambda^t\epsilon_t(0)$ captures the contribution of past errors, exponentially weighted over time, while the summation term captures the accumulation of newly introduced errors from labeled supervision.

We analyze three characteristic regimes of $\lambda$:

**(a)** $\lambda < 1$**:** The geometric series converges:

$$\sum_{k=0}^{t-1} \lambda^k = \frac{1-\lambda^t}{1-\lambda} \xrightarrow{t\to\infty} \frac{1}{1-\lambda}.$$

Therefore, the asymptotic value of the teacher error is:

$$\epsilon_\infty = \frac{(1-\alpha)(1-\gamma)\epsilon_0}{1-\lambda}. \tag{25}$$

Substituting $\lambda = \alpha + (1-\alpha)\gamma$, we find:

$$\epsilon_\infty = \frac{(1-\alpha)(1-\gamma)\epsilon_0}{(1-\alpha)(1-\gamma)} = \epsilon_0.$$

The teacher asymptotically retains the same error as if trained purely on ground-truth data.

**(b)** $\lambda = 1$**:** The summation becomes divergent and grows linearly with time:

$$\sum_{k=0}^{t-1} \lambda^k = t.$$

The error therefore exhibits *linear drift*:

$$\epsilon_t(t) = \epsilon_t(0) + (1-\alpha)(1-\gamma)\epsilon_0 \cdot t.$$

Errors are neither damped nor amplified exponentially, but drift slowly.

**(c)** $\lambda > 1$**:** The geometric series diverges exponentially:

$$\epsilon_t(t) = \lambda^t\epsilon_t(0) + (1-\alpha)(1-\gamma)\epsilon_0 \frac{\lambda^t - 1}{\lambda - 1}.$$

Here, $\epsilon_t$ grows exponentially leading to *confirmation bias*. The erroneous pseudo-labels reinforce the teacher, which in turn produces even more erroneous labels. Such behavior is typical when $\gamma$ is high (strong reliance on unlabeled data).

For bounded error propagation, we require $\lambda < 1$, which simplifies to $\gamma < 1$. However, in real semi-supervised settings, $|\mathcal{D}_u| \gg |\mathcal{D}_l|$, leading to an effective $\gamma$ close to 1. This makes pseudo-label drift and error amplification inevitable without refinement mechanisms. This theoretical observation provides a direct justification for *pseudo-label filtering and refinement* strategies to maintain $\lambda < 1$ in practice.

## C.3 PROTOTYPE-GUIDED PSEUDO-LABEL REFINEMENT

Let $z = f_\theta(x) \in \mathbb{R}^d$ denote the feature embedding of an unlabeled sample $x$, where $f_\theta$ is the student model with parameters $\theta$. Let $\mu_c \in \mathbb{R}^d$ be the prototype (mean feature) for class $c$, computed from the labeled set $\mathcal{D}_l^{(c)}$ of $N_c$ examples of class $c$:

$$\mu_c = \frac{1}{N_c} \sum_{x_i \in \mathcal{D}_l^{(c)}} f_\theta(x_i),$$

where $\mathcal{D}_l^{(c)}$ is the set of labeled samples belonging to class $c$, $N_c = |\mathcal{D}_l^{(c)}|$ is the number of labeled samples in class $c$. A pseudo-label $\hat{y}$ for input $x$ is accepted if both the model confidence and similarity satisfy thresholds:

$$p_T(\hat{y} \mid x) \geq \tau_{\text{conf}}, \qquad s(x, \hat{y}) = \frac{\langle f_\theta(x), \mu_{\hat{y}} \rangle}{\|f_\theta(x)\|\|\mu_{\hat{y}}\|} \geq \tau_{\text{sim}},$$

where $p_T(\hat{y} \mid x)$ is the predicted probability of class $\hat{y}$ from the teacher model $T$, $\mu_{\hat{y}} \in \mathbb{R}^d$ is the prototype (mean feature) for class $\hat{y}$, $\tau_{\text{conf}}$ and $\tau_{\text{sim}}$ are empirically determined thresholds for confidence and similarity, respectively.

This induces two measurable statistics over $\mathcal{D}_u$:

- **Coverage** $f \in (0, 1]$: fraction of pseudo-labels passing the filter,
$$f = \frac{1}{|\mathcal{D}_u|} \sum_{x \in \mathcal{D}_u} \mathbb{I}\big[p_T(\hat{y} \mid x) \geq \tau_{\text{conf}}, s(x, \hat{y}) \geq \tau_{\text{sim}}\big]$$

- **Precision** $\rho \in [0, 1]$: conditional probability that an accepted pseudo-label is correct,
$$\rho = \Pr[y = \hat{y} \mid \mathbb{I} = 1]$$
where $y$ is the ground-truth label.

*Note:* Since the true labels $y$ are not available for unlabeled data, $\rho$ serves as a conceptual measure of the expected correctness of accepted pseudo-labels.

Accounting for filtering, the effective contribution of pseudo-labeled samples is:
$$\gamma_{\text{eff}} = f\gamma,$$
where $f$ is the fraction of pseudo-labels that pass the acceptance criteria.

### C.3.1 PSEUDO-LABEL ERROR DYNAMICS UNDER PROTOTYPE REFINEMENT

The student update, accounting for filtering with precision $\rho$, can be expressed as:
$$\epsilon_s(t + 1) = (1 - f\gamma)\epsilon_0 + f\gamma(1 - \rho)\epsilon_t(t).$$
Here $(1 - \rho)$ captures the proportion of incorrect pseudo-labels that survive prototype filtering. When $\rho = 1$, all pseudo-labels are correct and the student fully benefits from unlabeled supervision; when $\rho = 0$, the model ignores pseudo-labels, reducing to a purely supervised learner.

The teacher parameters are updated through an exponential moving average:
$$\begin{aligned}
\epsilon_t(t + 1) &= \alpha\epsilon_t(t) + (1 - \alpha)\epsilon_s(t + 1) \\
&= \big[\alpha + (1 - \alpha)f\gamma(1 - \rho)\big]\epsilon_t(t) + (1 - \alpha)(1 - f\gamma)\epsilon_0,
\end{aligned}$$
where $\alpha$ is the decay rate of the EMA. Defining the effective recurrence coefficient:
$$\lambda_{\text{eff}} = \alpha + (1 - \alpha)f\gamma(1 - \rho),$$

For bounded error propagation (stability), $\lambda_{\text{eff}} < 1$, which yields:
$$f\gamma(1 - \rho) < 1, \tag{26}$$
$$\Rightarrow \quad \rho > 1 - \frac{1}{f\gamma}. \tag{27}$$

Equation 27 highlights that when pseudo-labels are highly reliable (high precision $\rho$), the student can safely use a larger fraction of pseudo-labeled data ($f\gamma$) without destabilizing training. Conversely, if pseudo-labels are less reliable (low $\rho$), only a smaller fraction of pseudo-labeled data should be used to prevent error amplification.

Provided that $\lambda_{\text{eff}} < 1$, Equation 25 shows that the teacher error asymptotically converges to:
$$\begin{aligned}
\epsilon_\infty &= \frac{(1 - \alpha)(1 - f\gamma)\epsilon_0}{1 - \lambda_{\text{eff}}} \\
&= \frac{(1 - f\gamma)\epsilon_0}{1 - f\gamma(1 - \rho)}. \tag{28}
\end{aligned}$$

Equation 28 quantifies how PLR influences long-term model performance.

**Theorem C.1** (Prototype-Guided Pseudo-Label Refinement). *Let $\epsilon_0 > 0$ denote the baseline teacher error on labeled data. In a student-teacher semi-supervised framework with pseudo-label weight $\gamma \in (0,1)$, EMA decay $\alpha \in [0,1)$, and prototype-guided refinement with coverage $f \in (0,1]$ and precision $\rho \in [0,1]$, the asymptotic teacher error with PRL is*

$$\epsilon_\infty^{\mathrm{PRL}} = \frac{(1 - f\gamma)\,\epsilon_0}{1 - f\gamma(1 - \rho)}.$$

*Under the stability condition $\lambda_{\mathrm{eff}} := \alpha + (1 - \alpha)f\gamma(1 - \rho) < 1$, the following hold:*

1. *(**Reduced asymptotic error**) $\epsilon_\infty^{\mathrm{PRL}} \leq \epsilon_0$, with strict inequality if $f\gamma > 0$ and $\rho > 0$.*
2. *(**Monotonicity in precision**) $\frac{\partial \epsilon_\infty^{\mathrm{PRL}}}{\partial \rho} < 0$, so higher precision strictly reduces asymptotic error.*
3. *(**Safe pseudo-label budget**) Stable error propagation is maintained for all $f\gamma < \frac{1}{1-\rho}$, allowing larger pseudo-label usage at higher precision.*

*Proof.* From the analysis in Section 3.3.1, the teacher error evolves as a first-order linear recurrence:

$$\epsilon_t(t + 1) = \lambda_{\mathrm{eff}}\epsilon_t(t) + (1 - \alpha)(1 - f\gamma)\epsilon_0,$$

where,

$$\lambda_{\mathrm{eff}} = \alpha + (1 - \alpha)f\gamma(1 - \rho).$$

For $|\lambda_{\mathrm{eff}}| < 1$, solving the recurrence yields the asymptotic error:

$$\epsilon_\infty^{\mathrm{PRL}} = \frac{(1 - \alpha)(1 - f\gamma)\epsilon_0}{1 - \lambda_{\mathrm{eff}}}.$$

Substituting $\lambda_{\mathrm{eff}}$ and simplifying the denominator:

$$\begin{aligned}
1 - \lambda_{\mathrm{eff}} &= 1 - \alpha - (1 - \alpha)f\gamma(1 - \rho) \\
&= (1 - \alpha)[1 - f\gamma(1 - \rho)],
\end{aligned}$$

we obtain:

$$\epsilon_\infty^{\mathrm{PRL}} = \frac{(1 - \alpha)(1 - f\gamma)\epsilon_0}{(1 - \alpha)[1 - f\gamma(1 - \rho)]} = \frac{(1 - f\gamma)\epsilon_0}{1 - f\gamma(1 - \rho)}.$$

**Part 1: Reduced Asymptotic Error.** Define the error ratio:

$$R := \frac{\epsilon_\infty^{\mathrm{PRL}}}{\epsilon_0} = \frac{1 - f\gamma}{1 - f\gamma(1 - \rho)}.$$

Rewriting the denominator:

$$1 - f\gamma(1 - \rho) = 1 - f\gamma + f\gamma\rho.$$

*Case 1:* If $f\gamma = 0$ (no pseudo-labels used) or $\rho = 0$ (all accepted pseudo-labels incorrect):

$$1 - f\gamma(1 - \rho) = 1 - f\gamma \implies R = 1 \implies \epsilon_\infty^{\mathrm{PRL}} = \epsilon_0.$$

*Case 2:* If $f\gamma > 0$ and $\rho > 0$:

$$f\gamma\rho > 0 \implies 1 - f\gamma + f\gamma\rho > 1 - f\gamma.$$

Under the stability condition, both numerator and denominator are positive, so:

$$R = \frac{1 - f\gamma}{1 - f\gamma + f\gamma\rho} < 1 \implies \epsilon_\infty^{\mathrm{PRL}} < \epsilon_0 = \epsilon_\infty^{\mathrm{noPRL}}.$$

This establishes Part (1). $\square$

**Part 2: Monotonicity in Precision.** Taking the derivative of $\epsilon_\infty^{\mathrm{PRL}}$ with respect to $\rho$:

$$\frac{\partial \epsilon_\infty^{\mathrm{PRL}}}{\partial \rho} = \frac{\partial}{\partial \rho}\left[\frac{(1 - f\gamma)\epsilon_0}{1 - f\gamma(1 - \rho)}\right].$$

Let $u = 1 - f\gamma(1 - \rho)$, so $\frac{\partial u}{\partial \rho} = f\gamma$. Applying the chain rule:

$$\frac{\partial \epsilon_\infty^{\text{PRL}}}{\partial \rho} = (1 - f\gamma)\epsilon_0 \cdot \frac{\partial}{\partial \rho}\left[\frac{1}{u}\right]$$

$$= (1 - f\gamma)\epsilon_0 \cdot \left(-\frac{1}{u^2}\right) \cdot f\gamma$$

$$= -\frac{(1 - f\gamma)f\gamma\epsilon_0}{[1 - f\gamma(1 - \rho)]^2}.$$

Since all factors are positive under stability ($f\gamma \in (0, 1)$, $\epsilon_0 > 0$, and $1 - f\gamma(1 - \rho) > 0$ from $\lambda_{\text{eff}} < 1$), we have:

$$\frac{\partial \epsilon_\infty^{\text{PRL}}}{\partial \rho} < 0.$$

Higher precision strictly reduces asymptotic error, establishing Part (2). $\square$

**Part 3: Safe Pseudo-Label Budget.** The stability condition requires $\lambda_{\text{eff}} < 1$:

$$\alpha + (1 - \alpha)f\gamma(1 - \rho) < 1$$
$$(1 - \alpha)f\gamma(1 - \rho) < 1 - \alpha$$
$$f\gamma(1 - \rho) < 1$$
$$f\gamma < \frac{1}{1 - \rho}.$$

This bound quantifies the maximum effective pseudo-label usage that maintains bounded error propagation. Higher precision $\rho$ allows proportionally larger pseudo-label budgets $f\gamma$ while maintaining identical stability guarantees. This establishes Part (3). $\square$

Prototype-guided pseudo-label refinement guarantees that the asymptotic teacher error satisfies $\epsilon_\infty^{\text{PRL}} \leq \epsilon_0$, ensuring that filtered pseudo-labels do not degrade performance. The asymptotic error decreases monotonically with pseudo-label precision, and the effective pseudo-label budget $f\gamma$ can be safely increased under the stability constraint:

$$f\gamma < \frac{1}{1 - \rho}.$$

This provides a quantitative criterion for safe pseudo-label utilization, showing that investing in pseudo-label quality via prototype-based filtering simultaneously reduces long-term error and enlarges the permissible pseudo-label budget, thereby expanding robust training regimes for semi-supervised learning.

$\square$

# D  ROBUSTNESS AND SENSITIVITY ANALYSIS OF GUIDED NOISE INJECTION (GNI)

GNI introduces no additional design hyperparameters. It follows,

$$\widetilde{w}_i = w_i + \tilde{G}_i\xi_i, \qquad \xi_i \sim \mathcal{N}(0, 1),$$

where $\tilde{G}_i$ is a normalized function of $1/(g_i^2 + \varepsilon)$. The term $\varepsilon$ is added for stability when $g_i \to 0$. For all practical purposes, we set the noise variance of $\xi_i$ to 1 and $\varepsilon = 1.0 \times 10^{-8}$.

**Varying $\varepsilon$:** We vary $\varepsilon$ using the following values: $1.0 \times 10^{-6}$, $1.0 \times 10^{-7}$, and $1.0 \times 10^{-9}$. We observe stable performance across settings. A smaller $\varepsilon$ (e.g., $1.0 \times 10^{-9}$) performs slightly better because it allows GNI to more clearly separate "important" parameters from "less important" ones. When $\varepsilon$ is larger, it can dominate the denominator for small gradients, reducing contrast across parameters. Since GNI normalizes all scales to $(0, 1]$, smaller $\varepsilon$ does not cause instability and simply provides sharper contrast.

**Varying the variance of $\xi_i$:** We evaluate two additional noise variances: 0.1 and 10. FoSSIL remains robust to changes in noise variance (0.1 and 1 perform equivalently), while the Vanilla model remains at Dice 0.076.

Table 10: FoSSIL sensitivity/robustness analysis across $\varepsilon$, noise variance, and number of shots $K$.

| Setting | Varying $\varepsilon$ | | | | Noise ($\xi_i$) variance | | | Shots ($K$) | | |
|---|---|---|---|---|---|---|---|---|---|---|
| | $10^{-6}$ | $10^{-7}$ | $10^{-8}$ | $10^{-9}$ | 0.1 | 1 | 10 | 3 | 4 | 5 |
| FoSSIL | 0.443 | 0.437 | 0.460 | 0.482 | 0.460 | 0.460 | 0.408 | 0.414 | 0.434 | 0.460 |
| Vanilla | | | | | 0.076 | | | | | |

Table 11: Retention analysis for different confidence and similarity thresholds. The selected setting ($\tau_{\text{conf}} = 0.7$, $\tau_{\text{sim}} = 0.7$) achieves the best balance between pseudo-label quantity and reliability.

| $\tau_{\text{conf}}$ | $\tau_{\text{sim}}$ | Retention (%) | Decision |
|---|---|---|---|
| 0.6 | 0.6 | 71.7 | Rejected |
| 0.6 | 0.7 | 23.6 | Rejected |
| 0.6 | 0.8 | 1.8 | Rejected |
| 0.7 | 0.6 | 73.7 | Rejected |
| 0.7 | 0.7 | 23.2 | **Selected** |
| 0.7 | 0.8 | 1.9 | Rejected |
| 0.8 | 0.6 | 72.6 | Rejected |
| 0.8 | 0.7 | 22.7 | Rejected |
| 0.8 | 0.8 | 1.8 | Rejected |

**Varying the number of shots** $K$: We monitor performance for different few-shot values $K$ (labeled samples per class). FoSSIL retains strong performance even under severe few-shot settings (e.g., $K = 3$).

Overall, these results show that FoSSIL remains fairly robust despite relying on only a minimal set of hyperparameters.

# E SENSITIVITY ANALYSIS OF PSEUDO-LABEL REFINEMENT (PRL) HYPERPARAMETERS

We analyse pseudo-labels selected using the confidence threshold $\tau_{\text{conf}}$ and similarity threshold $\tau_{\text{sim}}$, where retention (%) denotes the proportion passing both filters (Table 11). Lower thresholds (e.g., 0.6) increase retention but admit many low-confidence labels, whereas higher thresholds become overly strict and drastically reduce retention. The setting $\tau_{\text{conf}} = 0.7$ and $\tau_{\text{sim}} = 0.7$ offers the best balance, preserving a sufficient number of pseudo-labels while maintaining high reliability. We discarded the alternative $\tau_{\text{conf}} = 0.6$ and $\tau_{\text{sim}} = 0.7$ because it offered only a marginal 0.4% increase in retention relative to $\tau_{\text{conf}} = 0.7$ and $\tau_{\text{sim}} = 0.7$, but at the expense of lower confidence. The sensitivity analysis confirms robustness to nearby values: $\tau_{\text{conf}} = 0.8$, $\tau_{\text{sim}} = 0.7$ yields Dice 0.562; $\tau_{\text{conf}} = 0.7$, $\tau_{\text{sim}} = 0.75$ yields Dice 0.567; and $\tau_{\text{conf}} = 0.7$, $\tau_{\text{sim}} = 0.7$ yields Dice 0.554.

# F CHOICE OF LAYER FOR GNI

We evaluated the effect of applying GNI to different parts of the network when the entire model was unfrozen. Specifically, we injected GNI into (i) a deep feature extractor layer in the encoder, (ii) an intermediate decoder layer, and (iii) the final classifier layer. The results show a clear degradation when GNI is applied to deep feature extractors, and a consistent improvement when applied exclusively to the classifier:

- GNI at encoder feature extractor layer: **IoU 22.67**
- GNI at decoder feature extractor layer: **IoU 24.86**
- GNI at final classifier layer $F$: **IoU 27.84**

These results motivate restricting GNI to the final classifier layer $F$.

Table 12: Dataset splits for the Med FoSSIL-Disjoint benchmark. Session 0 (base) uses the **TotalSegmentator (TS)** domain; Session 1 uses **AMOS**; Session 2 uses **BCV**; Session 3 uses **MOTS**; Session 4 uses **BraTS**; and Session 5 uses **VerSe**. All incremental-session training sets follow a **5-shot** protocol.

| Class | Source | Train | Val | Test |
|---|---|---|---|---|
| Sacrum - 1 | TS | 136 | 27 | 43 |
| Stomach - 2 | TS | 124 | 29 | 53 |
| Lung upper lobe left - 3 | TS | 188 | 41 | 92 |
| Lung lower lobe left - 4 | TS | 179 | 39 | 83 |
| Brain - 5 | TS | 125 | 35 | 79 |
| Atrium (left) - 6 | TS | 127 | 25 | 52 |
| Ventricle (left) - 7 | TS | 123 | 24 | 50 |
| Pulmonary artery - 8 | TS | 116 | 17 | 43 |
| Aorta - 9 | TS | 200 | 39 | 85 |
| Gallbladder - 10 | TS | 120 | 27 | 47 |
| Trachea - 11 | TS | 157 | 30 | 68 |
| Rib left1 - 12 | TS | 156 | 28 | 66 |
| Rib right1 - 13 | TS | 156 | 29 | 67 |
| Rib left2 - 14 | TS | 156 | 29 | 67 |
| Rib right2 - 15 | TS | 155 | 29 | 66 |
| Pancreas - 16 | Amos | 120 | 30 | 60 |
| Duodenum - 17 | Amos | 120 | 30 | 60 |
| Bladder - 18 | Amos | 99 | 26 | 47 |
| Prostate/uterus - 19 | Amos | 96 | 26 | 47 |
| Postcava - 20 | Amos | 120 | 30 | 60 |
| Spleen - 21 | BCV | 18 | 4 | 8 |
| Liver - 22 | BCV | 18 | 4 | 8 |
| Left kidney - 23 | BCV | 18 | 4 | 8 |
| Right kidney - 24 | BCV | 18 | 4 | 8 |
| Left adrenal - 25 | BCV | 18 | 4 | 8 |
| Right adrenal - 26 | BCV | 18 | 4 | 8 |
| Hepatic Vessel - 27,28 | MOTS | 7 | 4 | 4 |
| Hepatic Vessel Tumor | MOTS | 7 | 4 | 4 |
| Colon Tumor - 29 | MOTS | 7 | 4 | 4 |
| Lung Tumor - 30 | MOTS | 7 | 4 | 4 |
| NCR — label 1 | BraTS | 40 | 4 | 8 |
| ED — label 2 | BraTS | 40 | 4 | 8 |
| ET — label 4 | BraTS | 37 | 4 | 8 |
| C3(3) | VerSe | 12 | 5 | 6 |
| C4(4) | VerSe | 12 | 5 | 6 |
| T1(8) | VerSe | 31 | 5 | 8 |
| T2(9) | VerSe | 26 | 5 | 7 |

The rationale is threefold:

- Parameter sensitivity differs across the network: Early encoder-decoder layers encode generic features that are reused across all sessions, whereas the final classifier layer is highly task-specific and receives the strongest gradient signals associated with class boundaries. Injecting noise into deep layers would perturb *shared* low-level representations and destabilize all previously learned classes, while injecting noise into $F$ regularizes only the task-specific decision boundaries where overfitting is the highest.

- Local curvature is highest at the classifier layer: In few-shot segmentation, most curvature and gradient magnitude variation appears in the final logits, not in the deep feature extractor. GNI relies on per-parameter gradient magnitude to allocate noise; applying it to layers with uniformly low curvature would add noise where it provides minimal benefit but risks corrupting the global feature space. Applying GNI to $F$ concentrates perturbations exactly where the loss landscape is sharpest and overfitting is most pronounced.

Table 13: Dataset splits for the Med FoSSIL-Mixed benchmark, where both classes and domains may repeat across sessions. Session 0 (base) uses the **AMOS** domain. Session 1 uses **BCV** and **MOTS**. Session 2 uses **TS** and **AMOS**, with three classes repeating (*Right kidney*, *Left kidney*, *Stomach*) from the TS domain. Session 3 uses **MOTS** and **TS**, and Session 4 uses **BraTS** and **VerSe**. All incremental sessions follow a **5-shot** training protocol.

| Class | Source | Train | Val | Test |
|---|---|---|---|---|
| Spleen - 1 | Amos | 203 | 66 | 68 |
| Right kidney - 2 | Amos | 234 | 77 | 77 |
| Left kidney - 3 | Amos | 233 | 78 | 78 |
| Gall bladder - 4 | Amos | 190 | 61 | 67 |
| Esophagus - 5 | Amos | 204 | 67 | 68 |
| Prostate/uterus - 6 | Amos | 167 | 55 | 52 |
| Stomach - 7 | Amos | 233 | 77 | 77 |
| Arota / aorta - 8 | Amos | 204 | 68 | 68 |
| Duodenum - 9 | Amos | 203 | 68 | 68 |
| Postcava - 10 | Amos | 204 | 68 | 68 |
| Pancreas - 11 | BCV | 18 | 6 | 6 |
| Inferior vena cava - 12 | BCV | 18 | 6 | 6 |
| Portal vein and splenic vein - 13 | BCV | 18 | 6 | 6 |
| Left adrenal - 14 | BCV | 18 | 6 | 6 |
| Right adrenal - 15 | BCV | 18 | 6 | 6 |
| Liver - 16 | BCV | 18 | 6 | 6 |
| Hepatic Vessel - 17 | MOTS | 7 | 4 | 4 |
| Hepatic Vessel Tumor - 18 | MOTS | 7 | 4 | 4 |
| Small bowel - 19 | TS | 60 | 20 | 20 |
| Hip_left - 20 | TS | 60 | 20 | 20 |
| Hip_right - 21 | TS | 60 | 20 | 20 |
| Lung upper lobe left - 22 | TS | 60 | 20 | 20 |
| Lung lower lobe left - 23 | TS | 60 | 20 | 20 |
| Bladder - 24 | Amos | 9 | 4 | 4 |
| Colon Tumor - 25 | MOTS | 7 | 4 | 4 |
| Lung Tumor - 26 | MOTS | 7 | 4 | 4 |
| Femur_left - 27 | TS | 18 | 6 | 6 |
| Femur_right - 28 | TS | 18 | 6 | 6 |
| T1 | BraTS | 40 | 4 | 8 |
| T2 | BraTS | 40 | 4 | 8 |
| Flair | BraTS | 37 | 4 | 8 |
| C3(3) | VerSe | 12 | 5 | 6 |
| C4(4) | VerSe | 12 | 5 | 6 |
| T1(8) | VerSe | 31 | 5 | 8 |
| T2(9) | VerSe | 26 | 5 | 7 |

Table 14: Unlabeled sample counts per session in Med Semi-Supervised-FoSSIL benchmark.

| Session | Count |
|---|---|
| Session 1 - Amos | 25 |
| Session 2 - BCV | 30 |
| Session 3 - MOTS | 8 |
| Session 4 - BraTS | 15 |
| Session 5 - VerSe | 20 |

- The feature extractor is shared across tasks; the classifier is task-adaptive: Deeper layers must remain stable to preserve knowledge from previous sessions. The classifier layer is the part of the model that adapts to new domains and novel classes, and therefore benefits

Table 15: Dataset splits for the Natural-FoSSIL benchmark. All incremental sessions follow a **10-shot** training protocol, with 3 validation samples per class.

| Class | Source | Train | Val | Test |
|---|---|---|---|---|
| road - 0 | BDD - Session 0 | 4691 | 1184 | 847 |
| sidewalk - 1 | BDD - Session 0 | 3173 | 796 | 612 |
| building - 2 | BDD - Session 0 | 4270 | 1070 | 758 |
| wall - 3 | BDD - Session 0 | 772 | 189 | 111 |
| fence - 4 | BDD - Session 0 | 1494 | 360 | 206 |
| traffic light - 5 | BDD - Session 0 | 2187 | 556 | 443 |
| traffic sign - 6 | BDD - Session 0 | 3625 | 919 | 660 |
| person - 7 | BDD - Session 0 | 1562 | 393 | 305 |
| car - 8 | BDD - Session 0 | 4738 | 1180 | 833 |
| truck - 9 | BDD - Session 0 | 1451 | 389 | 283 |
| Bridge/tunnel - 10 | IDD - Session 1 | 10 | 3 | 54 |
| Parking - 11 | IDD - Session 1 | 10 | 3 | 173 |
| rail track - 12 | IDD - Session 1 | 10 | 3 | 155 |
| Autorickshaw - 13 | IDD - Session 1 | 10 | 3 | 139 |
| pole - 14 | IDD - Session 1 | 10 | 3 | 220 |
| Bus - 15 (from BDD) | BDD - Session 2 | 10 | 3 | 58 |
| Sky - 16 (from BDD) | BDD - Session 2 | 10 | 3 | 139 |
| bicycle - 17 (from BDD) | BDD - Session 2 | 10 | 3 | 45 |
| person - 7 (repeat from IDD) - 18 | IDD - Session 2 | 10 | 3 | 82 |
| car - 8 (repeat from IDD) - 19 | IDD - Session 2 | 10 | 3 | 91 |

the most from geometry-aware noise. Perturbing shared layers would amplify forgetting, whereas perturbing only $F$ provides regularization without harming the backbone.

# G FoSSIL BENCHMARKS

**Med FoSSIL-Disjoint:** The dataset details are present in the Table 12. The models with U-Net backbone are evaluated for their robustness and ability to handle a large number of incremental sessions while facing diverse domain shifts and a scarce few-shot data. This represents a highly realistic setting in clinical domains, where annotations are costly, and data is collected over time from multiple medical institutions. We evaluated the baselines from this benchmark on only two incremental sessions, as nearly all baselines collapsed after two sessions. We evaluated FoSSIL with a U-Net backbone across all five incremental sessions. The results demonstrate that the framework effectively handles the constraints and maintains strong performance throughout all sessions. This underscores the inability of existing baselines to effectively handle multiple constraints in continual segmentation across a large number of sessions. NC-FSCIL, which performs well for classification in few-shot class-incremental learning, fails on this benchmark. This highlights that a semantic segmentation model with a classifier fixed as a simplex equiangular tight frame (ETF) performs significantly worse than a model with a learnable classifier. MDIL, despite handling different domains with a shared encoder, struggles under strict constraints and suffers a drop in performance. Moreover, having multiple domains is not a practical solution when the number of domains increases. Approaches like generative replay (Gen-Replay) struggle to generate 3D medical volumes even with a diffusion model, which leads to poor performance. Most methods that perform well in incremental or few-shot incremental learning fail when confronted with domain shifts, including segmentation approaches such as MiB, PIFS, and CLIP-CT. Representation and meta-learning-based approaches, such as MAML, SupCL, UnSupCL, and MTL, also fail to retain performance over the sessions. It is observed that feature replay and prototype-based methods, such as C-FSCIL, SoftNet, and our FoSSIL framework, as well as data synthesis-based methods like GAPS, perform significantly better under these constraints.

**Med FoSSIL-Mixed:** The dataset details are present in the Table 13. This is a more realistic clinical setting that involves the same classes appearing across different domains. The goal is to evaluate the model's ability to learn previously seen classes in new domains while retaining performance on the original domains. Additionally, multiple domains may appear within the same session, making learning and adaptation more challenging, as the model must simultaneously learn from diverse

domains while managing the usual constraints. In this setting, we evaluated different transformer backbones, including MedFormer, SwinUNetr, and a large pre-trained CLIP-driven U-Net model, which had already been exposed to most of the 35 classes. Results show that even robust transformer backbones are unable to maintain model performance across sessions. An interesting observation is that the CLIP-driven model, already pre-trained on 21 of the 35 classes, experiences a drop in performance across sessions, highlighting the severity of the constraints in the benchmark.

**Med Semi-Supervised-FoSSIL:** This benchmark is a variant of Med FoSSIL-Disjoint, where few-shot classes have access to unlabeled data. The details of unlabeled data are present in the Table 14. The inclusion of unlabeled data significantly boosts the performance of FoSSIL, an improvement not observed with other methods. Benchmark also highlights how existing semi-supervised approaches fail to handle the constraints and are unable to effectively leverage unlabeled data. **FoSSIL clearly demonstrates that using readily available unlabeled data can significantly improve multi-constraint continual learning for semantic segmentation.**

**Natural-FoSSIL:** The dataset details are present in the Table 15. In autonomous driving, scenes evolve over time with distributional shifts and limited labeled data, making this a highly realistic benchmark. Even large-scale pre-trained models like SAM (trained on over a billion natural scene masks) exhibit forgetting in this realistic and challenging setting. It can be observed that FoSSIL with a SAM backbone achieves further improvements, highlighting that the proposed framework enhances not only simple backbones (e.g., U-Net) and transformer-based backbones (e.g., MedFormer, SwinUNetr) but also heavily pre-trained models like SAM.

**Semi-Supervised Natural-FoSSIL:** The dataset details are present in the Table 16. This benchmark provides access to abundant unlabeled data, and all baselines are evaluated with a DeepLab backbone. In this setting, we employ FoSSIL as a plug-and-play module on top of the GAPS baseline, further improving its results, by leveraging unlabeled data. This demonstrates that unlabeled data can significantly enhance performance and that FoSSIL can effectively boost existing baselines. Similar to the medical benchmarks, existing semi-supervised methods perform poorly in this setting and are unable to effectively leverage unlabeled data. We also evaluated active learning approaches, such as HALO, and coreset selection methods, like RETRIEVE, to select the most informative data; however, these methods fail to achieve significant improvements.

**2D robotic surgery:** We designed a 2D robotic surgery benchmark using two domains, CholecSeg8k[2] and m2caiseg[3], across three sessions. We vary both the classes and the domains to segment different organs and surgical instruments, creating a challenging and realistic setting under a few-shot data regime. The base session (Session 0) includes larger organs to segment, Session 1 introduces tubular structures such as the intestine, and Session 2 contains surgical instruments to segment. In the incremental sessions, we use 50 samples per novel class. The dataset details are present in the Table 17. This benchmark is difficult due to the wide variation in the size, shape, and appearance of the objects to be segmented. Even under these constraints, FoSSIL achieves significantly better performance (Dice score) than CAT[4], which is a specialized baseline for this task as shown in Table 29.

## H  DETECTION RESULTS

To check the effectiveness of FoSSIL on detection tasks with distributional shifts, we tested FoSSIL on the detection dataset COCO-O[5] having different domains with RCNN[6] backbone as shown in Table 18. We created base session with 77 classes from 'Painting' domain and incremental session 1 from 'Weather' domain with three novel disjoint few-shot classes, 'person', 'car', and 'bench'. The performance of base model is 0.2840 mAP. We took 41, 224, 174 unlabeled samples from 'bench', 'car', and 'person' classes, respectively with 30 few-shot labeled samples from each class for session

---

[2] https://arxiv.org/pdf/2012.12453
[3] https://arxiv.org/abs/2008.10134
[4] https://ieeexplore.ieee.org/stamp/stamp.jsp?arnumber=10443356
[5] https://openaccess.thecvf.com/content/ICCV2023/papers/Mao_COCO-O_A_Benchmark_for_Object_Detectors_under_Natural_Distribution_Shifts_ICCV_2023_paper.pdf
[6] https://github.com/microsoft/SoftTeacher

Table 16: Dataset splits for the Semi-Supervised Natural-FoSSIL benchmark, which includes **400 unlabeled samples per class**. All incremental sessions use a **10-shot** training protocol and 3 validation samples per class.

| Class | Source | Train | Val | Test |
|---|---|---|---|---|
| road - 0 | BDD : Session 0 | 4691 | 1184 | 847 |
| sidewalk - 1 | BDD : Session 0 | 3173 | 796 | 612 |
| building - 2 | BDD : Session 0 | 4270 | 1070 | 758 |
| wall - 3 | BDD : Session 0 | 772 | 189 | 111 |
| fence - 4 | BDD : Session 0 | 1494 | 360 | 206 |
| traffic light - 5 | BDD : Session 0 | 2187 | 556 | 443 |
| traffic sign - 6 | BDD : Session 0 | 3625 | 919 | 660 |
| person - 7 | BDD : Session 0 | 1562 | 393 | 305 |
| car - 8 | BDD : Session 0 | 4738 | 1180 | 833 |
| truck - 9 | BDD : Session 0 | 1451 | 389 | 283 |
| parking - 10 | Cityscapes : Session 1 | 10 | 3 | 111 |
| vegetation - 11 | Cityscapes : Session 1 | 10 | 3 | 486 |
| rail track - 12 | IDD : Session 2 | 10 | 3 | 214 |
| sky - 13 | IDD : Session 2 | 10 | 3 | 300 |
| motorcycle - 14 | IDD : Session 3 | 10 | 3 | 609 |
| autorickshaw - 15 | IDD : Session 3 | 10 | 3 | 361 |
| bridge/tunnel - 16 | IDD : Session 3 | 10 | 3 | 65 |

Table 17: Dataset splits for 2D robotic surgery benchmark.

| Category | Class | Source | Train | Val | Test |
|---|---|---|---|---|---|
| misc | Background | CholecSeg8k | 3246 | 713 | 1304 |
| organ | Abdominal Wall | CholecSeg8k | 2774 | 639 | 1158 |
| organ | Liver | CholecSeg8k | 3246 | 713 | 1304 |
| organ | Gastrointestinal Tract | CholecSeg8k | 1832 | 399 | 715 |
| organ | Fat | CholecSeg8k | 2991 | 662 | 1199 |
| organ | Gallbladder | CholecSeg8k | 2810 | 594 | 1124 |
| organ | Connective Tissue | CholecSeg8k | 1116 | 154 | 330 |
| organ | Upper wall | m2caiseg | 50 | 50 | 142 |
| organ | Intestine | m2caiseg | 50 | 4 | 10 |
| instrument | Grasper | m2caiseg | 50 | 50 | 132 |
| instrument | Clipper | m2caiseg | 50 | 15 | 39 |

1. Table 18 illustrates that FoSSIL is able to improve the performance over the Vanilla backbones like RCNN, even on detection tasks with large domain shifts.

# I IMPLEMENTATION DETAILS

We have re-implemented and adapted the following baselines - **Class-incremental Learning**: UCL, MiB, CLIP-CT, Saving100x, C-Flat, UCB, STAR, YoooP, Adapt_replay; **Domain Incremental learning**: MDIL; **Few-shot Class-incremental Learning**: PIFS, Subspace, C-FSCIL, FACT, NC-FSCIL, Gen-Replay, GAPS, SoftNet, FSCIL-SS, FeCAM, BCM, CAT. **Regularization based methods:** Dropout/Weight Decay[7], Variational Dropout[8], Adagrad[9]; **Semi-Supervised Learning based approaches**: RETRIEVE, NNCSL, UaD-CE, CSL. **Other methods**: SupCL, UnSupCL, UnSupCL-HNM, MTL, MAML, CLIP-driven, HALO.

All baselines were adapted with their original settings and hyperparameters. Medical experiments were run for 200 epochs for both base and incremental sessions, while natural and 2D robotic surgery

---

[7]https://proceedings.neurips.cc/paper_files/paper/2020/file/518a38cc9a0173d0b2dc088166981cf8-Paper.pdf

[8]https://link.springer.com/article/10.1007/s10994-023-06487-7

[9]https://openreview.net/pdf?id=WwQKl1OrMX

Table 18: Performance of FoSSIL on COCO-O detection benchmark with one incremental session on three few-shot novel classes. We compared FoSSIL without Guided Noise Injection (GNI) and Pseudo-Label Refinement (PLR).

| Method | Session 1 (mAP) |
|---|---|
| RCNN (Vanilla with only unlabeled data) | 0.000 |
| FoSSIL | 0.063 |
| FoSSIL w/o GNI | 0.004 |
| FoSSIL w/o PLR | 0.051 |

Table 19: Configuration of FoSSIL in different settings.

| Method | Base Frozen (Yes/No) | Incremental Sessions Frozen (which layers) |
|---|---|---|
| Med FoSSIL-Disjoint (U-Net) | No | Yes (All but last) |
| Med FoSSIL-Mixed (MedFormer) | No | Yes (All but last) |
| Med FoSSIL-Mixed (SwinUNetr) | No | Yes (All but last) |
| Natural-FoSSIL (DeepLab) | No | No |
| Semi-Supervised Natural-FoSSIL (Deeplab) | No | No |
| Med Semi-Supervised-FoSSIL (U-Net) | No | Yes (All but last) |
| Med Semi-Supervised-FoSSIL (MedFormer) | No | Yes (All but last) |
| Natural-FoSSIL (SAM) | No | Yes (All but last) |
| 2D robotic surgery (U-Net) | No | No |
| 2D robotic surgery (MedFormer) | No | No |

experiments were run for 100 epochs per session. All SAM experiments were run for 10 epochs. We used one A100 40GB GPU for all experiments. Table 19 shows the layers frozen in the FoSSIL models for each benchmark.

## J  GNI vs. existing regularization-based continual learning methods

### J.1  GNI as a Gradient-Guided Stochastic Regularizer

Guided Noise Injection (GNI) introduces parameter-wise stochastic perturbations whose magnitudes depend on instantaneous gradient sensitivity. For each parameter $w_i$, GNI computes the squared gradient,

$$G_i = \left(\frac{\partial L}{\partial w_i}\right)^2,$$

and defines an inverse sensitivity,

$$G_i^{-1} = \frac{1}{G_i + \varepsilon}.$$

These values are then normalized into the interval $(0, 1]$ to produce noise scales $\tilde{G}_i$, and each parameter is perturbed as,

$$w_i \leftarrow w_i + \tilde{G}_i \xi_i, \qquad \xi_i \sim \mathcal{N}(0, 1).$$

Large-gradient parameters receive minimal perturbation, while flat or low-gradient directions receive proportionally larger perturbations. This produces a parameter-level perturbation scheme whose magnitude is determined by local curvature, allowing the model to explore flat regions while preserving stability along sharp directions.

### J.2  GNI vs. Adaptive Optimizers

Adaptive optimizers such as Adam, RMSProp, and Adagrad rescale gradients using accumulated statistics but do not inject *structured perturbations* into the parameters. For example, Adam updates a parameter $w_t$ as,

$$w_{t+1} = w_t - \eta \frac{m_t}{\sqrt{v_t} + \varepsilon},$$

where $m_t$ and $v_t$ are exponential moving averages of the gradient and its square. The second moment term $v_t$ acts as a diagonal preconditioner that provides a coarse and history based approximation of local curvature. However, these methods do not perform explicit exploration of the loss surface and do not incorporate true second order information that requires evaluation in a neighborhood of the current parameters. Their behavior depends on accumulated gradient statistics, which can become stale when the data distribution changes, leading to slower or less reliable adaptation in continual settings. GNI differs fundamentally by applying noise directly to the parameters and by basing its magnitude solely on the current sensitivity $G_i^{-1}$, enabling selective stability and plasticity even in few-shot incremental settings.

### J.3 GNI VS. UNCERTAINTY-BASED REGULARIZATION

Uncertainty-Based adaptive methods, such as UCL, maintain per-node uncertainty values $\sigma_i^2$ that are tied across all incoming weights of a neuron, which reduces the memory overhead relative to per-weight variational approaches. However, these uncertainty estimates depend on statistics accumulated from previous tasks and may become less informative under strong domain shift, since few-shot updates provide limited evidence to revise them. Because a single variance is shared at the node level, the resulting anisotropy is coarse. In contrast, GNI computes inverse-squared gradient values directly from the current data, requires no stored task-level statistics, and provides full parameter-level anisotropy.

### J.4 GNI VS. DROPOUT AND VARIATIONAL DROPOUT

Standard dropout introduces fixed noise independent of parameter sensitivity, while variational dropout can learn per-weight or per-unit dropout rates. The methods may permanently reduce the capacity of weights once they are heavily dropped, and neither variant uses curvature or gradient information to guide noise placement. As a result, dropout methods may struggle under domain shift and few-shot conditions because they inject noise independently of parameter sensitivity. GNI avoids these issues by deriving a parameter-specific noise scale from instantaneous gradients, assigning low noise to important weights and higher noise to less relevant ones. This dynamic adjustment preserves capacity across sessions and enables far more effective adaptation than static dropout mechanisms.

### J.5 GNI VS. WEIGHT DECAY

Weight decay introduces a uniform penalty,

$$L_{\mathrm{wd}} = \lambda \|W\|^2,$$

which shrinks all parameters equally regardless of their relevance or curvature. Important parameters may be unnecessarily suppressed, while flat directions receive no preferential treatment. GNI replaces such uniform shrinkage with gradient-guided noise whose magnitude is inversely related to curvature. The behaviour of $\tilde{G}_i$ is theoretically justified by analyzing the perturbed loss under a second-order Taylor approximation. This analysis shows that, after normalization, the resulting perturbations remain stable and bounded. The formulation suppresses noise in high-curvature (large-gradient) directions while promoting exploration along flat directions, thereby providing a geometry-aware alternative to classical $\ell_2$ regularization.

### J.6 GNI VS. SHARPNESS-BASED METHODS (SAM AND C-FLAT)

Sharpness-Aware Minimization (SAM)[10] performs a deterministic min-max optimization step that perturbs the parameters within a fixed-radius neighborhood and then updates them using the resulting worst-case gradient. This produces a perturbation defined over a fixed-radius neighborhood of the full parameter vector, which does not provide parameter-wise differentiation. As a result, SAM alters the optimization trajectory but provides no mechanism to identify which weights are important for previous tasks or which directions should be preserved for subsequent ones.

C-Flat extends SAM to continual learning by optimizing a surrogate objective that couples neighborhood perturbations with Hessian-vector directional curvature terms. This encourages the

---

[10]https://arxiv.org/pdf/2010.01412

Table 20: Performance comparison on Med FoSSIL-Disjoint benchmark showing that GNI provides substantial gains over existing regularization-based methods for continual learning.

| Method | Session 0 | Session 1 | Session 2 |
|---|---|---|---|
| Dropout / Weight Decay | 0.700 | 0.191 | – |
| Variational Dropout | 0.700 | 0.260 | – |
| Adagrad | 0.700 | 0.024 | – |
| UCL | 0.700 | 0.430 | 0.325 |
| C-Flat | 0.700 | 0.174 | 0.030 |
| STAR | 0.700 | 0.050 | 0.020 |
| **FoSSIL (GNI)** | **0.736** | **0.460** | **0.398** |

model to remain in locally smooth regions of the loss landscape and reduces sensitivity to task specific sharp minima. However, because its flatness and curvature terms operate on neighborhoods of the full parameter vector rather than individual parameters, C-Flat does not provide weight-level control over which directions should remain stable and which should remain adaptable. Consequently, C-Flat enforces a uniform notion of flatness across the network rather than assigning stability or plasticity based on parameter importance. In addition, its neighborhood radius and curvature coefficients introduce multiple interacting hyperparameters, increasing optimization complexity and limiting interpretability.

GNI is built on a fundamentally different design principle. Instead of imposing global stability constraints, it injects parameter wise adaptive noise derived directly from the instantaneous gradient. For each weight $w_i$, the corresponding gradient $g_i$ determines a noise scale $\tilde{G}_i$ which provides an automatic, data driven stability plasticity trade off at the parameter level. Weights with large gradients, which actively contribute to current task performance, receive negligible noise, preserving critical information. Weights with small gradients receive larger perturbations, enabling rapid adaptation without interfering with previously consolidated structure. This yields a local, lightweight, and interpretable mechanism for continual learning that avoids global curvature estimation and removes the need for complex objective balancing. Unlike C-Flat's uniform landscape smoothing, GNI selectively preserves what is important and selectively explores what is flexible, using a simple rule rooted in gradient geometry.

### J.7 GNI vs. PERTURBATION-BASED STABILITY METHODS (STAR)

The STAR method stabilizes learning by introducing an adversarial-style perturbation $\Delta w$ that amplifies model sensitivity on stored buffer samples and then penalizing the resulting output drift through a KL divergence term,

$$L_{\text{STAR}} = \text{KL}\big(f(x; w) \,\|\, f(x; w + \Delta w)\big).$$

The perturbation is obtained by initializing each layer with small noise to avoid zero gradients, then performing one gradient-ascent step on the KL divergence at the perturbed point, and finally normalizing the gradient *per layer* to control the perturbation magnitude.

This layer-wise normalization controls the perturbation scale at the layer granularity, rather than at the level of individual parameters. STAR also relies on stored buffer samples and requires additional forward and backward passes to compute both the perturbed outputs and the ascent direction, introducing nontrivial memory and computational overhead.

In contrast, GNI requires no buffer and no auxiliary training passes. Its stability mechanism arises directly from the instantaneous local gradient structure, enabling parameter-wise, curvature-sensitive regularization with minimal overhead.

Table 20 summarizes results on the Med FoSSIL-Disjoint benchmark, demonstrating the superior performance of FoSSIL with GNI.

## K LIMITATIONS & FUTURE WORK

### K.1 SEVERE DOMAIN SHIFT

Tables 3 and 7 reveal an extreme failure case at **Session 4**, where the Dice score for a U-Net backbone drops to nearly zero. This behavior is expected given the session ordering:

- Session 0: TotalSegmentator (**CT**)

- Session 1: AMOS (dominantly **CT**)

- Session 2: BCV (**CT**)

- Session 3: MOTS (**CT**)

- Session 4: BraTS (**MRI**)

- Session 5: Verse (**CT**)

Across the first four sessions, the model is exposed almost exclusively to CT scans. When the distribution abruptly shifts to MRI at Session 4, lightweight encoder-decoder architectures such as U-Net exhibit catastrophic forgetting of the CT domain, causing the Dice score to collapse to near-zero values. FoSSIL alleviates this drop to a limited extent: the Dice score improves from 0.025 (Table 3) to 0.0576 (Table 7) under the semi-supervised setting.

This trend underscores both the **severity and practical relevance** of the proposed FoSSIL benchmarks, while also exposing the limitations of simple convolutional architectures.

**Mitigation Strategies: (1) Transformer backbones -** Transformer-based architectures such as MedFormer exhibit stronger cross-domain stability and retain meaningful performance under severe shifts, achieving a Dice score of 0.323 in Session 4 with FoSSIL (Table 7). We recommend transformer backbones when facing substantial domain heterogeneity. **(2) Leveraging more unlabeled data -** we further study the effect of additional publicly available unlabeled CT data (TotalSegmentator, AMOS, BCV, MOTS) under a U-Net backbone.

Table 21: Impact of additional unlabeled CT data on Session 4 performance.

| Model (U-Net backbone) | Session 4 |
| --- | --- |
| FoSSIL (no unlabeled data) | 0.025 |
| FoSSIL (15 unlabeled samples) | 0.058 |
| FoSSIL (42 unlabeled samples) | 0.197 |

The steady improvement demonstrates the strength of the Prototype-Guided Pseudo-Label Refinement (PRL) mechanism in utilizing unlabeled data under domain shift.

### K.2 NOISY UNLABELED DATA

In real clinical scenarios, medical scans often suffer from noise, blur, and contrast degradation. To assess robustness under such realistic artifacts, we evaluate FoSSIL under moderate and heavy noise settings applied to the *unlabeled* data.

**Boundary-Aware Noise Generation:** We employ a boundary-aware degradation module. Sobel filtering extracts structural boundaries, which are converted into a soft mask $B(x, y, z) \in [0, 1]$. Blur, contrast shifts, and additive Gaussian noise are then applied primarily to high-boundary regions (Table 22), while homogeneous regions remain minimally affected. This degradation is applied to 70% of unlabeled samples to mimic natural variability.

FoSSIL shows strong resilience as heavy noise reduces performance by only **0.002** compared to the moderate setting. This indicates that the method is **highly robust to substantial degradations** in unlabeled images. Nevertheless, further improving robustness to noise in continual semi-supervised segmentation remains an important direction for future work.

Table 22: Noise schedules used for unlabeled image degradation.

| Operation | Randomness | Moderate | Heavy |
|---|---|---|---|
| Blur | Uniform | $\sigma \in [0.5, 1.5]$ | $\sigma \in [1.0, 2.5]$ |
| Contrast | Uniform | $1.0 \pm 0.15$ | $1.0 \pm 0.30$ |
| Additive Noise | Gaussian | $\sigma = 0.02$ | $\sigma = 0.05$ |

Table 23: Performance of FoSSIL under noise-corrupted unlabeled data.

| Method | Session 1 |
|---|---|
| FoSSIL (no noise) | 0.554 |
| FoSSIL (moderate noise) | 0.477 |
| FoSSIL (heavy noise) | 0.475 |
| UaD-CE | 0.082 |

### K.3 3D MEDICAL DOMAIN

**(i) Organ size and few-shot:** Segmentation performance in 3D medical datasets strongly depends on organ size. Small structures (e.g., adrenal glands, colon tumors, BraTS tumor subregions)[11] are known to be difficult to segment, and this difficulty is amplified in few-shot continual learning. **(ii) Base model accuracy affects retention:** Retention in continual learning depends on the initial accuracy of the base model. Methods starting near Dice 0.9 may retain high absolute performance after several sessions, whereas methods starting near Dice 0.6-0.7 may naturally yield lower absolute values even with good retention. Most FoSSIL medical benchmarks fall within the 0.6-0.7 range, which explains the absolute performance levels observed across sessions. **(iii) Long multi-session continual learning is inherently harder:** Existing medical continual segmentation works typically perform only 1-3 incremental steps (CLIP-CT), whereas FoSSIL stress-tests up to six incremental sessions across multiple modalities and classes. More sessions naturally lead to lower absolute accuracy, even when relative improvements and stability are strong.

### K.4 AUTONOMOUS DRIVING

**(i) Dataset complexity and backbone capacity:** Recent foundation models[12] achieve 50-60% IoU on BDD and Cityscapes even with a powerful ViT-H ($\sim$630M parameters) backbone. FoSSIL uses substantially smaller backbones (ViT-B, $\sim$90M parameters). The challenging nature of the dataset inherently restricts the maximum IoU that models can achieve. **(ii) Continual segmentation performance is generally low:** State-of-the-art continual semantic segmentation typically achieves only 20-50% IoU under realistic multi-domain settings[13]. FoSSIL additionally operates in a few-shot regime, where the scarcity of labels further reduces absolute accuracy.

### K.5 2D ROBOTIC SURGERY

This domain is exceptionally challenging due to rapid appearance changes in organs, tools, lighting, and camera viewpoints. As shown in CAT, existing methods achieve only 5-30% IoU for incremental classes in continual setups, underscoring the inherent difficulty of the task.

## L ADDITIONAL ABLATIONS

We evaluated FoSSIL across class-incremental learning, domain incremental learning, mixed class and domain shifts, and semi-supervised CIL settings with scarce labeled data, spanning 2D robotic surgery, 3D medical segmentation, and autonomous driving. Across all tasks, specialized methods, whether designed for CIL (CAT, BCM), DIL (SAM, CLIP), mixed class and domain shifts (MDIL), or semi-supervised CIL (UaD-CE), exhibit severe performance degradation over incremental sessions.

---

[11]https://arxiv.org/pdf/2501.09138
[12]https://arxiv.org/pdf/2510.27047
[13]https://ieeexplore.ieee.org/stamp/stamp.jsp?arnumber=10521870

In contrast, FoSSIL consistently maintains strong performance across all benchmarks, demonstrating robustness to class shifts, domain shifts, scarce labels, and unlabeled data, both individually and in combination.

## L.1 2D ROBOTIC SURGERY TASK

To study the effect of each setup, we performed class-incremental learning, domain incremental learning, and their combination (CDIL) under data-scarce conditions. We evaluated widely used backbones such as Medformer (a Transformer-based model) and U-Net for semantic segmentation.

This task uses two domains, CholecSeg8k and m2caiseg, and varies both the class set and the domain across sessions. For class-incremental learning, we vary the classes within CholecSeg8k. For domain incremental learning, we shift the domain from CholecSeg8k (Session 0) to m2caiseg (Session 1). For combined class and domain incremental learning, we vary both the domain and the class set when moving from CholecSeg8k (Session 0) to m2caiseg (Session 1).

Table 24: Performance of class-incremental learning (CIL), domain incremental learning (DIL), and their combination (CDIL) under data-scarce settings. S denotes the session; for example, S0 corresponds to Session 0.

| Model | CIL S0 | CIL S1 | CIL S2 | DIL S0 | DIL S1 | CDIL S0 | CDIL S1 | CDIL S2 |
|---|---|---|---|---|---|---|---|---|
| U-Net Vanilla | 0.770 | 0.003 | 0.000 | 0.770 | 0.223 | 0.770 | 0.033 | 0.007 |
| Medformer Vanilla | 0.762 | 0.026 | 0.002 | 0.762 | 0.166 | 0.762 | 0.011 | 0.002 |
| CAT U-Net | 0.770 | 0.102 | 0.006 | 0.770 | 0.258 | 0.770 | 0.166 | 0.001 |
| FoSSIL Medformer | 0.762 | 0.166 | 0.144 | 0.762 | 0.279 | 0.762 | 0.169 | 0.156 |
| FoSSIL U-Net | 0.770 | 0.212 | 0.159 | 0.770 | 0.305 | 0.770 | 0.244 | 0.163 |

In Table 24, we observe that all models are affected, and their performance drops significantly across these settings. Interestingly, transformer-based models show poorer performance compared to simpler encoder-decoder models such as U-Net.

## L.2 AUTONOMOUS DRIVING

To study the effect of each setup, we performed class-incremental learning, domain incremental learning, and their combination under data scarce conditions. We used SAM as the backbone. For class-incremental learning, we varied the classes within BDD (Berkeley Deep Drive). For domain incremental learning, we shifted the domain from BDD (Session 0) to IDD (Indian Driving Dataset, Session 1). For combined class and domain incremental learning, we varied both the domain and the class set when moving from BDD (Session 0) to IDD (Session 1).

Table 25: Performance of class-incremental learning (CIL), domain incremental learning (DIL), and their combination (CDIL) under data-scarce settings. S denotes the session; for example, S0 corresponds to Session 0.

| Model | CIL S0 | CIL S1 | DIL S0 | DIL S1 | CDIL S0 | CDIL S1 | CDIL S2 |
|---|---|---|---|---|---|---|---|
| SAM Vanilla | 66.0 | 30.49 | 66.0 | 30.43 | 66.0 | 32.6 | 30.81 |
| SAM FoSSIL | 66.0 | 31.39 | 66.0 | 30.64 | 66.0 | 33.2 | 31.22 |

Even heavily pre-trained models such as SAM show a noticeable drop in performance (Table 25) when evaluated under class-incremental and domain incremental learning setups.

**The above settings clearly show that models specifically designed for class-incremental learning, domain incremental learning, data scarcity, or the use of unlabeled data still fail to perform reliably. In contrast, the FoSSIL framework is able to handle each of these setups individually, as well as the more complex continual scenarios captured in the FoSSIL benchmarks.**

## M PRACTICAL GUIDELINES FOR RELIABLE PERFORMANCE

Below, we provide key recommendations for effectively deploying the FoSSIL framework in multi-constrained continual semantic segmentation settings.

Table 26: Total Drop (%) for all baselines on Med FoSSIL-Disjoint (lower is better).

| Method | Total Drop (%) |
|---|---|
| PIFS | 88.9 |
| NC-FSCIL | 80.4 |
| CLIP-CT | 70.3 |
| MiB | 86.3 |
| MDIL | 87.5 |
| C-FSCIL | 62.3 |
| SoftNet | 82.2 |
| GAPS | 63.8 |
| FSCIL-SS | 87.3 |
| Subspace | 84.4 |
| Gen-Replay | 89.1 |
| FeCAM | 94.0 |
| FACT | 92.2 |
| MAML | 99.9 |
| MAML + Reg. | 99.9 |
| MTL | 88.7 |
| UnSupCL | 94.4 |
| SupCL | 94.0 |
| UnSupCL-HNM | 95.0 |
| C-Flat | 95.7 |
| STAR | 96.9 |
| Saving100x | 92.6 |
| YoooP | 96.9 |
| UCB | 81.6 |
| BCM | 100.0 |
| Adapt_replay | 96.4 |
| UCL | 53.6 |
| **FoSSIL (U-Net)** | **45.9** |

## M.1 ROLE OF UNLABELED DATA

Domain-relevant unlabeled data plays a crucial role in FoSSIL. As observed in Session 4 when BraTS (MRI modality) was introduced, unlabeled data substantially augments novel few-shot classes and helps the model retain modality- and domain-specific structure when supervision is scarce. Since unlabeled data is far cheaper to acquire than labeled samples, it provides large practical gains $(0.025 \rightarrow 0.058 \rightarrow 0.197)$. We therefore recommend using unlabeled data whenever possible.

## M.2 ROLE OF $\varepsilon$

The hyperparameter $\varepsilon$ can be selected using a small validation set based on the deployment scenario. Preliminary observations show that appropriate choices of $\varepsilon$ consistently improve performance.

## M.3 TOTAL DROP (%)

Absolute Dice scores at the final session may recover despite catastrophic collapses during intermediate sessions, particularly in real-world incremental settings like medical imaging (scanner/protocol drift), autonomous driving (weather/city changes), robotics (task-by-task updates), and continual deployment scenarios. **Total Drop (%)** captures the full forgetting trajectory and provides a more reliable measure of stability over time.

$$\text{Total Drop} = \left( \frac{\sum_i \max(0, S_i - S_{i+1})}{S_0} \right) \times 100.$$

where $S_0$ is the base session and $S_i$ is the incremental session $i$. Table 26 shows that FoSSIL is the most reliable method, with the lowest Total Drop.

## M.4 Model Choice Under Budget Constraints

When resources permit, pre-trained models combined with FoSSIL offer the best performance and robustness. Under moderate budgets, U-Net or transformer backbones are effective choices. Transformers show greater resilience to severe domain shifts, whereas U-Nets remain strong in standard settings. For lightweight backbones, FoSSIL with unlabeled data provides the best trade-off, since unlabeled data is easy to obtain and significantly improves performance. This is especially useful when models must be updated frequently or across many continual sessions.

## M.5 Extreme Data-Scarce Settings

FoSSIL performs reliably even under extreme label scarcity. For Med FoSSIL-Disjoint, Session 1 performance with $K = 4$ and $K = 3$ labeled samples reaches 0.4343 and 0.4141, respectively, remaining close to the $K = 5$ performance of 0.460 and consistently outperforming baselines. This makes FoSSIL highly practical in settings where annotation is severely limited.

## N Additional results

We report class-wise results for FoSSIL and some baselines. We show how class performance varies across sessions. We report results of FoSSIL without guided noise injection in Table 34 and without pseudo-label refinement in Table 36. Table 32 and Table 44 highlight that the performance of pre-trained models deteriorates in the proposed multi-constraint benchmark. Figure 6 shows the performance of *seen* and *new* classes in different settings.

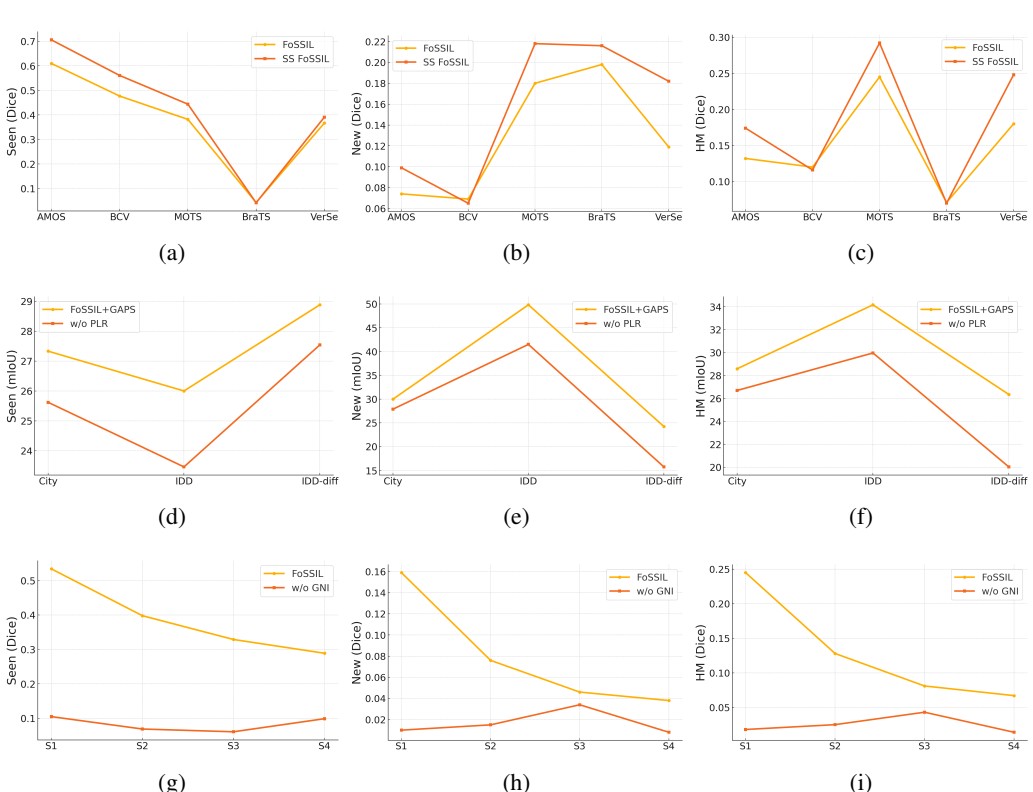

Figure 6: **(a-c)** Performance of FoSSIL on the Med FoSSIL-Disjoint benchmark and its variant, Med Semi-Supervised-FoSSIL (which includes additional unlabeled data), evaluated across incremental sessions (TS (Base) → AMOS → BCV → MOTS → BraTS → VerSe). SS refers to Semi-Supervised. SS FoSSIL denotes the performance of FoSSIL on the Med Semi-Supervised-FoSSIL benchmark. The reported results confirm that unlabeled data helps to boost the performance. **(d-f)** Performance of FoSSIL on the Semi-Supervised Natural-FoSSIL benchmark (which includes additional unlabeled data), evaluated across incremental sessions (BDD100K (Base) → Cityscapes → IDD → IDD-diff (with different classes from the previous session)). The results show that pseudo-label refinement (PRL) enhances the performance of FoSSIL compared to FoSSIL without pseudo-label refinement (w/o PRL). **(g-i)** Performance of FoSSIL (MedFormer) on the Med FoSSIL-Mixed benchmark evaluated across incremental sessions (Session 0 (Base) → Session 1 → Session 2 → Session 3 → Session 4) The results show that guided noise injection (GNI) enhances the performance of FoSSIL compared to FoSSIL without guided noise injection (w/o GNI). Seen refers to the average performance on classes the model has encountered in previous sessions, while New refers to the average performance on classes introduced in the current session. HM refers to Harmonic Mean.

Table 27: Class-wise performance of FoSSIL on Med FoSSIL-Disjoint with U-Net.

| Class | Session 0 | Session 1 | Session 2 | Session 3 | Session 4 | Session 5 |
|---|---|---|---|---|---|---|
| class_0 | 0.979 | 0.962 | 0.848 | 0.764 | 0.944 | 0.672 |
| class_1 | 0.792 | 0.614 | 0.611 | 0.553 | 0.002 | 0.791 |
| class_2 | 0.613 | 0.209 | 0.341 | 0.454 | 0.002 | 0.541 |
| class_3 | 0.856 | 0.762 | 0.739 | 0.717 | 0.003 | 0.834 |
| class_4 | 0.756 | 0.636 | 0.623 | 0.646 | 0.003 | 0.736 |
| class_5 | 0.910 | 0.769 | 0.730 | 0.601 | 0.458 | 0.877 |
| class_6 | 0.653 | 0.611 | 0.596 | 0.557 | 0.127 | 0.622 |
| class_7 | 0.704 | 0.551 | 0.559 | 0.568 | 0.029 | 0.656 |
| class_8 | 0.875 | 0.835 | 0.802 | 0.733 | 0.182 | 0.810 |
| class_9 | 0.768 | 0.719 | 0.711 | 0.687 | 0.048 | 0.753 |
| class_10 | 0.610 | 0.272 | 0.480 | 0.542 | 0.002 | 0.444 |
| class_11 | 0.803 | 0.711 | 0.687 | 0.683 | 0.030 | 0.825 |
| class_12 | 0.824 | 0.674 | 0.681 | 0.650 | 0.082 | 0.732 |
| class_13 | 0.729 | 0.638 | 0.592 | 0.579 | 0.069 | 0.652 |
| class_14 | 0.732 | 0.635 | 0.595 | 0.397 | 0.046 | 0.689 |
| class_15 | 0.705 | 0.431 | 0.500 | 0.539 | 0.019 | 0.654 |
| class_16 | – | 0.129 | 0.142 | 0.134 | 0.040 | 0.122 |
| class_17 | – | 0.043 | 0.058 | 0.062 | 0.017 | 0.084 |
| class_18 | – | 0.050 | 0.068 | 0.071 | 0.021 | 0.072 |
| class_19 | – | 0.053 | 0.017 | 0.012 | 0.016 | 0.015 |
| class_20 | – | 0.096 | 0.073 | 0.070 | 0.017 | 0.062 |
| class_21 | – | – | 0.032 | 0.432 | 0.031 | 0.165 |
| class_22 | – | – | 0.287 | 0.125 | 0.024 | 0.111 |
| class_23 | – | – | 0.073 | 0.093 | 0.003 | 0.018 |
| class_24 | – | – | 0.037 | 0.031 | 0.001 | 0.042 |
| class_25 | – | – | 0.004 | 0.008 | 0.015 | 0.010 |
| class_26 | – | – | 0.000 | 0.003 | 0.002 | 0.001 |
| class_27 | – | – | – | 0.075 | 0.000 | 0.087 |
| class_28 | – | – | – | 0.031 | 0.000 | 0.032 |
| class_29 | – | – | – | 0.338 | 0.000 | 0.292 |
| class_30 | – | – | – | 0.277 | 0.014 | 0.043 |
| class_31 | – | – | – | – | 0.109 | 0.060 |
| class_32 | – | – | – | – | 0.185 | 0.008 |
| class_33 | – | – | – | – | 0.299 | 0.158 |
| class_34 | – | – | – | – | – | 0.056 |
| class_35 | – | – | – | – | – | 0.133 |
| class_36 | – | – | – | – | – | 0.222 |
| class_37 | – | – | – | – | – | 0.066 |
| **Average** | 0.736 | 0.459 | 0.398 | 0.329 | 0.025 | 0.324 |

Table 28: Class-wise performance of FoSSIL on Med FoSSIL-Mixed with MedFormer.

| Class | Session 0 | Session 1 | Session 2 | Session 3 | Session 4 |
|---|---|---|---|---|---|
| class_0 | 0.973 | 0.892 | 0.916 | 0.855 | 0.842 |
| class_1 | 0.695 | 0.708 | 0.641 | 0.629 | 0.690 |
| class_2 | 0.705 | 0.524 | 0.606 | 0.682 | 0.653 |
| class_3 | 0.608 | 0.527 | 0.585 | 0.589 | 0.411 |
| class_4 | 0.367 | 0.263 | 0.300 | 0.318 | 0.300 |
| class_5 | 0.646 | 0.530 | 0.540 | 0.611 | 0.615 |
| class_6 | 0.533 | 0.341 | 0.437 | 0.462 | 0.442 |
| class_7 | 0.562 | 0.546 | 0.548 | 0.526 | 0.525 |
| class_8 | 0.870 | 0.861 | 0.693 | 0.867 | 0.734 |
| class_9 | 0.570 | 0.449 | 0.456 | 0.484 | 0.371 |
| class_10 | 0.709 | 0.592 | 0.483 | 0.514 | 0.512 |
| class_11 | – | 0.208 | 0.225 | 0.222 | 0.102 |
| class_12 | – | 0.466 | 0.436 | 0.248 | 0.404 |
| class_13 | – | 0.025 | 0.196 | 0.205 | 0.200 |
| class_14 | – | 0.002 | 0.042 | 0.050 | 0.046 |
| class_15 | – | 0.002 | 0.067 | 0.072 | 0.076 |
| class_16 | – | 0.520 | 0.294 | 0.232 | 0.224 |
| class_17 | – | 0.023 | 0.097 | 0.101 | 0.099 |
| class_18 | – | 0.029 | 0.087 | 0.073 | 0.076 |
| class_19 | – | – | 0.159 | 0.231 | 0.236 |
| class_20 | – | – | 0.084 | 0.164 | 0.085 |
| class_21 | – | – | 0.025 | 0.155 | 0.167 |
| class_22 | – | – | 0.045 | 0.073 | 0.069 |
| class_23 | – | – | 0.120 | 0.173 | 0.184 |
| class_24 | – | – | 0.018 | 0.120 | 0.115 |
| class_25 | – | – | – | 0.099 | 0.079 |
| class_26 | – | – | – | 0.067 | 0.184 |
| class_27 | – | – | – | 0.013 | 0.036 |
| class_28 | – | – | – | 0.008 | 0.071 |
| class_29 | – | – | – | – | 0.018 |
| class_30 | – | – | – | – | 0.101 |
| class_31 | – | – | – | – | 0.086 |
| class_32 | – | – | – | – | 0.025 |
| class_33 | – | – | – | – | 0.014 |
| class_34 | – | – | – | – | 0.019 |
| class_35 | – | – | – | – | 0.020 |
| **Average** | 0.626 | 0.367 | 0.299 | 0.285 | 0.228 |

Table 29: Performance of FoSSIL on 2D robotic surgery.

| Model | Session 0 | Session 1 | Session 2 |
|---|---|---|---|
| UNet Vanilla | 0.770 | 0.033 | 0.007 |
| Medformer Vanilla | 0.762 | 0.011 | 0.002 |
| CAT (UNet) | 0.770 | 0.166 | 0.001 |
| FoSSIL - Medformer | 0.762 | 0.169 | 0.156 |
| FoSSIL - UNet | 0.770 | 0.244 | 0.163 |

Table 30: Class-wise performance of FoSSIL on Med FoSSIL-Mixed with SwinUNetr.

| Class | Session 0 | Session 1 | Session 2 | Session 3 | Session 4 |
|-------|-----------|-----------|-----------|-----------|-----------|
| class_0 | 0.972 | 0.943 | 0.961 | 0.938 | 0.960 |
| class_1 | 0.802 | 0.805 | 0.754 | 0.749 | 0.788 |
| class_2 | 0.415 | 0.199 | 0.391 | 0.388 | 0.399 |
| class_3 | 0.551 | 0.576 | 0.506 | 0.526 | 0.560 |
| class_4 | 0.301 | 0.213 | 0.163 | 0.145 | 0.190 |
| class_5 | 0.657 | 0.567 | 0.618 | 0.548 | 0.593 |
| class_6 | 0.574 | 0.428 | 0.465 | 0.472 | 0.458 |
| class_7 | 0.592 | 0.306 | 0.531 | 0.549 | 0.537 |
| class_8 | 0.847 | 0.497 | 0.729 | 0.844 | 0.845 |
| class_9 | 0.524 | 0.363 | 0.463 | 0.471 | 0.444 |
| class_10 | 0.694 | 0.089 | 0.352 | 0.377 | 0.222 |
| class_11 | – | 0.339 | 0.124 | 0.159 | 0.192 |
| class_12 | – | 0.754 | 0.709 | 0.734 | 0.749 |
| class_13 | – | 0.044 | 0.048 | 0.032 | 0.028 |
| class_14 | – | 0.000 | 0.028 | 0.025 | 0.021 |
| class_15 | – | 0.000 | 0.025 | 0.030 | 0.016 |
| class_16 | – | 0.538 | 0.114 | 0.073 | 0.065 |
| class_17 | – | 0.011 | 0.058 | 0.045 | 0.048 |
| class_18 | – | 0.009 | 0.014 | 0.006 | 0.003 |
| class_19 | – | – | 0.181 | 0.027 | 0.016 |
| class_20 | – | – | 0.034 | 0.065 | 0.063 |
| class_21 | – | – | 0.082 | 0.094 | 0.088 |
| class_22 | – | – | 0.016 | 0.001 | 0.000 |
| class_23 | – | – | 0.086 | 0.011 | 0.011 |
| class_24 | – | – | 0.112 | 0.128 | 0.116 |
| class_25 | – | – | – | 0.337 | 0.017 |
| class_26 | – | – | – | 0.269 | 0.003 |
| class_27 | – | – | – | 0.000 | 0.004 |
| class_28 | – | – | – | 0.013 | 0.055 |
| class_29 | – | – | – | – | 0.050 |
| class_30 | – | – | – | – | 0.311 |
| class_31 | – | – | – | – | 0.000 |
| class_32 | – | – | – | – | 0.000 |
| class_33 | – | – | – | – | 0.169 |
| class_34 | – | – | – | – | 0.074 |
| **Average** | 0.596 | 0.319 | 0.275 | 0.254 | 0.211 |

Table 31: Per-class performance of FoSSIL across incremental Sessions on Med Semi-Supervised-FoSSIL benchmark.

| Class | Session 0 | Session 1 | Session 2 | Session 3 | Session 4 | Session 5 |
|---|---|---|---|---|---|---|
| class_0 | 0.985 | 0.969 | 0.842 | 0.763 | 0.936 | 0.709 |
| class_1 | 0.881 | 0.659 | 0.628 | 0.709 | 0.019 | 0.779 |
| class_2 | 0.575 | 0.250 | 0.478 | 0.540 | 0.001 | 0.575 |
| class_3 | 0.852 | 0.883 | 0.886 | 0.874 | 0.025 | 0.896 |
| class_4 | 0.770 | 0.720 | 0.711 | 0.712 | 0.022 | 0.735 |
| class_5 | 0.786 | 0.849 | 0.856 | 0.830 | 0.133 | 0.862 |
| class_6 | 0.733 | 0.715 | 0.713 | 0.715 | 0.081 | 0.695 |
| class_7 | 0.696 | 0.760 | 0.713 | 0.731 | 0.014 | 0.793 |
| class_8 | 0.665 | 0.710 | 0.762 | 0.782 | 0.151 | 0.771 |
| class_9 | 0.799 | 0.828 | 0.797 | 0.812 | 0.118 | 0.791 |
| class_10 | 0.280 | 0.572 | 0.517 | 0.549 | 0.002 | 0.447 |
| class_11 | 0.864 | 0.809 | 0.827 | 0.834 | 0.235 | 0.835 |
| class_12 | 0.697 | 0.760 | 0.748 | 0.729 | 0.119 | 0.828 |
| class_13 | 0.776 | 0.641 | 0.653 | 0.695 | 0.054 | 0.757 |
| class_14 | 0.761 | 0.720 | 0.749 | 0.739 | 0.086 | 0.797 |
| class_15 | 0.769 | 0.712 | 0.671 | 0.511 | 0.026 | 0.804 |
| class_16 | – | 0.178 | 0.172 | 0.171 | 0.051 | 0.165 |
| class_17 | – | 0.060 | 0.087 | 0.112 | 0.008 | 0.105 |
| class_18 | – | 0.109 | 0.089 | 0.077 | 0.004 | 0.106 |
| class_19 | – | 0.074 | 0.048 | 0.068 | 0.000 | 0.042 |
| class_20 | – | 0.074 | 0.112 | 0.104 | 0.007 | 0.092 |
| class_21 | – | – | 0.049 | 0.072 | 0.004 | 0.056 |
| class_22 | – | – | 0.165 | 0.079 | 0.026 | 0.138 |
| class_23 | – | – | 0.091 | 0.060 | 0.001 | 0.084 |
| class_24 | – | – | 0.039 | 0.002 | 0.000 | 0.010 |
| class_25 | – | – | 0.032 | 0.019 | 0.011 | 0.023 |
| class_26 | – | – | 0.014 | 0.023 | 0.000 | 0.054 |
| class_27 | – | – | – | 0.182 | 0.001 | 0.190 |
| class_28 | – | – | – | 0.026 | 0.000 | 0.050 |
| class_29 | – | – | – | 0.366 | 0.001 | 0.168 |
| class_30 | – | – | – | 0.298 | 0.052 | 0.099 |
| class_31 | – | – | – | – | 0.252 | 0.106 |
| class_32 | – | – | – | – | 0.171 | 0.106 |
| class_33 | – | – | – | – | 0.227 | 0.044 |
| class_34 | – | – | – | – | – | 0.160 |
| class_35 | – | – | – | – | – | 0.105 |
| class_36 | – | – | – | – | – | 0.201 |
| class_37 | – | – | – | – | – | 0.286 |
| **Average** | 0.736 | 0.554 | 0.449 | 0.414 | 0.058 | 0.374 |

Table 32: Performance of SAM and FoSSIL on Natural-FoSSIL benchmark.

| Class | Base | SAM Session 1 | FoSSIL Session 1 | SAM Session 2 | FoSSIL Session 2 |
|---|---|---|---|---|---|
| 1 | 95.29 | 86.54 | 88.42 | 88.18 | 88.72 |
| 2 | 64.01 | 35.13 | 34.07 | 40.03 | 39.89 |
| 3 | 92.77 | 89.24 | 89.12 | 82.23 | 84.21 |
| 4 | 46.02 | 21.70 | 24.45 | 35.56 | 39.80 |
| 5 | 57.90 | 16.48 | 17.70 | 42.10 | 40.99 |
| 6 | 48.19 | 16.31 | 18.14 | 17.24 | 15.33 |
| 7 | 53.94 | 33.65 | 37.09 | 39.76 | 37.75 |
| 8 | 60.06 | 28.26 | 39.18 | 56.20 | 57.56 |
| 9 | 90.56 | 87.93 | 87.03 | 84.44 | 80.50 |
| 10 | 51.58 | 45.65 | 38.13 | 35.77 | 37.38 |
| 11 | – | 16.96 | 16.23 | 2.17 | 2.13 |
| 12 | – | 1.67 | 1.23 | 0.03 | 0.02 |
| 13 | – | 3.01 | 3.48 | 0.19 | 0.21 |
| 14 | – | 1.98 | 0.53 | 0.00 | 0.01 |
| 15 | – | 4.42 | 3.72 | 1.01 | 0.98 |
| 16 | – | – | – | 0.65 | 1.13 |
| 17 | – | – | – | 28.94 | 35.42 |
| 18 | – | – | – | 0.00 | 0.01 |
| **Average** | 66.03 | 32.59 | 33.23 | 30.80 | 31.22 |

Table 33: Per-class performance of FoSSIL on Semi-Supervised Natural-FoSSIL benchmark.

| Class | Session 0 | Session 1 | Session 2 | Session 3 |
|---|---|---|---|---|
| road - 0 | 91.18 | 82.36 | 81.35 | 81.48 |
| sidewalk - 1 | 48.82 | 31.20 | 24.84 | 22.94 |
| building - 2 | 86.42 | 71.03 | 72.53 | 67.41 |
| wall - 3 | 20.84 | 1.95 | 1.39 | 1.14 |
| fence - 4 | 25.28 | 2.81 | 1.41 | 2.49 |
| traffic light - 5 | 38.34 | 10.82 | 0.53 | 1.23 |
| traffic sign - 6 | 31.79 | 4.12 | 0.18 | 0.06 |
| person - 7 | 28.33 | 0.59 | 0.81 | 0.08 |
| car - 8 | 81.19 | 56.30 | 42.89 | 41.92 |
| truck - 9 | 25.40 | 12.12 | 7.94 | 8.96 |
| parking - 10 | – | 5.20 | 7.69 | 6.23 |
| vegetation - 11 | – | 54.76 | 53.41 | 48.49 |
| rail track - 12 | – | – | 12.03 | 13.23 |
| sky - 13 | – | – | 79.62 | 73.67 |
| motorcycle - 14 | – | – | – | 17.53 |
| autorickshaw - 15 | – | – | – | 20.59 |
| bridge/tunnel - 16 | – | – | – | 24.58 |
| **Average** | 47.75 | 27.8 | 27.68 | 25.47 |

Table 34: FoSSIL performance on Med FoSSIL-Mixed benchmark without Guided noise injection. The results highlight the importance of the proposed mechanism.

| Class | Session 0 | Session 1 | Session 2 | Session 3 | Session 4 |
|---|---|---|---|---|---|
| class_0 | 0.971 | 0.746 | 0.810 | 0.792 | 0.804 |
| class_1 | 0.727 | 0.155 | 0.180 | 0.181 | 0.273 |
| class_2 | 0.738 | 0.103 | 0.120 | 0.119 | 0.194 |
| class_3 | 0.613 | 0.077 | 0.091 | 0.094 | 0.159 |
| class_4 | 0.373 | 0.038 | 0.047 | 0.050 | 0.094 |
| class_5 | 0.665 | 0.099 | 0.133 | 0.137 | 0.244 |
| class_6 | 0.573 | 0.116 | 0.147 | 0.158 | 0.250 |
| class_7 | 0.580 | 0.128 | 0.143 | 0.137 | 0.177 |
| class_8 | 0.859 | 0.133 | 0.162 | 0.170 | 0.289 |
| class_9 | 0.566 | 0.082 | 0.103 | 0.105 | 0.164 |
| class_10 | 0.701 | 0.122 | 0.149 | 0.147 | 0.204 |
| class_11 | – | 0.023 | 0.029 | 0.031 | 0.053 |
| class_12 | – | 0.016 | 0.025 | 0.026 | 0.092 |
| class_13 | – | 0.008 | 0.012 | 0.014 | 0.025 |
| class_14 | – | 0.013 | 0.018 | 0.019 | 0.026 |
| class_15 | – | 0.013 | 0.014 | 0.014 | 0.016 |
| class_16 | – | 0.096 | 0.082 | 0.080 | 0.091 |
| class_17 | – | 0.035 | 0.033 | 0.033 | 0.037 |
| class_18 | – | 0.007 | 0.006 | 0.006 | 0.008 |
| class_19 | – | – | 0.037 | 0.037 | 0.041 |
| class_20 | – | – | 0.009 | 0.010 | 0.015 |
| class_21 | – | – | 0.020 | 0.022 | 0.037 |
| class_22 | – | – | 0.004 | 0.004 | 0.006 |
| class_23 | – | – | 0.026 | 0.027 | 0.030 |
| class_24 | – | – | 0.027 | 0.026 | 0.042 |
| class_25 | – | – | – | 0.018 | 0.025 |
| class_26 | – | – | – | 0.049 | 0.055 |
| class_27 | – | – | – | 0.048 | 0.043 |
| class_28 | – | – | – | 0.009 | 0.014 |
| class_29 | – | – | – | – | 0.006 |
| class_30 | – | – | – | – | 0.022 |
| class_31 | – | – | – | – | 0.008 |
| class_32 | – | – | – | – | 0.016 |
| class_33 | – | – | – | – | 0.049 |
| class_34 | – | – | – | – | 0.009 |
| class_35 | – | – | – | – | 0.016 |
| **Average** | 0.623 | 0.070 | 0.067 | 0.063 | 0.080 |

Table 35: Class-wise performance of MedFormer Vanilla on Med FoSSIL-Mixed benchmark.

| Class | Session 0 | Session 1 | Session 2 | Session 3 | Session 4 |
|---|---|---|---|---|---|
| class_0 | 0.969 | 0.955 | 0.965 | 0.957 | 0.956 |
| class_1 | 0.639 | 0.000 | 0.000 | 0.000 | 0.000 |
| class_2 | 0.660 | 0.000 | 0.117 | 0.000 | 0.000 |
| class_3 | 0.704 | 0.000 | 0.053 | 0.000 | 0.000 |
| class_4 | 0.371 | 0.000 | 0.000 | 0.000 | 0.000 |
| class_5 | 0.596 | 0.000 | 0.000 | 0.000 | 0.000 |
| class_6 | 0.589 | 0.000 | 0.000 | 0.000 | 0.000 |
| class_7 | 0.565 | 0.000 | 0.232 | 0.000 | 0.000 |
| class_8 | 0.835 | 0.000 | 0.000 | 0.000 | 0.000 |
| class_9 | 0.454 | 0.000 | 0.000 | 0.000 | 0.000 |
| class_10 | 0.717 | 0.000 | 0.000 | 0.000 | 0.000 |
| class_11 | – | 0.404 | 0.000 | 0.000 | 0.000 |
| class_12 | – | 0.698 | 0.000 | 0.000 | 0.000 |
| class_13 | – | 0.468 | 0.000 | 0.000 | 0.000 |
| class_14 | – | 0.262 | 0.000 | 0.000 | 0.000 |
| class_15 | – | 0.236 | 0.000 | 0.000 | 0.000 |
| class_16 | – | 0.870 | 0.000 | 0.000 | 0.000 |
| class_17 | – | 0.532 | 0.000 | 0.000 | 0.000 |
| class_18 | – | 0.096 | 0.000 | 0.000 | 0.000 |
| class_19 | – | – | 0.530 | 0.000 | 0.000 |
| class_20 | – | – | 0.580 | 0.000 | 0.000 |
| class_21 | – | – | 0.587 | 0.000 | 0.000 |
| class_22 | – | – | 0.395 | 0.000 | 0.000 |
| class_23 | – | – | 0.473 | 0.000 | 0.000 |
| class_24 | – | – | 0.247 | 0.000 | 0.000 |
| class_25 | – | – | – | 0.720 | 0.000 |
| class_26 | – | – | – | 0.700 | 0.000 |
| class_27 | – | – | – | 0.030 | 0.000 |
| class_28 | – | – | – | 0.000 | 0.000 |
| class_29 | – | – | – | – | 0.277 |
| class_30 | – | – | – | – | 0.401 |
| class_31 | – | – | – | – | 0.586 |
| class_32 | – | – | – | – | 0.247 |
| class_33 | – | – | – | – | 0.279 |
| class_34 | – | – | – | – | 0.376 |
| class_35 | – | – | – | – | 0.153 |
| **Average** | 0.636 | 0.280 | 0.167 | 0.086 | 0.090 |

Table 36: FoSSIL performance on Semi-Supervised Natural-FoSSIL benchmark without Pseudo-label refinement.

| Class | Session 0 | Session 1 | Session 2 | Session 3 |
|---|---|---|---|---|
| road - 0 | 91.18 | 81.79 | 77.50 | 78.68 |
| sidewalk - 1 | 48.82 | 27.03 | 26.18 | 24.36 |
| building - 2 | 86.42 | 70.01 | 66.08 | 62.76 |
| wall - 3 | 20.84 | 3.13 | 0.75 | 0.78 |
| fence - 4 | 25.28 | 4.06 | 0.81 | 0.88 |
| traffic light - 5 | 38.34 | 6.35 | 1.31 | 4.36 |
| traffic sign - 6 | 31.79 | 2.64 | 0.45 | 0.45 |
| person - 7 | 28.33 | 0.63 | 0.02 | 0.03 |
| car - 8 | 81.19 | 49.48 | 43.93 | 42.89 |
| truck - 9 | 25.40 | 11.19 | 6.89 | 9.09 |
| parking - 10 | – | 4.16 | 4.29 | 8.90 |
| vegetation - 11 | – | 51.60 | 53.31 | 60.64 |
| rail track - 12 | – | – | 9.24 | 12.86 |
| sky - 13 | – | – | 73.77 | 78.93 |
| motorcycle - 14 | – | – | – | 14.33 |
| autorickshaw - 15 | – | – | – | 17.82 |
| bridge/tunnel - 16 | – | – | – | 15.08 |
| **Average** | 47.75 | 26.00 | 26.04 | 25.12 |

Table 37: Performance of GAPS on Natural-FoSSIL benchmark.

| Class | Session 0 | Session 1 | Session 2 |
|---|---|---|---|
| road - 0 | 91.18 | 81.01 | 66.32 |
| sidewalk - 1 | 48.82 | 29.91 | 5.24 |
| building - 2 | 86.42 | 77.24 | 62.31 |
| wall - 3 | 20.84 | 0.00 | 0.00 |
| fence - 4 | 25.28 | 3.93 | 0.04 |
| traffic light - 5 | 38.34 | 6.01 | 6.64 |
| traffic sign - 6 | 31.79 | 2.64 | 1.95 |
| person - 7 | 28.33 | 6.48 | 10.33 |
| car - 8 | 81.19 | 68.94 | 45.45 |
| truck - 9 | 25.40 | 1.97 | 0.01 |
| Bridge/tunnel - 10 | – | 7.48 | 8.55 |
| Parking - 11 | – | 22.11 | 20.01 |
| rail track - 12 | – | 14.03 | 6.20 |
| Autorickshaw - 13 | – | 16.80 | 8.02 |
| pole - 14 | – | 12.67 | 9.69 |
| Bus - 15 (from BDD) | – | – | 0.02 |
| Sky - 16 (from BDD) | – | – | 48.76 |
| bicycle - 17 (from BDD) | – | – | 0.81 |
| person - 7 (repeat from IDD) - 18 | – | – | – |
| car - 8 (repeat from IDD) - 19 | – | – | – |
| **Average** | 47.76 | 23.42 | 16.69 |

Table 38: Performance of C-FSCIL on Med FoSSIL-Disjoint benchmark.

| Class | Session 0 | Session 1 | Session 2 |
|-------|-----------|-----------|-----------|
| class_0 | 0.982 | 0.715 | 0.606 |
| class_1 | 0.779 | 0.438 | 0.552 |
| class_2 | 0.735 | 0.300 | 0.444 |
| class_3 | 0.877 | 0.725 | 0.563 |
| class_4 | 0.794 | 0.669 | 0.615 |
| class_5 | 0.802 | 0.810 | 0.821 |
| class_6 | 0.742 | 0.438 | 0.471 |
| class_7 | 0.866 | 0.519 | 0.439 |
| class_8 | 0.844 | 0.364 | 0.350 |
| class_9 | 0.869 | 0.476 | 0.521 |
| class_10 | 0.639 | 0.271 | 0.275 |
| class_11 | 0.854 | 0.492 | 0.300 |
| class_12 | 0.780 | 0.095 | 0.145 |
| class_13 | 0.780 | 0.288 | 0.326 |
| class_14 | 0.752 | 0.103 | 0.282 |
| class_15 | 0.688 | 0.139 | 0.247 |
| class_16 | – | 0.112 | 0.053 |
| class_17 | – | 0.067 | 0.068 |
| class_18 | – | 0.188 | 0.231 |
| class_19 | – | 0.088 | 0.067 |
| class_20 | – | 0.092 | 0.079 |
| class_21 | – | – | 0.410 |
| class_22 | – | – | 0.286 |
| class_23 | – | – | 0.054 |
| class_24 | – | – | 0.078 |
| class_25 | – | – | 0.010 |
| class_26 | – | – | 0.037 |
| **Average** | 0.787 | 0.334 | 0.253 |

Table 39: Performance of GAPS on Semi-Supervised Natural-FoSSIL benchmark.

| Class | Session 0 | Session 1 | Session 2 | Session 3 |
|-------|-----------|-----------|-----------|-----------|
| road - 0 | 91.18 | 79.10 | 69.08 | 67.17 |
| sidewalk - 1 | 48.82 | 21.52 | 3.48 | 1.20 |
| building - 2 | 86.42 | 60.16 | 46.16 | 42.60 |
| wall - 3 | 20.84 | 1.31 | 0.49 | 0.33 |
| fence - 4 | 25.28 | 0.00 | 0.00 | 0.09 |
| traffic light - 5 | 38.34 | 0.00 | 0.01 | 0.00 |
| traffic sign - 6 | 31.79 | 0.00 | 0.00 | 0.00 |
| person - 7 | 28.33 | 0.00 | 0.00 | 0.00 |
| car - 8 | 81.19 | 25.79 | 20.09 | 8.85 |
| truck - 9 | 25.40 | 8.55 | 5.09 | 0.55 |
| parking - 10 | – | 2.18 | 1.57 | 3.34 |
| vegetation - 11 | – | 38.12 | 31.43 | 25.91 |
| rail track - 12 | – | – | 5.04 | 5.68 |
| sky - 13 | – | – | 80.24 | 69.65 |
| motorcycle - 14 | – | – | – | 9.45 |
| autorickshaw - 15 | – | – | – | 9.87 |
| bridge/tunnel - 16 | – | – | – | 1.04 |
| **Average** | 47.76 | 19.73 | 18.76 | 14.45 |

Table 40: Performance of NNCSL on Med Semi-Supervised-FoSSIL benchmark.

| Class | Session 0 | Session 1 | Session 2 | Session 3 | Session 4 | Session 5 |
|-------|-----------|-----------|-----------|-----------|-----------|-----------|
| class_0 | 0.985 | 0.814 | 0.925 | 0.957 | 0.959 | 0.962 |
| class_1 | 0.831 | 0.068 | 0.024 | 0.015 | 0.011 | 0.009 |
| class_2 | 0.575 | 0.039 | 0.029 | 0.011 | 0.009 | 0.008 |
| class_3 | 0.802 | 0.038 | 0.022 | 0.011 | 0.010 | 0.006 |
| class_4 | 0.720 | 0.017 | 0.008 | 0.005 | 0.001 | 0.002 |
| class_5 | 0.786 | 0.034 | 0.012 | 0.017 | 0.014 | 0.010 |
| class_6 | 0.733 | 0.050 | 0.029 | 0.018 | 0.019 | 0.015 |
| class_7 | 0.695 | 0.005 | 0.008 | 0.005 | 0.004 | 0.005 |
| class_8 | 0.665 | 0.030 | 0.022 | 0.009 | 0.008 | 0.004 |
| class_9 | 0.799 | 0.023 | 0.008 | 0.005 | 0.003 | 0.008 |
| class_10 | 0.230 | 0.042 | 0.019 | 0.016 | 0.013 | 0.011 |
| class_11 | 0.814 | 0.016 | 0.013 | 0.004 | 0.004 | 0.002 |
| class_12 | 0.697 | 0.017 | 0.025 | 0.023 | 0.018 | 0.017 |
| class_13 | 0.726 | 0.013 | 0.007 | 0.008 | 0.003 | 0.003 |
| class_14 | 0.711 | 0.013 | 0.015 | 0.014 | 0.011 | 0.009 |
| class_15 | 0.719 | 0.020 | 0.018 | 0.015 | 0.012 | 0.010 |
| class_16 | – | 0.121 | 0.012 | 0.007 | 0.012 | 0.017 |
| class_17 | – | 0.071 | 0.022 | 0.012 | 0.016 | 0.024 |
| class_18 | – | 0.093 | 0.015 | 0.011 | 0.012 | 0.013 |
| class_19 | – | 0.189 | 0.012 | 0.008 | 0.007 | 0.005 |
| class_20 | – | 0.044 | 0.023 | 0.014 | 0.015 | 0.026 |
| class_21 | – | – | 0.023 | 0.003 | 0.002 | 0.002 |
| class_22 | – | – | 0.518 | 0.032 | 0.018 | 0.029 |
| class_23 | – | – | 0.276 | 0.021 | 0.019 | 0.018 |
| class_24 | – | – | 0.013 | 0.005 | 0.003 | 0.003 |
| class_25 | – | – | 0.019 | 0.008 | 0.008 | 0.012 |
| class_26 | – | – | 0.049 | 0.009 | 0.004 | 0.007 |
| class_27 | – | – | – | 0.166 | 0.011 | 0.005 |
| class_28 | – | – | – | 0.052 | 0.006 | 0.005 |
| class_29 | – | – | – | 0.276 | 0.016 | 0.014 |
| class_30 | – | – | – | 0.106 | 0.006 | 0.007 |
| class_31 | – | – | – | – | 0.023 | 0.016 |
| class_32 | – | – | – | – | 0.048 | 0.013 |
| class_33 | – | – | – | – | 0.002 | 0.003 |
| class_34 | – | – | – | – | – | 0.216 |
| class_35 | – | – | – | – | – | 0.413 |
| class_36 | – | – | – | – | – | 0.237 |
| class_37 | – | – | – | – | – | 0.289 |
| **Average** | 0.700 | 0.047 | 0.047 | 0.030 | 0.011 | 0.040 |

Table 41: Performance of HALO on Semi-Supervised Natural-FoSSIL.

| Class | Session 0 | Session 1 | Session 2 | Session 3 |
|---|---|---|---|---|
| road - 0 | 91.18 | 0.00 | 0.00 | 0.00 |
| sidewalk - 1 | 48.82 | 0.00 | 0.00 | 0.00 |
| building - 2 | 86.42 | 0.00 | 0.00 | 0.00 |
| wall - 3 | 20.84 | 0.00 | 0.00 | 0.00 |
| fence - 4 | 25.28 | 0.00 | 0.00 | 0.00 |
| traffic light - 5 | 38.34 | 0.00 | 0.00 | 0.00 |
| traffic sign - 6 | 31.79 | 0.00 | 0.00 | 0.00 |
| person - 7 | 28.33 | 0.00 | 0.00 | 0.00 |
| car - 8 | 81.19 | 0.00 | 0.00 | 0.00 |
| truck - 9 | 25.40 | 0.00 | 0.00 | 0.00 |
| parking - 10 | – | 0.55 | 0.00 | 0.00 |
| vegetation - 11 | – | 20.87 | 0.00 | 0.00 |
| rail track - 12 | – | – | 0.90 | 0.00 |
| sky - 13 | – | – | 27.32 | 0.00 |
| motorcycle - 14 | – | – | – | 1.15 |
| autorickshaw - 15 | – | – | – | 17.82 |
| bridge/tunnel - 16 | – | – | – | 2.60 |
| **Average** | 47.76 | 1.78 | 2.02 | 1.27 |

Table 42: Performance of MDIL (domain incremental) on Med FoSSIL-Disjoint benchmark.

| Class | Session 0 | Session 1 | Session 2 |
|---|---|---|---|
| class_0 | 0.982 | 0.972 | 0.967 |
| class_1 | 0.851 | 0.000 | 0.000 |
| class_2 | 0.790 | 0.000 | 0.000 |
| class_3 | 0.864 | 0.000 | 0.003 |
| class_4 | 0.785 | 0.000 | 0.000 |
| class_5 | 0.767 | 0.000 | 0.001 |
| class_6 | 0.733 | 0.000 | 0.000 |
| class_7 | 0.798 | 0.000 | 0.000 |
| class_8 | 0.787 | 0.000 | 0.002 |
| class_9 | 0.791 | 0.000 | 0.000 |
| class_10 | 0.475 | 0.000 | 0.001 |
| class_11 | 0.817 | 0.000 | 0.001 |
| class_12 | 0.825 | 0.000 | 0.000 |
| class_13 | 0.846 | 0.000 | 0.000 |
| class_14 | 0.784 | 0.000 | 0.001 |
| class_15 | 0.774 | 0.000 | 0.000 |
| class_16 | – | 0.374 | 0.000 |
| class_17 | – | 0.256 | 0.000 |
| class_18 | – | 0.599 | 0.000 |
| class_19 | – | 0.541 | 0.000 |
| class_20 | – | 0.522 | 0.001 |
| class_21 | – | – | 0.288 |
| class_22 | – | – | 0.618 |
| class_23 | – | – | 0.514 |
| class_24 | – | – | 0.451 |
| class_25 | – | – | 0.366 |
| class_26 | – | – | 0.298 |
| **Average** | 0.779 | 0.115 | 0.098 |

Table 43: Performance of Gen-Replay (generative replay with diffusion) on Med FoSSIL-Disjoint benchmark.

| Class | Session 0 | Session 1 | Session 2 |
|---|---|---|---|
| class_0 | 0.985 | 0.966 | 0.960 |
| class_1 | 0.831 | 0.008 | 0.018 |
| class_2 | 0.575 | 0.013 | 0.002 |
| class_3 | 0.802 | 0.002 | 0.003 |
| class_4 | 0.720 | 0.002 | 0.009 |
| class_5 | 0.786 | 0.006 | 0.023 |
| class_6 | 0.733 | 0.001 | 0.000 |
| class_7 | 0.696 | 0.000 | 0.000 |
| class_8 | 0.665 | 0.001 | 0.000 |
| class_9 | 0.799 | 0.017 | 0.001 |
| class_10 | 0.230 | 0.003 | 0.000 |
| class_11 | 0.814 | 0.000 | 0.000 |
| class_12 | 0.697 | 0.001 | 0.000 |
| class_13 | 0.726 | 0.000 | 0.000 |
| class_14 | 0.711 | 0.003 | 0.000 |
| class_15 | 0.719 | 0.002 | 0.001 |
| class_16 | – | 0.192 | 0.000 |
| class_17 | – | 0.070 | 0.000 |
| class_18 | – | 0.546 | 0.002 |
| class_19 | – | 0.334 | 0.024 |
| class_20 | – | 0.329 | 0.001 |
| class_21 | – | – | 0.507 |
| class_22 | – | – | 0.637 |
| class_23 | – | – | 0.411 |
| class_24 | – | – | 0.415 |
| class_25 | – | – | 0.166 |
| class_26 | – | – | 0.430 |
| **Average** | 0.700 | 0.076 | 0.102 |

Table 44: Performance of CLIP-driven method on Med FoSSIL-Mixed benchmark. The model is pre-trained on a large number of classes in the benchmark.

| Class | Session 0 | Session 1 | Session 2 | Session 3 | Session 4 |
|---|---|---|---|---|---|
| class_1 | 0.881 | 0.825 | 0.702 | 0.702 | 0.036 |
| class_2 | 0.844 | 0.460 | 0.368 | 0.368 | 0.000 |
| class_3 | 0.844 | 0.829 | 0.218 | 0.218 | 0.000 |
| class_4 | 0.699 | 0.598 | 0.174 | 0.174 | 0.000 |
| class_5 | 0.628 | 0.085 | 0.001 | 0.001 | 0.000 |
| class_6 | 0.273 | 0.009 | 0.001 | 0.001 | 0.000 |
| class_7 | 0.757 | 0.696 | 0.139 | 0.139 | 0.000 |
| class_8 | 0.850 | 0.013 | 0.000 | 0.000 | 0.000 |
| class_9 | 0.652 | 0.233 | 0.002 | 0.002 | 0.000 |
| class_10 | 0.740 | 0.726 | 0.000 | 0.000 | 0.000 |
| class_11 | – | 0.808 | 0.048 | 0.048 | 0.000 |
| class_12 | – | 0.834 | 0.000 | 0.000 | 0.014 |
| class_13 | – | 0.002 | 0.002 | 0.002 | 0.026 |
| class_14 | – | 0.013 | 0.000 | 0.000 | 0.000 |
| class_15 | – | 0.012 | 0.000 | 0.000 | 0.000 |
| class_16 | – | 0.946 | 0.941 | 0.941 | 0.241 |
| class_17 | – | 0.106 | 0.000 | 0.000 | 0.030 |
| class_18 | – | 0.314 | 0.261 | 0.261 | 0.006 |
| class_19 | – | – | 0.800 | 0.802 | 0.177 |
| class_20 | – | – | 0.840 | 0.842 | 0.000 |
| class_21 | – | – | 0.855 | 0.855 | 0.180 |
| class_22 | – | – | 0.000 | 0.000 | 0.000 |
| class_23 | – | – | 0.000 | 0.000 | 0.000 |
| class_24 | – | – | 0.098 | 0.098 | 0.000 |
| class_25 | – | – | – | 0.020 | 0.137 |
| class_26 | – | – | – | 0.000 | 0.000 |
| class_27 | – | – | – | 0.010 | 0.000 |
| class_28 | – | – | – | 0.009 | 0.001 |
| class_29 | – | – | – | – | 0.025 |
| class_30 | – | – | – | – | 0.148 |
| class_31 | – | – | – | – | 0.050 |
| class_32 | – | – | – | – | 0.430 |
| class_33 | – | – | – | – | 0.417 |
| class_34 | – | – | – | – | 0.729 |
| class_35 | – | – | – | – | 0.471 |
| **Average** | 0.716 | 0.417 | 0.227 | 0.196 | 0.089 |

