# OpenReview forum: "FoSSIL: A Unified Framework for Continual Semantic Segmentation in 2D and 3D Domains"
_ICLR.cc/2026/Conference — ICLR 2026 Conference Withdrawn Submission_

### Official Review · Reviewer_w3yv · 2025-10-30

**Soundness:** 3
**Presentation:** 2
**Contribution:** 2
**Rating:** 4
**Confidence:** 4

**Summary:**

This paper presents FOSSIL, a unified framework for continual semantic segmentation that simultaneously addresses class-incremental (CIL), domain-incremental (DIL), and few-shot challenges. The method introduces Guided Noise Injection (GNI) to mitigate overfitting in few-shot settings and Prototype-guided Pseudo-label Refinement (PLR) to effectively leverage unlabeled data. Experiments on a new, comprehensive 2D/3D benchmark show FOSSIL significantly outperforms existing methods.

**Strengths:**

1. The paper tackles a highly practical and challenging problem by unifying CIL, DIL, and few-shot learning. This problem formulation is novel and critical for real-world applications like autonomous driving and medical analysis.
2. The validation is extensive, testing against various methods on diverse backbones. The ablation studies in Figure 5 clearly demonstrate the necessity and contribution of both GNI and PLR.
3. The paper is well-written, with a clear problem definition , methodology, and analysis, making it easy to follow.

**Weaknesses:**

1. This principle of using gradient statistics to modulate network parameters seems not new. In my opinion, it is the core mechanic in adaptive optimizers (e.g., Adam, RMSProp), which use squared gradients to normalize learning rates. While GNI's application differs from optimization, I think the underlying concept is related. The paper needs to provide a discussion comparing GNI to other adaptive regularization schemes or existing noise injection methods (e.g., standard weight decay, dropout, or variational dropout) and justify why this specific gradient-based formulation is superior.
2. The paper lacks sensitivity analysis for the thresholds in PLR ($\tau_{conf}$, $\tau_{sim}$). It is unclear how these were "empirically determined" or how robust the model is to their variation.
3. GNI is only applied to the final classifier layer $F$. The paper needs to justify why it isn't applied to deeper feature extractor layers, which are also prone to overfitting in few-shot settings.

**Questions:**

I hope the authors can address the issues I raised in Weaknesses.

---

> ### Author Response · Authors · 2025-11-26
> **Response to reviewer w3yv**
>
> We thank the reviewer for the constructive feedback and for acknowledging the novelty, practical importance, clarity, and comprehensive empirical validation of our work, particularly the roles of GNI and PLR.
>
> ## **The paper needs to provide a discussion comparing GNI to other adaptive regularization schemes or existing noise injection methods (e.g., standard weight decay, dropout, or variational dropout) and justify why this specific gradient-based formulation is superior.**
>
> ---------------------------------------------------------------------
> 1. What is Guided Noise Injection (GNI)?
> ---------------------------------------------------------------------
>
> GNI introduces controlled stochastic perturbations into each weight $w_i$. For each parameter, we compute the squared gradient:
>
> $$
> G_i = \left( \frac{\partial L}{\partial w_i} \right)^2
> $$
>
> From this, we compute an inverse measure:
>
> $$
> Ginv_i = \frac{1}{G_i + \varepsilon}
> $$
>
> We then normalize these values so that every $\tilde{G}_i$ lies in the range (0, 1]:
>
> $$
> \tilde{G}_i = \frac{1 + Ginv_i - \min_j\, Ginv_j}{1 + \max_j\, Ginv_j - \min_j\, Ginv_j}
> $$
>
> Noise is then injected directly into the parameters:
>
> $$
> w_i \leftarrow w_i + \tilde{G}_i \cdot \xi_i
> \quad \text{with} \quad
> \xi_i \sim \mathcal{N}(0, 1)
> $$
>
> Key intuition:
>
> - Large gradients → parameter is important → $\tilde{G}_i$  is small → noise is suppressed. This preserves important parameters from being altered by noise.
> - Small gradients → parameter is less relevant → $\tilde{G}_i$ is large → noise encourages exploration and regularization.
>
> Thus, GNI provides anisotropic, geometry-aware noise that adapts to parameter sensitivity.
>
> ---------------------------------------------------------------------
> 2. Why GNI is NOT an Adaptive Optimizer like Adam or RMSProp
> ---------------------------------------------------------------------
>
> Adaptive optimizers such as Adam, RMSProp, and Adagrad rescale gradients using accumulated statistics but do not inject *structured perturbations* into the parameters. For example, Adam updates a parameter $w_t$ as
> $$
> w_{t+1} = w_t - \eta \cdot \frac{g_t}{\sqrt{v_t + \varepsilon}}
> $$
> where $m_t$ and $v_t$ are exponential moving averages of the gradient and its square. The second moment term $v_t$ acts as a diagonal preconditioner that provides a coarse and history based approximation of local curvature. However, these methods do not perform explicit exploration of the loss surface and do not incorporate true second order information that requires evaluation in a neighborhood of the current parameters. Their behavior depends on accumulated gradient statistics, which can become stale when the data distribution changes, leading to slower or less reliable adaptation in nonstationary settings.
> GNI differs fundamentally by applying noise directly to the parameters and by basing its magnitude solely on the current sensitivity $Ginv_i$, enabling selective stability and plasticity even in few-shot incremental settings.
>
> What GNI does:
>
> - Adds structured noise directly to parameters.
>
> - Uses *instantaneous* gradient magnitude. GNI relies solely on the current gradient sensitivity, without accumulating or averaging gradient statistics over time.
>
> - Acts independently of the underlying optimizer (e.g., we use Adam + GNI).
>
> - Adaptive optimizers influence the optimization trajectory by scaling gradients, whereas GNI injects structured, gradient-sensitive perturbations directly into the parameters, thereby regularizing the evolving parameter distribution throughout training.
>
> - Empirically improves generalization and stability under domain shifts.
>
> FoSSIL uses Adam with GNI. The consistent improvements indicate that GNI provides benefits that adaptive optimization alone cannot offer. Adam’s updates are shaped by historical gradient statistics that can become misaligned under domain shift, whereas GNI uses instantaneous gradients whose sensitivities naturally suppress noise on important weights, enabling both stability and adaptation.
>
> ---------------------------------------------------------------------
> 3. Why GNI is superior to Adaptive Regularization (e.g., UCL [1])
> ---------------------------------------------------------------------
>
> Adaptive Regularization methods such as UCL maintain per-node uncertainty values σₜ (variances). These uncertainties:
>
> - Depend on the distribution of past tasks.
> - May become stale under domain shift.
> - Require storing node-wise variances (a modest memory overhead).
> - Provide only partial anisotropy (UCL ties all incoming weights of a neuron to a single variance value, preventing the method from distinguishing the importance of individual weights connected to that neuron.).
>
> Few-shot data may provide insufficient informative gradients to correctly update UCL’s node-level variances, so the regularization remains dominated by old tasks and misguides learning on new few-shot classes.
>
>
> **(cont..)**

---

> ### Author Response · Authors · 2025-11-26
> **Response to reviewer w3yv**
>
> ## **The paper needs to provide a discussion comparing GNI to other adaptive regularization schemes or existing noise injection methods (e.g., standard weight decay, dropout, or variational dropout) and justify why this specific gradient-based formulation is superior.**
>
> What GNI provides instead:
>
> - Computes inverse-squared gradient values directly from current data.
> - Requires no memory of previous tasks.
> - Provides full parameter-level anisotropy (each weight has its own noise scale).
> - Adapts its parameter-wise noise scaling immediately to gradients arising from new domains or classes, allowing rapid adjustment under moderate domain shift.
> - Works effectively even when each session contains only a few samples per class.
>
> UCL-based methods may perform poorly under adaptation, whereas FoSSIL (using GNI) remains stable across incremental sessions with few-shot data.
>
> [1] Uncertainty-based Continual Learning with Adaptive Regularization, Ahn et. al., NeurIPS 2019.
>
> ---------------------------------------------------------------------
> 4. Why GNI outperforms Variational Dropout and other noise injection methods
> ---------------------------------------------------------------------
>
> Standard dropout and variational dropout inject uniform or learned but task-static (task-invariant) noise.
>
> These have key limitations in continual learning: Variational dropout learns a dropout rate $\alpha_i$ for each unit or weight, but this rate is fixed once learned and cannot adapt dynamically when weight importance shifts across tasks. Once a weight is pruned or heavily dropped, its effective capacity is difficult to recover, reducing plasticity. These methods do not use instantaneous curvature or gradient information to guide where noise should be injected. As a result, they struggle under domain shifts and few-shot regimes, where rapid, gradient-driven adaptation is crucial.
>
> Dropout-based methods inject noise independently of gradient geometry: they mask activations or weights based on uniform or learned per-unit noise rates. In contrast, GNI computes a parameter-specific noise scale directly from the current gradient magnitude, allowing instantaneous, curvature-sensitive adaptation at the level of individual parameters.
>
> GNI solves these issues as:
>
> - Noise is dynamic and changes every session based on the current gradient geometry.
> - Weights important for the current task have low noise (protected).
> - Weights unimportant for the current task get more noise (regularized).
> -  Noise scales adapt automatically across sessions as gradient sensitivities change.
> - Capacity is not permanently reduced, since noise levels can shrink when a weight becomes important - unlike dropout-based methods where suppressed weights often remain inactive.
>
> ---------------------------------------------------------------------
> 5. Why GNI is better than Standard Weight Decay (L2 Regularization)
> ---------------------------------------------------------------------
>
> Weight decay penalizes all parameters uniformly:
>
> $$
> \text{penalty} = \lambda \cdot \lVert W \rVert^2
> $$
>
> This has several limitations:
>
> • It does not account for gradient geometry or curvature.
> • It penalizes important and unimportant weights equally.
> • It may shrink parameters that are critical for previously learned classes.
> • It lacks directional selectivity.
>
> GNI instead aligns its noise with curvature:
>
> • Flat directions (small curvature) → large noise allowed → safe exploration.
> • Sharp directions (large curvature) → noise suppressed → protection.
>
> Weight decay cannot differentiate between these directions, but GNI can.
>
> ---------------------------------------------------------------------
> 6. Why the “Inverse-Squared Gradient” formulation is optimal
> ---------------------------------------------------------------------
>
> We emphasize that using 1 / (gradient²) is not an arbitrary design choice. It aligns with second-order geometry.
> A Taylor expansion of the perturbed loss shows that GNI places noise in directions where curvature is least harmful.
> (Appendix A.2 and A.3)
>
> To ensure numerical stability, 1 / (gradient²) combined with normalization keeps the noise bounded in (0, 1].
>
> GNI is not a variant of Adam, not a variant of uncertainty-based regularization, and not a variant of dropout. It provides a unique combination of properties:
>
> 1. Uses instantaneous gradient geometry to guide noise.
> 2. Produces fully anisotropic, per-parameter noise scales.
> 3. Protects critical parameters while regularizing non-critical ones.
> 4. Automatically adapts across tasks and domains through per-step gradient sensitivity.
> 5. Requires no stored statistics.
> 6. Works effectively in an extreme few-shot settings.
> 7. Aligns its perturbations with second-order curvature structure.
> 8. Admits PAC-Bayes–motivated bounds that are tighter under GNI's anisotropic noise than under isotropic perturbations,. (Appendix B)
>
> **(cont..)**

---

> ### Author Response · Authors · 2025-11-26
> **Response to reviewer w3yv**
>
> 9. Demonstrates consistent empirical gains across mutiple datasets and 2D/3D domains.
> 10. Complements, rather than replaces, standard optimizers like Adam.
>
> GNI therefore constitutes a principled, theoretically grounded, and empirically validated advance beyond existing regularization and optimization methods.
>
> Empirical validation:
>
> On Med FoSSIL-Disjoint benchmark:
> | Method                         | Session 0 | Session 1 | Session 2 |
> |--------------------------------|-----------|-----------|-----------|
> | Dropout / Weight Decay [2]     | 0.700     | 0.191     | –         |
> | Variational Dropout [3]        | 0.700     | 0.260     | –         |
> | Adagrad [4]                    | 0.700     | 0.024     | –         |
> | UCL                            | 0.700     | 0.430     | 0.325     |
> | FoSSIL                         | 0.736     | 0.460     | 0.398     |
>
> We ran UCL for more sessions to get clarity.
> As reported by [4], Adagrad performed the best out of the optimizers tested, followed by RMSprop and Adam so we used it.
> We thank the reviewer for motivating us to examine the strengths of FoSSIL more thoroughly and to more deeply compare it against other regularization-based methods.
>
> [2] Understanding the Role of Training Regimes in Continual Learning, Mirzadeh et. al., NeurIPS 2020.
>
> [3] Continual Variational Dropout: A View of Auxiliary Local Variables in Continual Learning, Hai et. al., Springer Machine Learning Journal, 2024.
>
> [4] Hard ASH: Sparsity and the right optimizer make a continual learner, Keskinen, ICLR TinyPaper 2024.
>
>
> ## **The paper lacks sensitivity analysis for the thresholds in PLR ($\tau_{conf}$ and $\tau_{sim}$). It is unclear how these were "empirically determined" or how robust the model is to their variation.**
>
>
> We analysed the pseudo labels selected by the model using the confidence
> threshold $\tau_{\text{conf}}$ and the similarity threshold $\tau_{\text{sim}}$.
> Retention (\%) denotes the proportion of pseudo labels that pass both filters.
> Our goal is to retain a set of pseudo labels that is both sufficiently large
> and highly reliable.
>
> A lower threshold such as $\tau_{\text{conf}} = 0.6$  or $\tau_{\text{sim}} = 0.6$  produced high
> retention but admitted many low-confidence pseudo labels, which would introduce
> noise into training. Very high confidence thresholds, on the other hand, were
> too strict and led to extremely low retention.
>
> The setting $\tau_{\text{conf}} = 0.7$ and $\tau_{\text{sim}} = 0.7$ provided
> the best trade-off: it preserved a reasonable number of pseudo labels while
> maintaining high reliability. We discarded the alternative
> $\tau_{\text{conf}} = 0.6$ and $\tau_{\text{sim}} = 0.7$ because it offered only
> a marginal 0.4% increase in retention relative to
> $\tau_{\text{conf}} = 0.7$ and $\tau_{\text{sim}} = 0.7$, but at the expense of
> lower confidence.
>
> Therefore, we selected $\tau_{\text{conf}} = 0.7$ and $\tau_{\text{sim}} = 0.7$
> as the most effective balance between pseudo-label quantity and quality.
>
>
> | **τ_conf** | **τ_sim** | **Retention (%)** | **Decision** |
> | ---------- | --------- | ----------------- | ------------ |
> | 0.6        | 0.6       | 71.7              | Rejected     |
> | 0.6        | 0.7       | 23.6              | Rejected     |
> | 0.6        | 0.8       | 1.8               | Rejected     |
> | 0.7        | 0.6       | 73.7              | Rejected     |
> | 0.7        | 0.7       | 23.2              | **Selected** |
> | 0.7        | 0.8       | 1.9               | Rejected     |
> | 0.8        | 0.6       | 72.6              | Rejected     |
> | 0.8        | 0.7       | 22.7              | Rejected     |
> | 0.8        | 0.8       | 1.8               | Rejected     |
>
> A similar setting is suggested by [5].
>
> [5] Dual Mean-Teacher: An Unbiased Semi-Supervised Framework for Audio-Visual Source Localization, Guo et. al., NeurIPS, 2023.
>
> We performed sensitivity analysis with $\tau_{\text{conf}} = 0.8$ and $\tau_{\text{sim}} = 0.7$ \& $\tau_{\text{conf}} = 0.7$ and $\tau_{\text{sim}} = 0.75$ and found that the model is robust to these variations.
>
> - $\tau_{\text{conf}} = 0.8$ and $\tau_{\text{sim}} = 0.7$ => Dice 0.562
>
> - $\tau_{\text{conf}} = 0.7$ and $\tau_{\text{sim}} = 0.75$ => Dice 0.567
>
> - $\tau_{\text{conf}} = 0.7$ and $\tau_{\text{sim}} = 0.7$ => Dice 0.554
>
> ## **GNI is only applied to the final classifier layer $\textit{F}$. The paper needs to justify why it isn't applied to deeper feature extractor layers, which are also prone to overfitting in few-shot settings.**
>
> We applied GNI to two layers—one in the deep feature extractor of the encoder and one in the decoder—when the model was fully unfrozen. We found that this configuration performed worse compared to applying GNI only at the last layer.
>
> - GNI at feature extractor layer in Encoder: IoU 22.67
> - GNI at feature extractor layer in Decoder: IoU 24.86
> - GNI at final classifier layer in Decoder: IoU 27.84
>
> **(cont...)**

---

> > ### Author Response · Authors · 2025-11-26
> > **Response to reviewer w3yv**
> >
> > GNI is intentionally applied only to the final classifier layer $\textit{F}$,
> > rather than to deeper feature extractor layers, for the following reasons.
> >
> > - Parameter sensitivity differs across the network. Early encoder-decoder layers encode generic features that are reused across all sessions, whereas the final classifier layer is highly task-specific and
> > receives the strongest gradient signals associated with class boundaries.
> > Injecting noise into deep layers would perturb *shared* low-level representations
> > and destabilize all previously learned classes, while injecting noise into
> > $\textit{F}$ regularizes only the task-specific decision boundaries where
> > overfitting is the highest.
> >
> > - Local curvature is highest at the classifier layer. In few-shot segmentation, most curvature and gradient magnitude variation
> > appears in the final logits, not in the deep feature extractor. GNI relies on per-parameter gradient magnitude to
> > allocate noise; applying it to layers with uniformly low curvature would add
> > noise where it provides minimal benefit but risks corrupting the global feature
> > space. Applying GNI to $\textit{F}$ concentrates perturbations exactly where
> > the loss landscape is sharpest and overfitting is most pronounced.
> >
> > - The feature extractor is shared across tasks; the classifier is
> > task-adaptive. Deeper layers must remain stable to preserve knowledge from previous sessions.
> > The classifier layer is the part of the model that adapts to new domains and
> > novel classes, and therefore benefits the most from geometry-aware noise.
> > Perturbing shared layers would amplify forgetting, whereas perturbing only
> > $\textit{F}$ provides regularization without harming the backbone.

---

### Official Review · Reviewer_y7bk · 2025-10-31

**Soundness:** 3
**Presentation:** 3
**Contribution:** 2
**Rating:** 4
**Confidence:** 4

**Summary:**

The paper proposes FoSSIL, a unified framework and benchmark for continual semantic segmentation across 2D natural scenes and 3D medical volumes. It targets realistic settings that combine class-incremental (CIL) and domain-incremental (DIL) shifts under few-shot supervision, optionally augmented with unlabeled data. The authors also release five challenging benchmarks (three 3D-medical, two 2D-driving) with multi-session protocols and scarce labels per session.

**Strengths:**

This paper has the following strengths:

- The paper ambitiously unifies several learning paradigms - continual, few-shot, and semi-supervised segmentation - under one benchmark suite.

- The proposed benchmarks (Med FoSSIL and Natural FoSSIL) cover realistic multi-session setups and will likely be valuable to the community.

- The evaluation is broad, including 25+ baselines, multiple backbones (U-Net, DeepLabv3+, SwinUNetr, SAM), and detailed ablations on proposed modules.

- The paper is generally well organized, with mathematical formulations that are easy to follow.

**Weaknesses:**

I have recognize these cons:

- The core mechanisms - prototype replay, noise-based regularization, and pseudo-label filtering - are adaptations of known techniques. The contribution is mainly in *integration* and *benchmarking*, not in introducing fundamentally new algorithms.

- The paper does not analyze *why* the guided noise injection helps beyond empirical performance. No theoretical link to stability–plasticity balance is made.

- Some baselines were not originally designed for few-shot or semi-supervised continual segmentation (e.g., MiB, MDIL), which could exaggerate FoSSIL’s relative advantage.

- Details of data splits, unlabeled data sampling, and hyperparameter tuning are missing. For a benchmark paper, this is a major weakness.

- The work lacks qualitative examples or discussions of cases where FoSSIL fails, such as severe domain shifts or noisy unlabeled data.

- The paper repeats motivation and design explanations across sections, and some figures (e.g., Figure 2–4) are not deeply analyzed.

**Questions:**

Some questions should be answered:

1. How is guided noise injection different from existing gradient-based regularization (e.g., SAM, weight perturbation)?

2. Are prototypes recomputed from all sessions or updated incrementally?

3. What is the additional computational overhead compared to vanilla training?

---

> ### Author Response · Authors · 2025-11-26
> **Response to reviewer y7bk**
>
> We thank the reviewer for the positive feedback and for recognizing the ambition of unifying continual, few-shot, and semi-supervised segmentation within a single benchmark suite, as well as the value of our different setups, the breadth of our evaluations, and the clarity of our mathematical formulations.
>
> ## **The contribution is mainly in integration and benchmarking, not in introducing fundamentally new algorithms.**
>
> Each component of FoSSIL introduces a unique and independently novel contribution.
>
> ### Prototype-based learning
> ​​Prior prototype-based methods [1,2,3] use prototypes purely for replay. In contrast, FoSSIL introduces dense class-wise prototypes that serve a dual, design-level role: (i) replaying old classes at the decoder's final layer, and (ii) acting as a feature-space validator to filter out over-confident but incorrect pseudo-labels when leveraging unlabeled data. *This dual use is not present in existing prototype approaches and is naturally integrated into FoSSIL’s training loop*.
>
> In few-shot incremental sessions, novel classes are extremely data-scarce. While unlabeled data can augment them, it also risks amplifying incorrect labels. **FoSSIL’s prototype-guided filtering ensures that only pseudo-labels consistent with the learned class prototypes are retained, directly improving performance rather than causing degradation. This constitutes a natural and logically unified design that goes beyond standard replay-only prototype usage.**
>
> This is also corroborated by the FoSSIL performance, where this novel prototype design improves upon FoSSIL.
>
> | Method                           | Session 0 | Session 1 | Session 2 |
> |----------------------------------|-----------|-----------|-----------|
> | YoooP [3]                           | 0.700     | 0.176     | 0.028     |
> | Saving100x [1]                      | 0.700     | 0.072     | 0.053     |
> | Adapt_replay [2]                    | 0.700     | 0.044     | 0.027     |
> | FoSSIL                           | 0.736     | 0.460     | 0.398     |
> | FoSSIL + prototype-guided filtering | 0.736     | 0.554     | 0.445     |
>
> [1] Saving 100x Storage: Prototype Replay for Reconstructing Training Sample Distribution in Class-Incremental Semantic Segmentation, Chen et. al., NeurIPS 2023.
>
> [2] Adaptive Prototype Replay for Class Incremental Semantic Segmentation, Zhu et. al., AAAI 2025.
>
> [3] YoooP: You Only Optimize One Prototype per Class for Non-Exemplar Incremental Learning, Kong et. al., TMLR 2025.
>
> ### Guided Noise Injection (GNI)
> Why GNI is fundamentally new (not a variant of an existing method):
>
> GNI introduces a combination of properties that no
> optimizer, adaptive regularizer, or noise-injection method combines.
> Its mechanism is not an incremental variation but a functionally new algorithm.
>
> 1. No prior method injects per-parameter noise scaled by the inverse of the squared instantaneous gradient. GNI perturbs each parameter as:
>   $$
>    \quad w_i ←   w_i +  \tilde{G_i} · \xi_i
> \quad \text{with}  \quad  \xi_i \sim \mathcal{N}(0, 1).
>   $$
>    $\tilde{G_i}$ is a normalized function of $1/(g_i^2 + \varepsilon)$ where $g_i$ is instantaneous gradient.
>
>    - Adaptive optimizers (Adam, RMSProp, Adagrad) modify the deterministic update magnitude using running statistics of the gradient. They do not inject stochastic perturbations, and they do not compute or use instantaneous curvature to scale noise or updates.
>    - Dropout, Gaussian noise, and variational dropout inject noise, but their noise levels do not depend on the instantaneous gradient magnitude.
>
>    - Uncertainty-based continual learning methods (UCL [4]) rely on stored past
>      uncertainties rather than instantaneous gradient curvature.
>
>   **To the best of our knowledge, a noise-injection rule explicitly tied to per-step gradient curvature does
>    not exist in prior literature to handle multiple constraints.**
>
> 2. No prior method provides per-parameter anisotropic noise placement based on
>    instantaneous curvature.
>
>    Existing regularizers:
>    - are isotropic (same noise everywhere) – dropout, Gaussian noise, weight decay
>    - apply stochasticity or uncertainty at coarse granularity (unit, layer, or node level) – variational dropout, UCL; whereas GNI, to our knowledge, is the first method that assigns per-parameter noise based on the instantaneous gradient
>    - are historical / moving-average–based – Adam, RMSProp (these optimizers adapt update magnitudes using past gradient statistics but do not inject noise or use instantaneous curvature for noise placement)
>
>    GNI is the first method to:
>    - measure current sensitivity via $g_i^2$
>    - invert it into $1/(g_i^2 + \varepsilon)$
>    - normalize it to remain bounded
>    - inject noise asymmetrically only into low-curvature (flat) directions
>
>    **This produces fully anisotropic, gradient-guided perturbations.**
>
> **(cont...)**

---

> ### Author Response · Authors · 2025-11-26
> **Response to reviewer y7bk**
>
> ## **The contribution is mainly in integration and benchmarking, not in introducing fundamentally new algorithms.**
>
> 3. GNI provides a strong theoretical motivation underpinning its design.
>
>    GNI is backed by (Appendix A and B):
>    - second-order Taylor analysis showing reduced loss in high-curvature
>      directions
>    - a Hessian-aligned spectral interpretation
>
>     - PAC-Bayes bounds that are strictly tighter than those obtained with isotropic noise.
>
> 4. To the best of our knowledge, GNI is the first “gradient-guided stochastic perturbation” for domain adaptive/continual/few-shot learning for semantic segmentation.
>
> No prior method:
> - perturbs parameters based on instantaneous, per-step gradients,
> - injects per-parameter noise whose magnitude changes every iteration,
> - uses a bounded inverse-curvature rule,
> - adapts noise placement dynamically in each session without storing statistics from previous tasks.
>
>    **This is a new algorithmic contribution, not a recombination of known techniques.**
>
> GNI maintains stable performance over multiple incremental sessions, operates effectively with extremely few samples (K = 5), preserves important parameters without storing old-task statistics (as in existing regularizers), allows prototype replay to succeed even without exemplars, avoids catastrophic forgetting where dropout, variational dropout, Gaussian noise, and adaptive optimizers perform poorly in the reasltic setup of DIL, CIL and data-scarce semantic segmentation.
>
> Noise-based regularizers inject noise independently of gradient information. Standard dropout uses uniform masking. Variational dropout learns per-unit noise rates, but these rates do not depend on the instantaneous gradient magnitude. Because noise placement, magnitude, and anisotropy depend directly on the current gradient rather than on static or learned dropout rates, GNI cannot be reduced to dropout, variational dropout, Gaussian noise, or other existing noise-based regularization schemes. Please refer to "Response to reviewer w3yv" for empirical comparison with noise-based regularization methods.
>
> ### Pseudo-label refinement (PLR) \& unlabeled data
>
> Unlabeled data is inexpensive and easy to obtain, unlike pixel-level annotations
> that require substantial expert effort. Yet very few continual semantic
> segmentation methods exploit such readily available unlabeled data. These
> unlabeled samples can come from any aligned domain that is publicly accessible
> and free of cost. FoSSIL explicitly addresses this gap by leveraging unlabeled
> data and introducing a pseudo-label refinement (PLR) strategy designed for
> incremental segmentation.
>
> While pseudo-labeling is widely used, our approach is not a reimplementation of
> existing refinement techniques. Standard pseudo-label selection relies on either
> a single confidence threshold or simple consistency checks, which do not handle
> the high-volume, high-noise pseudo labels typical in few-shot incremental
> segmentation. In contrast, **FoSSIL introduces a dual-criterion filtering
> mechanism that uses both model confidence (τ_conf) and feature similarity
> (τ_sim) to class prototypes**. This pair of constraints is specifically designed
> for continual and few-shot segmentation, where confident but incorrect pseudo-labels often
> destabilize training across sessions.
>
> Recent semi-supervised pseudo-labeling methods (CSL [5]) collapse under domain shift or
> few-shot regimes due to confirmation bias. Our dual-threshold PLR is tuned to
> filter aggressively during early sessions and remain stable across multiple
> domain-incremental steps.
>
> | Method | Session 0 | Session 1 | Session 2 | Session 3 | Session 4 | Session 5 |
> |--------|-----------|-----------|-----------|-----------|-----------|-----------|
> | CSL    | 0.659     | 0.040     | 0.020     | 0.000     | 0.000     | 0.000     |
> | FoSSIL | 0.640     | 0.431     | 0.368     | 0.335     | 0.323     | 0.293     |
>
> Prototypes also play a unique role by regulating how GNI behaves: only
> semantically consistent pseudo labels that pass PLR produce strong and reliable
> gradients, which automatically reduce GNI noise on important parameters. Any
> borderline pseudo-labels that still pass filtering produce weak gradients, which
> receive higher noise. This results in a natural, self-correcting pipeline—
> prototype-based filtering → strong/weak gradients → gradient-guided noise
> scaling—that does not exist in prior continual or semi-supervised methods.
>
> To the best of our knowledge, no existing pseudo-label or regularization method
> exhibits this behaviour: (i) **jointly filtering pseudo-labels using both
> confidence and prototype similarity**, (ii) stabilizing pseudo-label quality
> across successive incremental sessions, and (iii) generating gradient signals
> that interact naturally with GNI to reduce noise on reliable parameters while
> increasing noise on uncertain ones.
>
> **(cont...)**

---

> ### Author Response · Authors · 2025-11-26
> **Response to reviewer y7bk**
>
> **FoSSIL provides all these capabilities with negligible overhead and consistently strong performance across 37+ baselines and 12 public datasets (post-rebuttal), marking a substantial advancement in continual semantic segmentation.**
>
> [4] Uncertainty-based Continual Learning with Adaptive Regularization, Ahn et. al., NeurIPS 2019.
>
> [5] When Confidence Fails: Revisiting Pseudo-Label Selection in Semi-supervised Semantic Segmentation, Liu et. al. ICCV 2025.
>
> ## **The paper does not analyze why the guided noise injection helps beyond empirical performance. No theoretical link to stability–plasticity balance is made.**
>
> We provide a theoretical analysis in the paper (Section 3.5, Appendix A and B).
>
> Guided Noise Injection (GNI) establishes a principled stability–plasticity
> mechanism because every parameter $w_i$ receives a noise scale $\tilde G_i$ that
> is inversely proportional to its instantaneous squared gradient $g_i^2$. The
> GNI update is $\widetilde w_i = w_i + \tilde G_i \xi_i$ with $\xi_i \sim
> \mathcal{N}(0,1)$, where $\tilde G_i$ is a normalized function of
> $1/(g_i^2 + \varepsilon)$.
>
> Stability follows because parameters with large gradients (i.e., important or
> high-sensitivity weights) automatically receive small noise: if $|g_i|$ is
> large, then $1/(g_i^2 + \varepsilon)$ is small, and the normalization forces
> $\tilde G_i$ close to zero. Thus these parameters are minimally perturbed and
> remain stable across sessions.
>
> Plasticity follows because parameters with small gradients receive larger noise.
> If $g_i \approx 0$, then $1/(g_i^2 + \varepsilon)$ is large and the normalized
> $\tilde G_i$ approaches one, allowing stochastic exploration along directions
> that are currently flat or unused. This encourages repurposing unimportant
> weights when new classes or domains appear.
>
> The expected perturbed loss (from a second-order Taylor expansion) contains the
> term $\sum_i H_{ii} \tilde G_i^2$, where $H_{ii}$ are diagonal entries of the
> Hessian. Since $\tilde G_i$ is small exactly where $H_{ii}$ is typically large,
> GNI suppresses perturbations in high-curvature directions and allows larger
> perturbations in low-curvature directions. This alignment with curvature
> formally produces stability on sharp directions and plasticity on flat ones.
>
> From a Bayesian view, the perturbation defines a diagonal Gaussian posterior
> $q = \mathcal{N}(W, \mathrm{diag}(\tilde G_i^2))$. Compared to any isotropic
> posterior with matched total variance, the KL divergence $\mathrm{KL}(q\|p)$
> against an isotropic Gaussian prior is smaller due to the convexity of
> $x - \log x - 1$, giving GNI a strictly tighter PAC–Bayes generalization bound.
> This provides a formal guarantee that GNI regularizes in a way that preserves
> critical parameters while permitting adaptation in low-sensitivity directions.
>
> Because $\tilde G$ depends only on instantaneous gradients (not accumulated
> statistics), GNI adapts immediately after domain or task shifts without
> requiring stored Fisher matrices or task-specific uncertainties. This avoids
> the staleness problem in methods such as UCL and provides a clean,
> stateless mechanism for stability–plasticity control.
>
> In summary: (1) high gradients $\Rightarrow$ small $\tilde G_i$ $\Rightarrow$
> stability; (2) low gradients $\Rightarrow$ large $\tilde G_i$ $\Rightarrow$
> plasticity; (3) curvature-aligned expected loss ensures selective protection;
> (4) PAC–Bayes analysis confirms a tighter generalization bound than isotropic
> noise. Together, these properties give GNI a theoretically grounded and
> parameter-wise stability–plasticity mechanism.
>
> ## **Some baselines were not originally designed for few-shot or semi-supervised continual segmentation (e.g., MiB, MDIL), which could exaggerate FoSSIL’s relative advantage.**
>
> In Table 6, we compared with following methods:
>
> - MDIL - CIL, DIL
> - MiB - CIL
> - UaD-CE - CIL, Few-shot, Semi-supervised
> - NNCSL - CIL, Semi-supervised
> - HALO - Active Learning, domain shift
> - RETRIEVE - Semi-supervised
> - GAPS - Few-shot, CIL
>
> We compare against a broad mix of approaches because no existing baseline jointly addresses CIL, DIL, few-shot, and semi-supervised learning—FoSSIL is the first to do so. Similarly, we wanted to evaluate how well standard class-incremental and domain-incremental learning approaches perform in a strict few-shot setting, and to understand how far they can go without using any unlabeled data, especially when compared to methods that can leverage unlabeled data. Nevertheless, FoSSIL outperforms few-shot and semi-supervised approaches as well.
> To ensure fairness and broader coverage, we compared FoSSIL against 37+ methods spanning different combinations of these settings, and FoSSIL outperforms all of them. Please see Appendix I (IMPLEMENTATION DETAILS) for all baselines.

---

> ### Author Response · Authors · 2025-11-27
> **Response to reviewer y7bk**
>
> ## **Details of data splits, unlabeled data sampling, and hyperparameter tuning are missing.**
>
> We have provided detailed information for all datasets (3D medical, autonomous driving, and 2D surgical tasks) under both the few-shot and the few-shot semi-supervised settings, mentioning data splits, unlabeled data in the Appendix G (FOSSIL BENCHMARKS).
>
> For hyperparameter tuning please see "Response to reviewer w3yv" for $\tau_{conf}$ and $\tau_{sim}$ and "Response to reviewer WWNg" for GNI hyperparameters (though they are minimal).
>
> ## **The work lacks qualitative examples or discussions of cases where FoSSIL fails, such as severe domain shifts or noisy unlabeled data.**
>
> Thank you for pointing this out — it will help us further explore the strengths of the FoSSIL framework. We have also analyzed the following cases where FoSSIL performs sub-optimally:
>
> ### 1. Severe domain shift — In Table 3 and its semi-supervised variant (Table 7), we observe that at Session 4, the Dice score with a U-Net backbone drops to nearly zero, clearly highlighting the challenge posed by extreme domain shift.
>
> What exactly is happening in this scenario?
>
> - Session 0 - TotalSegmentator (Modality - **CT**)
> - Session 1 - AMOS (Modality - dominantly **CT**)
> - Session 2 - BCV (Modality - **CT**)
> - Session 3 - MOTS (Modality - **CT**)
> - Session 4 - BraTS (Modality - **MRI**)
> - Session 5 - Verse (Modality - **CT**)
>
> The model encountered predominantly CT data across earlier sessions, but when it was suddenly exposed to MRI data from BraTS, lightweight encoder–decoder architectures such as **U-Net catastrophically forgot the CT domain during MRI training**, causing performance to drop to nearly zero. Although FoSSIL mitigates this to some extent, it only increases the Dice score slightly—from 0.025 (Table 3) to 0.0576 (Table 7) for U-Net.
> **This underscores both the severity and real-world relevance of the FoSSIL benchmarks, while also revealing the limitations of simple models such as U-Net.**
>
> Remedy:
>
> - Transformers: Medformer adapts to the shift and retains a decent score of 0.323 as shown in Table 7 with FoSSIL. We suggest using transformers for severe domain shifts to avoid catastrophically forgetting.
>
> - More unlabeled data - as it's easy to acquire unlabeled data, we experimented with more unlabeled data from publicly available CT domains like TotalSegmentator, AMOS, BCV, MOTS, using a U-Net backbone.
> We made the following observations:
>
> | Model (U-Net backbone)                        | Session 4 |
> |------------------------------|-----------|
> | FoSSIL (no unlabeled data)   | 0.025     |
> | FoSSIL (15 unlabeled samples)| 0.058     |
> | FoSSIL (42 unlabeled samples)      | 0.197     |
>
> **This highlights the strength of the novel Prototype-guided pseudo-label refinement strategy for unlabeled data.**
>
> ### 2. Noisy unlabeled data — In real-world clinical settings, medical scans often exhibit noise, blur, and contrast degradation. To assess robustness under such conditions, we evaluate FoSSIL with both moderate and high noise intensities applied to the unlabeled data.
>
> We simulate realistic acquisition artifacts using a boundary-aware degradation module. First, Sobel edge detection is used to extract structural boundaries, which are converted into a soft boundary mask $B(x,y,z)\in[0,1]$ with high values along edges. Blur, contrast shifts, and noise perturbations are then applied only in boundary regions, while homogeneous areas remain mostly unaffected. This degradation is applied to 70% of the unlabeled images to model natural heterogeneity in real-world scans.
>
> We follow the following noise schedule to vary blur, contrast and additive gaussian noise in unlabeled images:
> - moderate
> - high
>
> | Operation | Randomness         | Moderate      | Heavy        |
> |-----------|-----------------------------------|-----------------------|----------------------|
> | Blur      | Uniform         | σ ∈ [0.5, 1.5]        | σ ∈ [1.0, 2.5]       |
> | Contrast  | Uniform         | 1.0 ± 0.15            | 1.0 ± 0.30           |
> | Additive  | Gaussian (Pixel Intensity)        | σ = 0.02              | σ = 0.05             |
>
> Following performances are observed on Med Semi-Supervised-FoSSIL benchmark:
>
> | Method                     | Session 1 |
> |----------------------------|-----------|
> | FoSSIL (no noise)          | 0.554     |
> | FoSSIL (moderate noise)    | 0.477     |
> | FoSSIL (heavy noise)       | 0.475     |
> | UaD-CE (best baseline/no noise)     | 0.082     |
>
> FoSSIL is fairly able to handle different noise intensities in the unlabeled data, with heavy noise reducing performance by only 0.002 points relative to the moderate-noise setting, **indicating strong robustness to large degradations in unlabeled images**. Nevertheless, enhancing robustness to severe noise in unlabeled continual semantic segmentation remains an important direction for future work.
>
> **(cont...)**

---

> ### Author Response · Authors · 2025-11-27
> **Response to reviewer y7bk**
>
> We thank the reviewer for motivating us to examine the strengths of FoSSIL more thoroughly, particularly in settings involving severe domain shift and robustness.
>
> ## **The paper repeats motivation and design explanations across sections, and some figures (e.g., Figure 2–4) are not deeply analyzed.**
>
> Thanks for pointing it out. We have corrected it in the paper.
>
> - Figure 2 (now Figure 1 in the paper after rebuttal) shows that the common Session 0 base model reaches its highest performance when fine-tuned with no constraints, but loses substantial accuracy when trained to learn new classes (CI), uses very few labeled images (FS) to fine-tune the base model, or adapts to a new domain (DI). These settings stress the model in different ways. **When new classes arrive, the classifier must shift its decision boundaries, disrupting feature directions learned for earlier classes.** **Under a few-shot condition, gradients become unreliable, and the model overfits the small labeled set. Under domain shift, feature distributions change, causing earlier class representations to misalign with new inputs.** The large drops relative to the unconstrained baseline indicate that the model’s feature space is being reorganized in ways that destabilize previously learned representations, explaining why naive training fails with these constraints.
>
>
> - Figure 3 (now Figure 2 in the paper post-rebuttal) shows that naive fine-tuning of the base model in Session 0 is unstable across all backbones. When most layers are frozen in Session 1, the model has limited capacity to learn new features, restricting its ability to learn new classes or adapt to a new domain. However, when all layers are unfrozen in Session 1, updates propagate into the early feature extractor, shifts in the representation space, and disrupt the structures learned for earlier classes. The drop in performance across different backbones **highlights that neither freezing nor fully unfreezing provides a stable solution, and that incremental training requires mechanisms that preserve earlier representations while still allowing controlled adaptation to new data.**
>
> - Figure 4a shows how performance evolves across epochs within an incremental session where the model encounters new classes and a new domain under few-shot supervision. *Both Vanilla SwinUNetr and FoSSIL-SwinUNetr start Session 1 from the same Vanilla base model trained in Session 0*, ensuring identical initialization. At epoch 0, both perform poorly because the new classes were unseen in the previous base session. As the training proceeds, their performances diverge. The Vanilla model shows minimal improvement, indicating that few-shot gradients are too weak and unstable to support reliable adaptation and disrupt previously learned features. In contrast, **FoSSIL adapts effectively to the new classes, new domain, and limited labels. Its consistently higher and more stable trajectory in Figure 4a demonstrates controlled and reliable learning during the incremental session, while the Vanilla model remains unstable and fails to improve.**
>
> - Figure 4b compares FoSSIL’s performance in the same incremental session with (SS FoSSIL-U-Net) and without (FoSSIL-U-Net) access to unlabeled data. In this session, the model must learn new classes or a new domain from only a few labeled examples, so incorporating unlabeled samples can significantly expand the available supervision, provided they are used reliably. The figure shows that using unlabeled data yields a clear performance improvement. **By adding unlabeled examples, the model gains broader coverage of the new session’s data distribution, reducing overfitting to the small labeled set and stabilizing the new representations**. The improvement shown in Figure 4b demonstrates that semi-supervised incremental learning (SS FoSSIL-U-Net) is effective when pseudo-labels are selectively accepted, and **highlights the role of FoSSIL’s filtering mechanism in maintaining stability while increasing useful supervision.**
>
> ## **How is guided noise injection different from existing gradient-based regularization (e.g., SAM, weight perturbation)?**
>
> Earlier, we detailed how GNI is fundamentally different from and superior to existing noise-based regularization methods, weight perturbation techniques, and other commonly used regularization schemes.
> Now we will analyze two **recent** approaches on Sharpness-aware minimization (SAM) and weight perturbation.
>
> Classical SAM [1] (Sharpness-Aware Minimization)
> performs a deterministic min-max optimization step that perturbs the parameters
> within a fixed-radius neighborhood and then updates them using the resulting
> worst-case gradient. This produces a perturbation defined over a fixed-radius neighborhood of the full parameter vector, which does not provide parameter-wise differentiation.
>
> **(cont....)**

---

> ### Author Response · Authors · 2025-11-27
> **Response to reviewer y7bk**
>
> As a result, SAM alters the optimization trajectory, but provides no mechanism to identify which weights
> are important for previous tasks or which directions should be preserved for subsequent ones.
>
> C-Flat extends SAM to continual learning by optimizing a surrogate objective that
> couples neighborhood perturbations with Hessian-vector directional curvature
> terms. This encourages the model to remain in locally smooth regions of the loss
> landscape and reduces sensitivity to task specific sharp minima. However, because its flatness and curvature terms operate on neighborhoods of the full parameter vector rather than individual parameters, C-Flat does not provide weight-level control over which directions should remain stable and which should remain adaptable. Consequently, C-Flat enforces a uniform notion of flatness across the
> network rather than assigning stability or plasticity based on parameter
> importance. In addition, its neighborhood radius and curvature coefficients
> introduce multiple interacting hyperparameters, increasing optimization
> complexity and limiting interpretability.
>
> GNI is built on a fundamentally different design principle. Instead of imposing
> global stability constraints, it injects parameter wise adaptive noise derived
> directly from the instantaneous gradient. For each weight $w_i$, the
> corresponding gradient $g_i$ determines a noise scale $\tilde{G}_i$
> which provides an automatic, data driven stability plasticity trade off at the
> parameter level. Weights with large gradients, which actively contribute to
> current task performance, receive negligible noise, preserving critical
> information. Weights with small gradients receive larger perturbations, enabling
> rapid adaptation without interfering with previously consolidated structure. This
> yields a local, lightweight, and interpretable mechanism for continual learning
> that avoids global curvature estimation and removes the need for complex
> objective balancing. Unlike C-Flat's uniform landscape smoothing, GNI
> selectively preserves what is important and selectively explores what is
> flexible, using a simple rule rooted in gradient geometry.
>
> For weight perturbation, we have opted for the recent method STAR [3], suitable for continual learning.
>
> The STAR method stabilizes learning by introducing an adversarial-style perturbation
> $\Delta w$ that amplifies model sensitivity on stored buffer samples and then
> penalizing the resulting output drift through a KL divergence term,
> $$
> L_{\text{STAR}} = \mathrm{KL}\big(f(x; w) | | f(x; w + \Delta w)\big).
> $$
> The perturbation is obtained by initializing each layer with small noise to avoid zero gradients, then performing one gradient-ascent step on the KL divergence at the perturbed point, and finally normalizing the gradient per layer to control the perturbation magnitude.
> This layer-wise normalization controls the perturbation scale at the layer granularity, rather than at the level of individual parameters. STAR also relies on stored buffer samples and requires additional forward
> and backward passes to compute both the perturbed outputs and the ascent direction,
> introducing nontrivial memory and computational overhead.
>
> In contrast, GNI requires no buffer and no auxiliary training passes. Its stability
> mechanism arises directly from the instantaneous local gradient structure, enabling
> parameter-wise, curvature-sensitive regularization with minimal overhead.
>
> Empirical validation:
>
> | Method | Session 0 | Session 1 | Session 2 |
> |--------|-----------|-----------|-----------|
> | C-Flat [2] | 0.700     | 0.174     | 0.030     |
> | STAR [3]   | 0.700     | 0.050     | 0.020     |
> | FoSSIL | 0.736     | 0.460     | 0.398     |
>
> [1] Sharpness-Aware Minimization for Efficiently Improving Generalization, Foret, et. al., ICLR 2021.
>
> [2] C-Flat - Make Continual Learning Stronger via C-Flat, Bian et. al., NeurIPS 2024.
>
> [3] STAR: Stability Inducing Weight Perturbation for Continual Learning, Eskandar et. al., ICLR 2025.
>
> ## **Are prototypes recomputed from all sessions or updated incrementally?**
>
> No. We compute a single prototype (only a few KBs) for each class at the end of the session where the class first appears, and we reuse this same prototype in all subsequent incremental sessions to replay at the segmentation decoder’s final layer. This design ensures that even in long, continual-learning sequences with many sessions, classes forgotten long ago can be effectively relearned, recovering performance close to what was achieved in the original session where the class first appeared.

---

> ### Author Response · Authors · 2025-11-27
> **Response to reviewer y7bk**
>
> ## **What is the additional computational overhead compared to vanilla training?**
>
> ### Cost Analysis
> w/o FoSSIL => Vanilla
> | **Setting**                     | **Parameters**       |**Parameters**            | **FLOPs**          |  **FLOPs**           | **Training Time**   |  **Training Time**          |
> |---------------------------------|--------------------|-----------|------------------|-----------|------------------|-----------|
> |                                 | FoSSIL             | w/o FoSSIL | FoSSIL           | w/o FoSSIL | FoSSIL           | w/o FoSSIL |
> | **Med_FoSSIL-Disjoint**         | 16.27M             | 16.27M     | 0.52T            | 0.52T      | 4hrs 6mins       | 4hrs 5mins |
> | **Med_Semi-Supervised-FoSSIL** | 39.59M             | 39.59M     | 1.1T             | 1.1T       | 5hrs 18mins      | 5hrs 8mins |
> | **Natural-FoSSIL (SAM vit-b)**  | 88.9M              | 88.9M      | 0.37T            | 0.37T      | 1hrs 43mins      | 1hrs 35mins |
>
> We report the computational analysis for Session 1 in Table 9. **The results clearly show that FoSSIL does not incur any additional computational cost while significantly improving performance.**

---

### Official Review · Reviewer_WWNg · 2025-10-31

**Soundness:** 3
**Presentation:** 2
**Contribution:** 2
**Rating:** 4
**Confidence:** 4

**Summary:**

This paper tackles continual semantic segmentation in a very real-world way: data arrives in sessions, with a fully supervised base session and then a stream of few-shot increments that either introduce new classes, shift the domain, or both, while unlabeled data is abundant. The proposed FoSSIL framework keeps old knowledge without storing raw images by replaying compact class prototypes, steadies training with a gradient-guided noise injection scheme to curb few-shot overfitting, and makes semi-supervision actually work by filtering pseudo-labels using both confidence and prototype consistency in a mean-teacher setup.

**Strengths:**

1. The paper directly tackles continual semantic segmentation across CIL, DIL, and few-shot learning in one framework. FoSSIL addresses this limitation and leverages unlabeled data to augment scarce few-shot classes.
2. Extensive evidence across 3D medical and 2D autonomous driving benchmarks, with multi-session protocols, shows consistent gains and robustness over strong baselines.
3. FoSSIL integrates cleanly with diverse backbones, indicating strong architecture-agnostic generalization rather than narrow tuning.
4. This paper proposes a readily deployable solution to real-world deployment pain points—privacy, limited annotations, and domain shift.

**Weaknesses:**

1. The paper lacks a clear pipeline/architecture diagram, which would make the method easier to grasp at a glance.
2. The noise injection module has no analysis of hyperparameters or other strategies (e.g., sensitivity and robustness studies).
3. In several tables (Table 2, Table 4, Table 6, Table 7), multiple methods report identical Session-0 results; the authors should explain why.
4. In Table 3, FoSSIL (U-Net) drops to 0.025 at Session 4 and then rebounds to 0.324 at Session 5; this large fluctuation should be verified (typesetting/statistics) or clearly explained.
5. The paper should report computational costs, including training/inference time and memory/parameter overhead.
6. Although FoSSIL improves over multiple sessions, the absolute Dice/IoU remains low, which may limit practical applicability; this seems at odds with the paper’s motivation and should be discussed.
7. How are τ_conf and τ_sim selected/tuned? Are they shared across datasets? Please provide threshold sensitivity and retention/coverage statistics.

**Questions:**

Please refer to the points listed under Weaknesses; if the authors can satisfactorily address these concerns, I will consider raising my score.

---

> ### Author Response · Authors · 2025-11-28
> **Response to reviewer WWNg**
>
> We sincerely thank the reviewer for the thoughtful and encouraging assessment of our work. We are also grateful for the reviewer’s recognition of the extensive experimental evidence.  We also appreciate the reviewer noting FoSSIL’s architecture-agnostic generalization.  Finally, we thank the reviewer for acknowledging the practical relevance of our contributions.
>
> ## **The paper lacks a clear pipeline/architecture diagram, which would make the method easier to grasp at a glance.**
>
> Thank you for pointing this out. We have added it to the paper (Figure 3).
>
> ## **The noise injection module has no analysis of hyperparameters or other strategies (e.g., sensitivity and robustness studies).**
>
> GNI introduces *no additional design hyperparameters*. It follows, $\widetilde w_i = w_i + \tilde G_i \xi_i$ with $\xi_i \sim
> \mathcal{N}(0,1)$, where $\tilde G_i$ is a normalized function of
> $1/(g_i^2 + \varepsilon)$.
>
> $\varepsilon $ has been added for stability if $g_i$ goes to 0. For all practical purposes, we have set noise variance to 1 for $\xi_i $ and $\varepsilon$  to $1.0 \times 10^{-8}$.
>
> - vary $\varepsilon$
>
> We vary $\varepsilon$ by taking following values: $1.0 \times 10^{-6}$, $1.0 \times 10^{-7}$, $1.0 \times 10^{-9}$.
>
> | $\varepsilon$    | Dice  |
> |--------------|--------|
> | $1.0 \times 10^{-6}$ | 0.443 |
> | $1.0 \times 10^{-7}$ | 0.437 |
> | $1.0 \times 10^{-8}$ | 0.460 |
> | $1.0 \times 10^{-9}$ | 0.482 |
> |--------------|--------|
> | Vanilla model|  0.076 |
>
> We observe fairly robust results. Smaller epsilon (e.g., $1.0 \times 10^{-9}$) works somewhat better because it allows GNI to more clearly separate “important” parameters from “less important” ones. When epsilon is very small, the scaling term $1/(g_i^2 + \varepsilon)$ is dominated by the actual gradient value. This means parameters with large gradients receive very small noise (they are preserved), while parameters with tiny gradients receive larger noise (they stay flexible). When epsilon is slightly larger ($1.0 \times 10^{-6}$ or $1.0 \times 10^{-7}$), it can start to dominate the denominator when gradients are small, and many parameters may end up with almost the same noise scale regardless of their gradients. Importantly, GNI normalizes all scales to the range (0,1], so using a smaller $\varepsilon$ does not cause instability. It simply provides sharper and more useful contrast, which might be the reason why  $1.0 \times 10^{-9}$ gives somewhat better performance.
>
> - vary variance of noise $\xi_i$
>
> We consider two noise settings, with variances set to 0.1 and 10.
>
> | Noise Variance | Dice  |
> |----------------|-------|
> | 0.1       | 0.460 |
> | 1      | 0.460 |
> | 10     | 0.408 |
> |--------------|--------|
> | Vanilla model|  0.076 |
>
> FoSSIL is clearly robust to changes in variance of noise added (performance of variances 0.1 and 1 are the same), as the Vanilla model here is still at dice 0.076.
>
> As GNI regularizes the model when novel few-shot classes appear, we monitor performance across different shot values (K), corresponding to the number of labeled samples available for each class in that session.
>
> - vary K
>
> | K| Dice  |
> |----------------|-------|
> | 3      | 0.414 |
> | 4     | 0.434 |
> | 5   | 0.460 |
> |--------------|--------|
> | Vanilla model (K=5) |  0.076 |
>
> It is clear that under few shot-based stress testing, FoSSIL retains strong performance even at K = 3, with only three labeled samples per few-shot class.
>
> **This clearly demonstrates that FoSSIL remains robust despite relying on only a minimal set of hyperparameters.** Please note that $\varepsilon$, and K are not *typically* treated as hyperparameters.
>
> We also evaluated FoSSIL’s sensitivity to noise in the unlabeled data. Please refer to “Response to Reviewer y7bk” for details.
>
> ## **In several tables (Table 2, Table 4, Table 6, Table 7), multiple methods report identical Session-0 results; the authors should explain why.**
>
> Some methods apply their approach only from the incremental sessions onward, assuming that a pretrained base model is already available. In such cases, the base model is a heavily trained backbone (for example U-Net, DeepLab, Transformer or SAM) learned from a large number of labeled samples, and the novel few shot classes appear starting from Session 1. For these methods, the base session is shared and identical across approaches.
>
> In contrast, other methods apply their idea already during the base session. For example, Forward Compatible Few-Shot Class Incremental Learning (Zhou et al., CVPR 2022) (FACT in Table 2) reserves representation space for virtual novel classes during the base session itself. Since such methods modify the base session training, their performance can differ, sometimes slightly and sometimes substantially, depending on the nature of their design.

---

> ### Author Response · Authors · 2025-11-28
> **Response to reviewer WWNg**
>
> ## **In Table 3, FoSSIL (U-Net) drops to 0.025 at Session 4 and then rebounds to 0.324 at Session 5; this large fluctuation should be verified (typesetting/statistics) or clearly explained.**
>
> Thank you for pointing it out.
>
> In Table 3 and its semi-supervised variant (Table 7), we observe that at Session 4, the Dice score with a U-Net backbone drops to nearly zero, clearly highlighting the challenge posed by extreme domain shift.
>
> What exactly is happening in this scenario?
>
> - Session 0 - TotalSegmentator (Modality - **CT**)
> - Session 1 - AMOS (Modality - dominantly **CT**)
> - Session 2 - BCV (Modality - **CT**)
> - Session 3 - MOTS (Modality - **CT**)
> - Session 4 - BraTS (Modality - **MRI**)
> - Session 5 - Verse (Modality - **CT**)
>
> The model encountered predominantly CT data across earlier sessions, but when it was suddenly exposed to MRI data from BraTS, lightweight encoder–decoder architectures such as **U-Net catastrophically forgot the CT domain during MRI training**, causing performance to drop to nearly zero. Although FoSSIL mitigates this to some extent, it only increases the Dice score slightly—from 0.025 (Table 3) to 0.0576 (Table 7) for U-Net.
>
> **This underscores both the severity and real-world relevance of the FoSSIL benchmarks, while also revealing the limitations of simple models such as U-Net.**
>
> Remedy:
>
> - Transformers: Medformer adapts to the shift and retains a decent score of 0.323 as shown in Table 7 with FoSSIL. We suggest using transformers for severe domain shifts to avoid catastrophically forgetting.
>
> - More unlabeled data - as it's easy to acquire unlabeled data, we experimented with more unlabeled data from publicly available CT domains like TotalSegmentator, AMOS, BCV, MOTS, using a U-Net backbone.
> We made the following observations:
>
> | Model (U-Net backbone)                        | Session 4 |
> |------------------------------|-----------|
> | FoSSIL (no unlabeled data)   | 0.025     |
> | FoSSIL (15 unlabeled samples)| 0.058     |
> | FoSSIL (42 unlabeled samples)      | 0.197     |
>
> | Session 0 | Session 1 | Session 2 | Session 3 | Session 4 | Session 5 |
> |-----------|-----------|-----------|-----------|-----------|-----------|
> | 0.736     | 0.554     | 0.445     | 0.414     | 0.197**     | 0.346     |
>
>  ** more unlabeled data
>
> **This highlights the strength of the novel Prototype-guided pseudo-label refinement strategy for unlabeled data.**
>
> We thank the reviewer for raising this point, as it helped us highlight the strengths of FoSSIL.
>
> ## **The paper should report computational costs, including training/inference time and memory/parameter overhead.**
>
> ### Cost Analysis
> w/o FoSSIL => Vanilla
> | **Setting**                     | **Parameters**       |**Parameters**            | **FLOPs**          |  **FLOPs**           | **Training Time**   |  **Training Time**          |
> |---------------------------------|--------------------|-----------|------------------|-----------|------------------|-----------|
> |                                 | FoSSIL             | w/o FoSSIL | FoSSIL           | w/o FoSSIL | FoSSIL           | w/o FoSSIL |
> | **Med_FoSSIL-Disjoint**         | 16.27M             | 16.27M     | 0.52T            | 0.52T      | 4hrs 6mins       | 4hrs 5mins |
> | **Med_Semi-Supervised-FoSSIL** | 39.59M             | 39.59M     | 1.1T             | 1.1T       | 5hrs 18mins      | 5hrs 8mins |
> | **Natural-FoSSIL (SAM vit-b)**  | 88.9M              | 88.9M      | 0.37T            | 0.37T      | 1hrs 43mins      | 1hrs 35mins |
>
> We report the computational analysis for Session 1 in Table 9. **The results clearly show that FoSSIL does not incur any additional computational cost while significantly improving performance.**  FoSSIL framework is lightweight with minimal hyperparameters, broadly adaptable across backbones, and achieves strong results on about 12 datasets compared with around 37 baselines.
>
> ## **Although FoSSIL improves over multiple sessions, the absolute Dice/IoU remains low, which may limit practical applicability; this seems at odds with the paper’s motivation and should be discussed.**
>
> We individually analyze the performance of state-of-the-art methods across various applications, backbones, and experimental setups to better understand their absolute performance on different datasets. We also examine the unique challenges posed by the FoSSIL benchmarks. Finally, we outline the specific strengths of FoSSIL and present some approaches for its further improvement.
>
> ### 3D domain
>
> **A)** Segmentation performance in 3D medical domains depends on the target organ. Smaller organs are more difficult to segment, whereas larger organs are generally easier. The few-shot setting further increases this difficulty, as the model is expected to segment a new class, potentially a small organ, using only five labeled samples, and to do so in a continual setup with changing domains.
>
> **(cont....)**

---

> ### Author Response · Authors · 2025-11-28
> **Response to reviewer WWNg**
>
> ## **Although FoSSIL improves over multiple sessions, the absolute Dice/IoU remains low, which may limit practical applicability; this seems at odds with the paper’s motivation and should be discussed.**
>
> The incremental domains we evaluate—BCV, BraTS, and MOTS—contain organs and structures that vary significantly in size, making the setup particularly challenging in the few-shot regime. For instance, adrenal glands in BCV, colon tumors in MOTS, and the tumor subregions in BraTS are all known to be difficult to segment with limited supervision, as corroborated by the following papers:
>
> - Few-Shot Adaptation of Training-Free Foundation
> Model for 3D Medical Image Segmentation, He et. al., 2025. (Table III - Adrenal glands in BCV (or BTCV)) - BraTS classes). The dice score is around 0.4.
>
> - CCQ: Cross-Class Query Network for Partially Labeled Organ Segmentation, Liu et. al., AAAI 2023. (Table 1, colon tumor dice is around 0.55). The changing modalities and the need to adapt from CT to MRI (and vice versa) further aggravate this problem.
> **So the size and shape of the organs drive the absolute scores in each session.**
>
> **B)** The amount of performance retained in later incremental sessions is inherently influenced by the base model’s initial accuracy. When a model starts with a very high Dice score (close to 1.0), even a moderate relative drop results in a high absolute score being preserved. In contrast, models that begin with a lower baseline performance (e.g., Dice 0.6–0.7) have less margin to retain accuracy, and even small degradations result in noticeably lower absolute scores. **This difference in starting performance, therefore, plays a significant role in interpreting retention across incremental sessions.** Most of the FoSSIL benchmarks have Dice scores around 0.6 to 0.7.
>
> **(C)** Most existing methods in medical continual segmentation (example: *Continual Learning for Abdominal Multi-Organ
> and Tumor Segmentation, Zhang et. al., MICCAI 2023*) perform only a few incremental sessions, whereas the proposed FoSSIL stress tests the model for up to six incremental sessions with multiple classes and domains. **So the number of incremental sessions drives the absolute scores in each session**.
>
> ### Autonomous Driving
>
> **A)** Recent foundation models show performance in the range of 50–60% (IoU) on autonomous driving datasets such as BDD and Cityscapes, as reported in *AD SAM: Fine Tuning the Segment Anything Vision Foundation Model for Autonomous Driving Perception (Camarena et al., 2025)*. Notably, this result is obtained using a ViT-H backbone ( around 630M parameters), which is significantly stronger than the ViT-B ( around 90M parameters) backbones used by FoSSIL for autonomous driving. **The inherent nature and complexity of these autonomous-driving datasets largely determine the absolute performance.**
>
> **B)** **The absolute performance of state-of-the-art methods in continual semantic segmentation is typically in the range of 20–50% IoU**, as reported in *Learning With Style: Continual Semantic Segmentation Across Tasks and Domains by Toldo et al., IEEE TPAMI 2024*. Specifically, Table IX on the IDD, BDD, and Cityscapes (CS) datasets highlights that even strong baselines operate within this performance band under realistic continual learning conditions. Additionally, FoSSIL operates under several other challenging conditions, including the few-shot regime, where the scarcity of annotated examples further amplifies the difficulty of continual semantic segmentation.
>
> ### 2D robotic surgery
>
> This is an **extremely challenging and highly practical problem**, where organs and surgical tools must be segmented across sessions spanning different domains. As shown in Table I of *Privacy-Preserving Synthetic Continual Semantic Segmentation for Robotic Surgery by Xu et al., IEEE TMI 2024*, the absolute IoU achieved by existing methods in such settings ranges only from 5–30% for incremental classes, underscoring the inherent difficulty of the task.
>
> Please note that continual semantic segmentation is still an evolving research area, unlike continual classification, where higher absolute scores are generally observed, due to the greater complexity of segmentation tasks.
>
> Nevertheless, we believe that **FoSSIL is able to retain fairly high absolute scores, even when almost all baselines collapse to zero, either using a simple encoder–decoder U-Net model or with a somewhat bigger backbone like SAM (Vit-B) while the model is subjected to CIL, DIL, and few-shot challenges simultaneously.**
>
> | Method                       | Session 0 | Session 1 | Session 2 | Session 3 |
> |------------------------------|-----------|-----------|-----------|-----------|
> | FoSSIL                       | 0.736     | 0.460     | 0.329     | 0.398     |
> | FoSSIL (with unlabeled data) | 0.736     | 0.554     | 0.445     | 0.414     |
>
> **(cont...)**

---

> ### Author Response · Authors · 2025-11-28
> **Response to reviewer WWNg**
>
> | Model       | Base | Session 1 | Session 2 |
> |-------------|------|-----------|-----------|
> | SAM Vanilla | 66.00 | 32.60    | 30.81     |
> | SAM FoSSIL  | 66.00 | 33.20    | 31.22     |
>
> ### Practical considerations
>
> - Role of unlabeled data: Domain-relevant unlabeled data plays a key role, as observed earlier in Session 4 when BraTS (MRI modality) was introduced. Unlabeled data helps augment novel few-shot classes, enables the model to retain modality and domain-specific characteristics when needed, and is far cheaper to acquire than labeled data. It helps to significantly improve performance (0.025 => 0.058 => 0.197).
>
> - Role of $\varepsilon$: Based on the practical application, $\varepsilon$ can be selected (using a small validation dataset), as preliminary observations indicate that it can help boost performance. We thank the reviewer for helping us identify this.
>
> - Total drop %: Total Drop (%) measures the cumulative forgetting across all sessions and is
> more informative than absolute accuracy when models are updated incrementally.
> In real-world settings, medical imaging (scanner/protocol shifts), autonomous
> driving (weather/city changes), robotics (task-by-task updates), or any
> continual deployment models may recover at the final session, making absolute
> scores misleading. Total Drop (%) captures all intermediate collapses and
> reflects true stability over time.
>
> $$
> \text{Total Drop} = \left( \frac{\sum_i \max(0, S_i - S_{i+1})}{S_0} \right) \times 100
> $$
>
> | Method             | Session 0 | Session 1 | Session 2 | Total Drop (%) (Lower is better) |
> |--------------------|-----------|-----------|-----------|----------------|
> | PIFS               | 0.700     | 0.129     | 0.078     | 88.9%          |
> | NC-FSCIL           | 0.394     | 0.077     | 0.081     | 80.4%          |
> | CLIP-CT            | 0.475     | 0.186     | 0.141     | 70.3%          |
> | MiB                | 0.700     | 0.271     | 0.096     | 86.3%          |
> | MDIL               | 0.779     | 0.115     | 0.097     | 87.5%          |
> | C-FSCIL            | 0.787     | 0.334     | 0.297     | 62.3%          |
> | SoftNet            | 0.820     | 0.305     | 0.146     | 82.2%          |
> | GAPS               | 0.700     | 0.334     | 0.253     | 63.8%          |
> | FSCIL-SS           | 0.700     | 0.115     | 0.089     | 87.3%          |
> | Subspace           | 0.257     | 0.054     | 0.040     | 84.4%          |
> | Gen-Replay         | 0.700     | 0.076     | 0.102     | 89.1%          |
> | FeCAM              | 0.700     | 0.048     | 0.042     | 94.0%          |
> | FACT               | 0.357     | 0.071     | 0.028     | 92.2%          |
> | MAML               | 0.700     | 0.001     | 0.059     | 99.9%          |
> | MAML + Reg.        | 0.700     | 0.001     | 0.062     | 99.9%          |
> | MTL                | 0.700     | 0.079     | 0.088     | 88.7%          |
> | UnSupCL            | 0.700     | 0.039     | 0.088     | 94.4%          |
> | SupCL              | 0.700     | 0.058     | 0.042     | 94.0%          |
> | UnSupCL-HNM        | 0.700     | 0.035     | 0.068     | 95.0%          |
> | **FoSSIL (U-Net)** | **0.736** | **0.460** | **0.398** | **45.9%**       |
>
> This clearly shows that on practical metrics such as Total Drop (%), **FoSSIL maintains the best performance compared to prominent baselines. Given its negligible computational overhead, FoSSIL is highly suitable for practical deployment.**
>
> A few more practical considerations are discussed in “Response to Reviewer t59L.”
>
> ## **How are τ_conf and τ_sim selected/tuned? Are they shared across datasets? Please provide threshold sensitivity and retention/coverage statistics.**
>
> We analysed the pseudo labels selected by the model using the confidence
> threshold $\tau_{\text{conf}}$ and the similarity threshold $\tau_{\text{sim}}$.
> Retention (\%) denotes the proportion of pseudo labels that pass both filters.
> Our goal is to retain a set of pseudo labels that is both sufficiently large
> and highly reliable.
>
> A lower threshold such as $\tau_{\text{conf}} = 0.6$  or $\tau_{\text{sim}} = 0.6$  produced high
> retention but admitted many low-confidence pseudo labels, which would introduce
> noise into training. Very high confidence thresholds, on the other hand, were
> too strict and led to extremely low retention.
>
> The setting $\tau_{\text{conf}} = 0.7$ and $\tau_{\text{sim}} = 0.7$ provided
> the best trade-off: it preserved a reasonable number of pseudo labels while
> maintaining high reliability. We discarded the alternative
> $\tau_{\text{conf}} = 0.6$ and $\tau_{\text{sim}} = 0.7$ because it offered only
> a marginal 0.4% increase in retention relative to
> $\tau_{\text{conf}} = 0.7$ and $\tau_{\text{sim}} = 0.7$, but at the expense of
> lower confidence.
>
> **(cont....)**

---

> ### Author Response · Authors · 2025-11-28
> **Response to reviewer WWNg**
>
> Therefore, we selected $\tau_{\text{conf}} = 0.7$ and $\tau_{\text{sim}} = 0.7$
> as the most effective balance between pseudo-label quantity and quality.
>
> | **τ_conf** | **τ_sim** | **Retention (%)** | **Decision** |
> | ---------- | --------- | ----------------- | ------------ |
> | 0.6        | 0.6       | 71.7              | Rejected     |
> | 0.6        | 0.7       | 23.6              | Rejected     |
> | 0.6        | 0.8       | 1.8               | Rejected     |
> | 0.7        | 0.6       | 73.7              | Rejected     |
> | 0.7        | 0.7       | 23.2              | **Selected** |
> | 0.7        | 0.8       | 1.9               | Rejected     |
> | 0.8        | 0.6       | 72.6              | Rejected     |
> | 0.8        | 0.7       | 22.7              | Rejected     |
> | 0.8        | 0.8       | 1.8               | Rejected     |
>
> A similar setting is suggested by *Dual Mean-Teacher: An Unbiased Semi-Supervised Framework for Audio-Visual Source Localization, Guo et. al., NeurIPS, 2023.*
>
> $\tau_{\text{conf}} = 0.7$ and $\tau_{\text{sim}} = 0.7$ are shared across the datasets.
>
> We performed sensitivity analysis with $\tau_{\text{conf}} = 0.8$ and $\tau_{\text{sim}} = 0.7$ \& $\tau_{\text{conf}} = 0.7$ and $\tau_{\text{sim}} = 0.75$ and found that the model is robust to these variations.
>
> - $\tau_{\text{conf}} = 0.8$ and $\tau_{\text{sim}} = 0.7$ => Dice 0.562
>
> - $\tau_{\text{conf}} = 0.7$ and $\tau_{\text{sim}} = 0.75$ => Dice 0.567
>
> - $\tau_{\text{conf}} = 0.7$ and $\tau_{\text{sim}} = 0.7$ => Dice 0.554

---

### Official Review · Reviewer_t59L · 2025-11-01

**Soundness:** 2
**Presentation:** 3
**Contribution:** 2
**Rating:** 4
**Confidence:** 4

**Summary:**

This paper tries to study continual learning in a complex setting that includes class-incremental, domain-incremental, and few-shot learning. To this end, the authors build a benchmark and propose a method.

**Strengths:**

Pros:
- Continual learning is indeed an important topic in the field.
- The paper is well written and nicely organized. I can tell the authors put a lot of effort into polishing it, and I appreciate that.
- The experiments cover a wide range of methods, which is nice to see.

**Weaknesses:**

Cons:
- Honestly, I don't really like this kind of work. It feels like a mixture of everything — several settings thrown together without a clear focus. While this may have some meaning academically, in real-world scenarios (e.g., in industry), people usually prefer to train a specialized model rather than deal with such a complicated continual setup.
- Even if we accept the setting, the experimental analysis is not very systematic. The paper doesn't really explore how different setups affect different models, nor does it provide useful insights for choosing models in practice.
- The paper starts with a quote from Confucius: "I hear and I forget. I see and I remember. I do and I understand." Who is Confucius in this context? Is he a machine learning expert? I don't quite get the connection between this quote and the paper's content.
- The introduction could be smoother in logic. For example, when guided noise injection first appears, I didn't understand why it suddenly shows up or how it relates to the context. Please consider improving the narrative flow there.
- As far as I know, many important medical imaging modalities are 2D. Why aren't they included in the experiments?
- I appreciate that the authors included many methods in the benchmark, but they’re mostly from 2020–2023. There's no mention of newer work (2024–2025), including recent arXiv papers. This weakens the experimental conclusions to some extent.
- Minor issues: e.g., in the introduction, the numbering goes (i)–(ii)–(ii)–(iii), which should be fixed.

**Questions:**

See Weaknesses.

---

> ### Author Response · Authors · 2025-11-25
> **Response to reviewer t59L**
>
> We thank the reviewer for taking the time to review our paper and for providing highly valuable comments. We also appreciate the positive remarks regarding the quality of our writing and the thoroughness of our experiments. We are pleased that our efforts to present a clear and comprehensive research work are recognized.
>
>
> ## **It feels like a mixture of everything — several settings thrown together without a clear focus. While this may have some meaning academically, in real-world scenarios (e.g., in industry), people usually prefer to train a specialized model rather than deal with such a complicated continual setup.**
>
> Continual learning has rapidly evolved in recent years, with numerous works exploring diverse problem settings such as class-incremental learning (CIL), domain-incremental learning (DIL), and data-scarce few-shot settings. More recently, several works have also begun to incorporate unlabeled data-driven continual setups to better reflect realistic training conditions. In this context, the setting adopted by FoSSIL aligns closely with what recent literature characterizes as a practical and future-oriented continual learning paradigm.
>
> - *Zhang et al., Few-shot Class Incremental Learning for Classification and Object Detection: A Survey (IEEE TAPMI 2025)*, identifies several practical settings in FSCIL, including **cross-domain FSCIL**, which combines **class increment**, **domain increment**, and **few-shot constraints**. They note that **real-world domain shifts**, such as changes in imaging conditions or environment, require models to be robust under cross-domain variation. This is directly reflected in all our proposed benchmarks. The survey also points out that the no repetition constraint commonly used in FSCIL is unrealistic because recurrence of classes is common in practical and industry scenarios. Our Med FoSSIL Mixed benchmark explicitly supports such recurrence. Further, the survey highlights **semi-supervised FSCIL** as a **real-world and practical setting**, which is covered in our Semi-Supervised Natural FoSSIL benchmarks.
>
> - *Yuan et al., A Survey on Continual Semantic Segmentation (IEEE TAPMI 2024)*, emphasizes in future work the importance of **cross-modality incremental adaptation**, noting its relevance for **real-world** and open-world understanding. They outline the challenge of maintaining compatibility between new and old knowledge when working with strongly differing modalities, which our medical benchmarks already incorporate (CT/MRI). In the section on data-free methods, the survey encourages the use of foundation models such as SAM and CLIP to support **few-shot continual semantic segmentation**, which we have also integrated. The survey further highlights autonomous driving as a practical application area for both **class incremental and domain incremental learning**.
>
> - *Xu et al., Privacy Preserving Synthetic Continual Semantic Segmentation for Robotic Surgery (IEEE TMI 2024)*, a practical work in robotic surgery, performs **continual semantic segmentation under data-scarce and few-shot conditions** and identifies **incremental domain adaptation** as an important future direction.
>
> - *Kwak et al., Towards Realistic Incremental Scenario in Class Incremental Semantic Segmentation (CoLLAs 2024)*, stress the growing demand from **industry** for realistic incremental scenarios and note that CIL research has expanded toward setups with additional data constraints, including **few-shot class incremental semantic segmentation**, which our benchmarks also support.
>
> - *Zhu et al., Class Incremental Medical Image Segmentation via Prototype Guided Calibration and Dual Aligned Distillation (AAAI 2026)*, highlight the need for generalization across **diverse domains under continual learning**, a limitation that our medical benchmarks already address.
>
> - *Xue et al., Toward Few-Shot Learning in the Open World: A Review and Beyond (IEEE TPAMI 2025)*, emphasizes that **incremental few-shot** and **cross-domain few-shot learning** reflect the dynamic nature of the open world and help bridge the gap between human and machine learning. This further supports the practical and real-world relevance of the settings adopted in our benchmarks.
>
> The future work sections of these recent references consistently highlight the need for a framework like FoSSIL, one that addresses real-world constraints such as class incremental learning, domain incremental variation, and data-scarce conditions, while also using unlabeled data to improve performance. This combination represents a highly realistic and practically important setting, and solving it is essential for progress in both research and industry, as identified by all other reviewers as well.
>
> **(cont...)**

---

> ### Author Response · Authors · 2025-11-25
> **Response to reviewer t59L**
>
> ## **It feels like a mixture of everything — several settings thrown together without a clear focus. While this may have some meaning academically, in real-world scenarios (e.g., in industry), people usually prefer to train a specialized model rather than deal with such a complicated continual setup.**
>
> We monitor the performance of specialized models, which are well-known to handle different constraints in practical applications like 3D medical segmentation, autonomous driving, and 2D robotic surgery.
>
> ### 1. Class-incremental learning
>
>   - For the 2D robotic surgery task (CAT) [1], we designed a setting where only the classes change across sessions, as this method is specifically suited for such scenarios. We used the CholecSeg8k dataset [2], which contains three distinct groups of classes within the same surgical domain. The base session (Session 0) includes larger organs, Session 1 introduces smaller and thinner structures such as veins, and Session 2 contains surgical instruments. This results in a highly practical and challenging setup that reflects real robotic surgery conditions, requiring segmentation of organs and instruments of widely varying sizes and shapes.
>
>  |  Model              | Session 0 | Session 1 | Session 2 |
>  |---------------------|--------|-----------|-----------|
>  | UNet Vanilla        | 0.770  | 0.003     | 0.000     |
>  | Medformer Vanilla   | 0.762  | 0.026     | 0.002     |
>  | CAT – UNet          | 0.770  | 0.102     | 0.006     |
>  | FoSSIL – Medformer  | 0.762  | 0.166     | 0.144     |
>  | FoSSIL – UNet       | 0.770  | 0.212     | 0.159     |
>
> - CLIP [3,4] is a widely used continual learner with strong few-shot and zero-shot capabilities. We used the CLIP variant from [5], which is adapted for 3D medical segmentation, and applied it to a class incremental setting with novel classes having scarce labels from two closely related modality domains: TotalSegmentator (CT modality, Session 0) and AMOS (mostly CT, Session 1). We also included BCM [6], a method specifically designed to handle few-shot class incremental learning in semantic segmentation.
>
> | Model               | Session 0 | Session 1 |
> |---------------------|--------|-----------|
> | CLIP [5]  | 0.475 | 0.186 |
> |BCM| 0.700  | 0.014 |
> |FoSSIL| 0.736 | 0.460|
>
> ### 2. Domain incremental learning
>
> - Designed for adaptability, SAM generalizes well across diverse segmentation tasks and can handle unseen objects and domains with zero-shot capability. This makes it highly valuable for autonomous driving [7]. We create a setting where only the domain varies (BDD Session 0) to (IDD Session 1) with the same classes in both sessions. Please note that SAM is already pretrained on 1-Billion mask dataset (SA-1B).
>
> | Model               | Session 0 | Session 1 |
> |---------------------|--------|-----------|
> | SAM  | 66.0 | 30.43 |
> |FoSSIL| 66.0  | 30.64 |
>
> - Since CLIP [4] is an effective continual learner capable of handling domain incremental learning and suitable for few-shot/zero-shot settings, we used the specialized CLIP model from [8] to adapt to novel domains while keeping the classes unchanged. In our setup, Session 0 uses the AMOS domain, and Session 1 uses a mixture of BCV and MOTS domains, while the class set remains the same across sessions. It is important to note that the CLIP model in [8] is already pretrained on medical domains, which makes it particularly well-suited for this task.
>
> | Model               | Session 0 | Session 1 |
> |---------------------|--------|-----------|
> | CLIP [8]  | 0.716 | 0.392 |
> |FoSSIL| 0.687  | 0.405 |
>
> ### 3. Class and domain incremental learning
>
> MDIL [9] is specifically designed to handle class incremental and domain incremental learning in autonomous driving. We evaluated MDIL on Session 0 (BDD) and Session 1 (IDD), where both the classes and the domains vary.
>
> | Model               | Session 0 | Session 1 |
> |---------------------|--------|-----------|
> | MDIL  | 48.54 |1.59 |
> |FoSSIL| 47.76  | 24.54 |
>
> ### 4. Semi-supervised Class-incremental learning
>
>  UaD-CE [10] is specifically designed to handle class incremental learning with scarce class labels and can effectively use unlabeled data. We applied it to a class incremental setting where the novel classes have limited labelled samples, using two closely related modality domains: TotalSegmentator (CT modality, Session 0) and AMOS (mostly CT, Session 1). The model is also provided with unlabeled data, as it is designed to leverage it.
>
> | Model               | Session 0 | Session 1 |
> |---------------------|--------|-----------|
> | UaD-CE  | 0.700 | 0.082 |
> |FoSSIL| 0.736  | 0.554 |
>
> **(cont...)**

---

> ### Author Response · Authors · 2025-11-25
> **Response to reviewer t59L**
>
> The above settings clearly show that **models specifically designed for class incremental learning, domain incremental learning, data scarcity, or the use of unlabeled data still fail to perform reliably**. In contrast, the **FoSSIL framework is able to handle each of these setups individually**, as well as the more complex continual scenarios captured in the FoSSIL benchmarks. This demonstrates that the *FoSSIL framework can effectively address CIL, DIL, few-shot settings, and unlabeled data both individually and in combination*. We thank the reviewer for motivating us to examine the strengths of FoSSIL more thoroughly in individual setups, too.
>
>
> [1] Privacy-Preserving Synthetic Continual Semantic Segmentation for Robotic Surgery, Xu et. al., IEEE TMI, 2024.
>
> [2] CholecSeg8k: A Semantic Segmentation Dataset for Laparoscopic Cholecystectomy Based on Cholec80, Hong et. al., 2020.
>
> [3] CLIP model is an Efficient Online Lifelong Learner, Wang et. al., 2024.
>
> [4] CLIP model is an Efficient Continual Learner, Thengane et. al., 2022.
>
> [5] Continual Learning for Abdominal Multi-Organ and Tumor Segmentation, Zhang et. al., MICCAI 2023.
>
> [6] A Surprisingly Simple Approach to Generalized Few-Shot Semantic Segmentation, Sakai et. al., NeurIPS 2024.
>
> [7] A Survey of Autonomous Driving from a Deep Learning Perspective, Zhao et. al., ACM Computing Surveys, 2025.
>
> [8] CLIP-Driven Universal Model for Organ Segmentation and Tumor Detection, Liu et. al., ICCV 2023.
>
> [9] Multi-Domain Incremental Learning for Semantic Segmentation, Garg et. al., WAVC 2022.
>
> [10] Uncertainty-Aware Distillation for Semi-Supervised Few-Shot Class-Incremental Learning, Cui et. al., IEEE TNNLS 2024.

---

> ### Author Response · Authors · 2025-11-25
> **Response to reviewer t59L**
>
> ## **The paper doesn't really explore how different setups affect different models, nor does it provide useful insights for choosing models in practice.**
>
> ### 1. 2D robotic surgery task
>
> To study the effect of each setup, we performed class incremental learning, domain incremental learning, and their combinations under data-scarce conditions. For this analysis, we evaluated widely used backbones such as MedFormer (a Transformer-based model) and U-Net for semantic segmentation.
>
> In this task, we use two domains, CholecSeg8k and m2caiseg [1]. We vary both the classes (CIL) and the domains (DIL) across sessions to segment different organs and surgical instruments, creating a challenging and realistic setting. For class incremental learning, we vary the classes within CholecSeg8k. For domain incremental learning, we shift the domain from CholecSeg8k (Session 0) to m2caiseg (Session 1). For combined class and domain incremental learning (CDIL), we vary both the domain and the class set when moving from CholecSeg8k (Session 0)  to m2caiseg (Session 1).
>
> S=>Session
> | Model               | CIL S0 | CIL S1 | CIL S2 | DIL S0 | DIL S1 | CDIL S0 | CDIL S1 | CDIL S2 |
> |---------------------|----------|--------|--------|----------|--------|-----------|---------|---------|
> | UNet Vanilla        | 0.770    | 0.003  | 0.000  | 0.770    | 0.223  | 0.770     | 0.033   | 0.007   |
> | Medformer Vanilla   | 0.762    | 0.026  | 0.002  | 0.762    | 0.166  | 0.762     | 0.011   | 0.002   |
> | CAT – UNet          | 0.770    | 0.102  | 0.006  | 0.770    | 0.258  | 0.770     | 0.166   | 0.001   |
> | FoSSIL – Medformer  | 0.762    | 0.166  | 0.144  | 0.762    | 0.279  | 0.762     | 0.169   | 0.156   |
> | FoSSIL – UNet       | 0.770    | 0.212  | 0.159  | 0.770    | 0.305  | 0.770     | 0.244   | 0.163   |
>
>
> We observe that **all models are affected, and their performance drops significantly across these settings**. Interestingly, *transformer-based models show poorer performance compared to simpler encoder-decoder models such as U-Net*.
>
> ### 2. Autonomous driving
>
> To study the effect of each setup, we performed class incremental learning, domain incremental learning, and their combinations under data-scarce conditions. For this analysis, we used SAM as the backbone. For class incremental learning, we varied the classes within BDD (Berkeley Deep Drive). For domain incremental learning, we shifted the domain from BDD (Session 0) to IDD (Indian Driving Dataset) - Session 1. For combined class and domain incremental learning (CDIL), we varied both the domain and the class set when moving from BDD (Session 0) to IDD (Session 1).
>
> S=>Session
> | Model        | CIL S0 | CIL S1 | DIL S0 | DIL S1 | CDIL S0 | CDIL S1 | CDIL S2 |
> |--------------|----------|--------|----------|--------|-----------|---------|---------|
> | SAM Vanilla  | 66.0       | 30.49  | 66.0       | 30.43  | 66.0        | 32.6    | 30.81   |
> | SAM FoSSIL   | 66.0       | 31.39  | 66.0       | 30.64  | 66.0        | 33.2    | 31.22   |
>
> Even heavily pretrained models such as **SAM show a noticeable drop in performance when evaluated under class incremental and domain incremental setups**.
>
> ### 3. FoSSIL benchmarks
> It is observed that *feature replay and prototype-based methods, such as C-FSCIL, SoftNet, and our FoSSIL framework, as well as data synthesis-based methods like GAPS, perform significantly better under these constraints, as shown in Table 2*. As noted earlier, the performance of transformers such as Medformer and SwinUNetr is often similar to or worse than simpler models like UNet under the multi-constrained FoSSIL benchmarks (Table 5 and Table 7). A notable exception arises under severe domain shift: in Table 7, when Session 4 introduces the Brats domain, which is MRI, while the previous sessions are CT or mostly CT (for example, AMOS), U-Net experiences a substantial drop in performance even with FoSSIL, whereas MedFormer with FoSSIL retains much stronger performance. **This highlights that certain transformer architectures can be more resilient to extreme modality shifts**.
> *Pretrained models such as CLIP (Table 5) and SAM (Table 4) generally show better resistance to the constraints than lighter models like UNet or DeepLab*, or even some transformers. However, their performance still declines, reflecting the difficulty and severity of the FoSSIL benchmarks. **Existing semi-supervised methods remain unreliable** in these continual setups and are often unsuitable.
> FoSSIL, on the other hand, makes effective use of unlabeled data to improve performance. As shown in Table 7, the Session 1 score increases from 0.460 (Table 3, Session 1) to 0.554 when unlabeled data is incorporated through the FoSSIL framework. This is important because unlabeled data is inexpensive and broadly accessible, and can be sourced from any publicly available dataset that aligns with the task.
>
> **(cont..)**

---

> ### Author Response · Authors · 2025-11-25
> **Response to reviewer t59L**
>
> ## **The paper doesn't really explore how different setups affect different models, nor does it provide useful insights for choosing models in practice.**
>
> We further improve U-Net performance in Session 4 (Table 7) by using publicly available unlabeled data from a variety of medical domains, including AMOS, TotalSegmentator, BCV (or any other similar domain unbaled data can be used). Since FoSSIL allows the use of any amount of unlabeled data, this results in a meaningful improvement, raising performance from 0.0576 to 0.197, which is a substantial gain.
>
> ### How to pick the models in practice:
>
> First, across the FoSSIL benchmarks, all models and even the strongest existing methods show a sharp drop in performance. A practical guideline:
>
> - When resources permit, pretrained models combined with FoSSIL should be preferred, as they generally offer better performance and robustness.
>
> - When the budget is moderate, U-Net-based models or transformer-based models are reasonable choices with the FoSSIL framework for continual semantic segmentation. *Transformers provide better resistance to severe domain shifts*, while UNet-based models remain competitive in standard settings.
>
> - *For lightweight backbones, the most effective strategy is to use FoSSIL with unlabeled data*, which is easy to obtain and leads to noticeable performance gains. *Lighter models also become more suitable when there are many continual sessions or frequent updates*, since unlabeled data can be collected far more easily than labelled data.
>
> - For extreme data-scarce settings, where K equals 4 or 3 (K denotes few-shot samples), the performance for Session 1 in Med FoSSIL Disjoint is 0.434299 and 0.41405, respectively. This remains better than the baselines and is close to the performance with K equal to 5, which is 0.460 as shown in Table 2 and Table 3. This makes FoSSIL a very practical setting in extremely scarce data settings.
>
>
> [1] m2caiSeg: Semantic Segmentation of Laparoscopic Images using Convolutional Neural Networks, Maqbool et. al., 2020.
>
> ## **The paper starts with a quote from Confucius: "I hear and I forget. I see and I remember. I do and I understand." Who is Confucius in this context? Is he a machine learning expert? I don't quite get the connection between this quote and the paper's content.**
> The Confucius quote is a philosophical metaphor, not a reference to an ML expert. It loosely mirrors continual learning: “I hear and I forget” reflects catastrophic forgetting, “I see and I remember” reflects overfitting to recent few-shot data, and “I do and I understand” reflects stable learning across sessions, which FoSSIL aims to achieve. But this connection is conceptual, not technical. The quote offers metaphor, but no technical relevance, so removing it makes the introduction clearer and better aligned with the paper’s technical focus. Thank you for pointing it out.
>
> ## **The introduction could be smoother in logic. For example, when guided noise injection first appears, I didn't understand why it suddenly shows up or how it relates to the context. Please consider improving the narrative flow there.**
> Thank you for pointing that out. Corrected in the paper.
>
> ## **As far as I know, many important medical imaging modalities are 2D. Why aren't they included in the experiments?**
> We designed a 2D robotic surgery benchmark using two domains, CholecSeg8k and m2caiseg, across three sessions. We vary both the classes and the domains to segment different organs and surgical instruments, creating a challenging and realistic setting under a few-shot data regime. The base session (Session 0) includes larger organs, Session 1 introduces tubular structures such as the intestine, and Session 2 contains surgical instruments. In the incremental sessions, we use 50 samples per novel class. This benchmark is difficult due to the wide variation in the size, shape, and appearance of the objects to be segmented. Even under these constraints, FoSSIL achieves significantly better performance than CAT, which is a specialized baseline for this task.
>
> | Model               | Session 0   | Session 1 | Session 2 |
> |---------------------|--------|-----------|-----------|
> | UNet Vanilla        | 0.770  | 0.033     | 0.007     |
> | Medformer Vanilla   | 0.762  | 0.011     | 0.002     |
> | CAT – UNet          | 0.770  | 0.166     | 0.001     |
> | FoSSIL – Medformer  | 0.762  | 0.169     | 0.156     |
> | FoSSIL – UNet       | 0.770  | 0.244     | 0.163     |

---

> ### Author Response · Authors · 2025-11-25
> **Response to reviewer t59L**
>
> ## **There's no mention of newer work (2024–2025), including recent arXiv papers.**
> We have included 8 new baselines from 2024-2025 as below (Table 2 and 7):
> - BCM - A Surprisingly Simple Approach to Generalized Few-Shot Semantic Segmentation, Sakai et. al., NeurIPS 2024.
>
> Handles, class incremental, few-shot setups.
>
> - UCB - Enhancing Continual Semantic Segmentation via Uncertainty and Class Balance Re-Weighting, Liang et. al., IEEE TIP, 2025.
>
> Handles, class incremental.
>
> - YoooP: You Only Optimize One Prototype per Class for Non-Exemplar Incremental Learning, Kong et. al., TMLR 2025.
>
> Prototype-based.
>
> - STAR: Stability Inducing Weight Perturbation for Continual Learning, Eskandar et. al., ICLR 2025.
>
> Regularisation-based (weight perturbation).
>
> - C-Flat - Make Continual Learning Stronger via C-Flat, Bian et. al., NeurIPS 2024.
>
> Sharpness-aware minimization.
>
> - CAT - Privacy-Preserving Synthetic Continual Semantic Segmentation for Robotic Surgery, Xu et. al., IEEE TMI 2024.
>
> 2D Surgery baseline.
>
> - CSL - When Confidence Fails: Revisiting Pseudo-Label Selection in Semi-supervised Semantic Segmentation, Liu et. al., ICCV 2025.
>
> Handles unlabeled data.
>
> - Adapt_replay - Adaptive Prototype Replay for Class Incremental Semantic Segmentation, Zhu et. al., AAAI 2025.
>
> Prototype-based.
>
>
>
> ### Med FoSSIL-Disjoint Benchmark
> | Method        | Session 0 | Session 1 | Session 2 |
> |---------------|-----------|-----------|-----------|
> | C-Flat        | 0.700     | 0.174     | 0.030     |
> | STAR          | 0.700     | 0.050     | 0.020     |
> | YoooP         | 0.700     | 0.176     | 0.028     |
> | UCB           | 0.700     | 0.267     | 0.127     |
> | BCM           | 0.700     | 0.014     | 0.000     |
> | Adapt_replay | 0.700     | 0.044     | 0.027     |
> | FoSSIL        | 0.736     | 0.460     | 0.398     |
>
> ### Med Semi-Supervised-FoSSIL (MedFormer)
> | Method | Session 0 | Session 1 | Session 2 | Session 3 | Session 4 | Session 5 |
> |--------|-----------|-----------|-----------|-----------|-----------|-----------|
> | CSL    | 0.659     | 0.040     | 0.020     | 0.000     | 0.000     | 0.000     |
> | FoSSIL | 0.640     | 0.431     | 0.368     | 0.335     | 0.323     | 0.293     |
>
> ## **Minor issues: e.g., in the introduction, the numbering goes (i)–(ii)–(ii)–(iii), which should be fixed.**
> Thank you for pointing that out. Corrected in the paper.

---

### Author Response · Authors · 2025-11-28
**Summary**

## Dear Reviewers,

We thank you for your constructive and positive comments. We are especially grateful because each reviewer identified valuable directions that allowed us to further strengthen and refine the FoSSIL framework.

Here we summarize the highlights of the rebuttal:

- Established the motivation for FoSSIL, explaining why it is needed in current continual semantic segmentation and why it is practical and relevant.

- Analyzed and compared the performance of FoSSIL on benchmarks for CIL, DIL, and data scarcity constraints separately.

- Analyzed and compared specialized models with FoSSIL.

- Provided practical guidance on selecting models and discussed how each setup (CIL, DIL, and data scarcity constraints) affects model behavior.

- Demonstrated the performance of the FoSSIL framework on a 2D robotic surgery task.

- Compared FoSSIL with eight relevant methods released in 2024 to 2025.

- Added a figure in the paper illustrating the overall FoSSIL framework.

- Conducted sensitivity and hyperparameter analysis for guided noise injection.

- Explained the causes of abrupt performance drops and discussed ways to address them.

- Performed a computational analysis of the FoSSIL framework.

- Explained practical aspects of the FoSSIL framework in different settings and applications, along with interesting future directions.

- Provided analysis of the hyperparameters used in the pseudo-label refinement method.

- Highlighted the novel aspects of the FoSSIL framework and clarified how it differs from existing approaches.

- Presented a theoretical analysis of guided noise injection and its role in handling stability and plasticity.

- Discussed certain limitations of the FoSSIL framework.

- Analyzed and compared FoSSIL with several other regularization based, weight perturbation, and noise injection methods, both theoretically and empirically.

- Examined the implementation of guided noise injection at different layers of the model.

We kindly request the reviewers to let us know if any additional points need clarification. Thank you.

---

> ### Author Response · Authors · 2025-12-04
> **Post-rebuttal paper changes**
>
> ## **Dear AC's**
>
> The modified or newly added sections, lines, tables, captions, and figures are marked in **red**.
> We mark all relevant parts in the uploaded paper where reviewer comments have been addressed.
>
> ## Reviewer t59L
> **1. It feels like a mixture of everything — several settings thrown together without a clear focus. While this may have some meaning academically, in real-world scenarios (e.g., in industry), people usually prefer to train a specialized model rather than deal with such a complicated continual setup.**
> - Lines - 068 - 071 (references highlight why FoSSIL is important in real world)
> - Appendix L (ADDITIONAL ABLATIONS) highlights how specialized models fail
>
> **2. The paper doesn't really explore how different setups affect different models, nor does it provide useful insights for choosing models in practice.**
> - Lines 413-416, Lines 427-429 (shows how Transformer/U-Net perform), Lines 456-459
> - Appendix K.1
> - Appendix L
> - Appendix M (specifically M.1 and M.4)
>
> **3. The paper starts with a quote from Confucius: "I hear and I forget. I see and I remember. I do and I understand." Who is Confucius in this context? Is he a machine learning expert? I don't quite get the connection between this quote and the paper's content.**
> Figure 1 and the Confucius quote from the previous submission are removed as they do not convey much technical information
>
> **4. The introduction could be smoother in logic. For example, when guided noise injection first appears, I didn't understand why it suddenly shows up or how it relates to the context. Please consider improving the narrative flow there.**
> - Lines 096-102
>
> **5. As far as I know, many important medical imaging modalities are 2D. Why aren't they included in the experiments?**
> - Appendix G (2D robotic surgery)
> - Table 17 and 29
>
> **6. There's no mention of newer work (2024–2025), including recent arXiv papers.**
> - Table 2, 7 and 29.
>
> **7. Minor issues: e.g., in the introduction, the numbering goes (i)–(ii)–(ii)–(iii), which should be fixed.**
> - Line 118-127
>
> ## Reviewer WWNg
> **1. The paper lacks a clear pipeline/architecture diagram, which would make the method easier to grasp at a glance.**
> - Added Figure 3.
>
> **2. The noise injection module has no analysis of hyperparameters or other strategies (e.g., sensitivity and robustness studies).**
> - Line 469-470
> - Table 8
> - Appendix D
>
> **3. In several tables (Table 2, Table 4, Table 6, Table 7), multiple methods report identical Session-0 results; the authors should explain why.**
>
> - clarified at Lines 410-411
>
> **4. In Table 3, FoSSIL (U-Net) drops to 0.025 at Session 4 and then rebounds to 0.324 at Session 5; this large fluctuation should be verified (typesetting/statistics) or clearly explained.**
>
> - Mentioned at line 428, lines 457-459
> - detailed in Appendix K.1
>
> **5. The paper should report computational costs, including training/inference time and memory/parameter overhead.**
>
> - Table 9 (Lines 483-485)
>
> **6. Although FoSSIL improves over multiple sessions, the absolute Dice/IoU remains low, which may limit practical applicability; this seems at odds with the paper’s motivation and should be discussed.**
>
> - Detailed discussion in Appendix K.1, K.3, K.4, K.5
> - Details in Appendix M
>
> **7. How are τ_conf and τ_sim selected/tuned? Are they shared across datasets? Please provide threshold sensitivity and retention/coverage statistics.**
>
> - Lines 470-474
> - Appendix E
> - Table 11
>
> ## Reviewer y7bk
>
> **1. The contribution is mainly in integration and benchmarking, not in introducing fundamentally new algorithms.**
>
> - Highlighted at: Lines 254-258; Lines 301-305
> - GNI as a novel algorithm (Appendix A \& B), Appendix J
> - Pseudo-Label refinement contribution Appendix C
>
> **2. The paper does not analyze why the guided noise injection helps beyond empirical performance. No theoretical link to stability–plasticity balance is made.**
>
> - Added section 3.5
> - Appendix A \& B
>
> **3. Some baselines were not originally designed for few-shot or semi-supervised continual segmentation (e.g., MiB, MDIL), which could exaggerate FoSSIL’s relative advantage.**
>
> - All compared baselines and their categories are mentioned in Appendix I (IMPLEMENTATION DETAILS)
>
> **4. Details of data splits, unlabeled data sampling, and hyperparameter tuning are missing.**
>
> - Already available in Appendix G (FOSSIL BENCHMARKS)
> - Also available through https://github.com/anony34/FoSSIL
>
> **5. The work lacks qualitative examples or discussions of cases where FoSSIL fails, such as severe domain shifts or noisy unlabeled data.**
>
> - Added in Appendix K
>
> **6. The paper repeats motivation and design explanations across sections, and some figures (e.g., Figure 2–4) are not deeply analyzed.**
>
> - Corrected in the paper
> - Figures detailed at lines 059-064; lines 112-117; lines 460-468; lines 423-428
>
> **7. How is guided noise injection different from existing gradient-based regularization (e.g., SAM, weight perturbation)?**
>
> - Added in Appendix J (J.6, J.7)
>
> **(cont..)**

---

> ### Author Response · Authors · 2025-12-04
> **Post-rebuttal paper changes**
>
> - Table 20
>
> **8. Are prototypes recomputed from all sessions or updated incrementally?**
>
> - Line 215
>
> **9. What is the additional computational overhead compared to vanilla training?**
>
> - Lines 483-485
> - Table 9
>
> ## Reviewer  w3yv
>
> **1. The paper needs to provide a discussion comparing GNI to other adaptive regularization schemes or existing noise injection methods (e.g., standard weight decay, dropout, or variational dropout) and justify why this specific gradient-based formulation is superior.**
>
> - Detailed in Appendix J
> - Table 20
> - Appendix I (IMPLEMENTATION DETAILS) - Regularization-based methods
>
> **2. The paper lacks sensitivity analysis for the thresholds in PLR ($\tau_{conf}$ and $\tau_{sim}$). It is unclear how these were "empirically determined" or how robust the model is to their variation.**
>
> - Lines 470-474
> - Appendix E
> - Table 11
>
> **3. GNI is only applied to the final classifier layer $\textit{F}$. The paper needs to justify why it isn't applied to deeper feature extractor layers, which are also prone to overfitting in few-shot settings.**
>
> - Appendix F
>
> Thank you.

---

### Note · Authors · 2026-01-26

I have read and agree with the venue's withdrawal policy on behalf of myself and my co-authors.

---

### Meta-Review · Area_Chair_ypTB · 2026-01-06

**Summary:**

Across reviewers, the main concerns converge on limited conceptual novelty, insufficient analysis and justification of key design choices, and gaps in experimental rigor and clarity.

(a) While the paper integrates several known techniques (e.g., prototype replay, noise-based regularization, pseudo-label filtering) into a unified continual segmentation framework, reviewers generally view the contribution as largely incremental, with the novelty lying more in integration and benchmarking than in fundamentally new ideas. In particular, the guided noise injection (GNI) mechanism is repeatedly criticized for lacking theoretical motivation or clear distinction from existing regularization or gradient-based methods, and its hyperparameters and design scope (e.g., why applied only to certain layers) are insufficiently analyzed.

(b) Experimentally, reviewers note a lack of systematic evaluation, missing sensitivity analyses, limited discussion of failure cases, unclear implementation details, and insufficient reporting of computational cost, all of which weaken the practical and scientific insights.

(c) Additional concerns include incomplete coverage of recent related work, unclear presentation (e.g., missing diagrams), and questions about the real-world relevance and applicability of the proposed continual learning setup given the relatively low absolute performance of the framework.

**Reviewer Concerns:**

The authors addressed several presentation- and evaluation-related concerns in the rebuttal. In particular, they provided a clearer pipeline diagram and additional discussion regarding practical considerations (e.g., model selection and absolute performance), which improves the overall clarity of the paper. Moreover, the rebuttal includes detailed computational cost analyses, sensitivity studies, and comparisons with more recent baselines, which partially address reviewers’ concerns about experimental rigor and validation of the proposed continual semantic segmentation framework.

However, several core concerns remain only partially addressed, particularly regarding technical novelty and theoretical grounding. (a) Although the authors provided additional theoretical analysis of Guided Noise Injection (GNI) and conceptual comparisons with other regularization-based continual learning methods, the theoretical arguments for the superiority of GNI are limited in scope and rigor. For example, the presented PAC-Bayes analysis considers only specific conditions (e.g., anisotropic versus isotropic noise under matched variance), and tighter bounds in this setting do not necessarily imply practical superiority over optimizers such as SGD with momentum, Adam/AdamW, or other curvature-aware or adaptive-noise methods, especially given the looseness of PAC-Bayes bounds in practice and the absence of considering alternative noise schedules, preconditioning, or implicit regularization effects of deterministic optimizers.
(b) The empirical evidence supporting the claimed advantages of GNI remains preliminary. Results referenced in Table 20 and in the rebuttal response to reviewer w3yv lack sufficient experimental details and appear incomplete, which weakens the fairness and interpretability of the comparison.
(c) I concur with the reviewers that the overall technical novelty remains limited. While the rebuttal clarifies the intended contributions (e.g., dual-criterion filtering and GNI), the proposed filtering mechanism appears incremental, and the motivation for preferring GNI over commonly used regularization strategies is not convincingly established, either theoretically or empirically.
(d) Finally, the paper still feels somewhat diffuse in scope, with multiple mechanisms introduced without a clearly articulated central contribution. A more focused positioning of the paper would substantially improve comprehensibility and impact.

Overall, while the rebuttal strengthens the paper in terms of clarity and empirical coverage, the key concerns regarding novelty, theoretical justification, and coherence of contributions remain insufficiently resolved.

**Reviewer Scores:**

I expect that all reviewers would likely maintain their original scores, as their primary concerns substantially overlap. While the authors addressed several issues related to experimental evaluation and presentation clarity in the rebuttal, the central concerns regarding technical novelty and the significance of the contributions remain only partially resolved.

---

### Decision · Program_Chairs · 2026-01-26

Reject